# Matrix-regulated integrin $\alpha_v\beta_5$ maintains $\alpha_5\beta_1$-dependent desmoplastic traits prognostic of neoplastic recurrence

Janusz Franco-Barraza[1], Ralph Francescone[1], Tiffany Luong[1], Neelima Shah[1], Raj Madhani[1], Gil Cukierman[1], Essel Dulaimi[2], Karthik Devarajan[3], Brian L Egleston[4], Emmanuelle Nicolas[5], R Katherine Alpaugh[6], Ruchi Malik[1], Robert G Uzzo[1,7], John P Hoffman[7], Erica A Golemis[4], Edna Cukierman[1]*

[1]Department of Cancer Biology, Fox Chase Cancer Center, Philadelphia, United States; [2]Department of Pathology, Fox Chase Cancer Center, Philadelphia, United States; [3]Department of Cancer Epigenetics, Fox Chase Cancer Center, Philadelphia, United States; [4]Department of Molecular Therapeutics, Fox Chase Cancer Center, Philadelphia, United States; [5]Programs in Genomics, Fox Chase Cancer Center, Philadelphia, United States; [6]Protocol Support Lab, Fox Chase Cancer Center, Philadelphia, United States; [7]Department of Surgical Oncology, Fox Chase Cancer Center, Philadelphia, United States

*For correspondence: edna.cukierman@fccc.edu

Competing interests: The authors declare that no competing interests exist.

**Abstract** Desmoplasia, a fibrotic mass including cancer-associated fibroblasts (CAFs) and self-sustaining extracellular matrix (D-ECM), is a puzzling feature of pancreatic ductal adenocarcinoma (PDACs). Conflicting studies have identified tumor-restricting and tumor-promoting roles of PDAC-associated desmoplasia, suggesting that individual CAF/D-ECM protein constituents have distinguishable tumorigenic and tumor-repressive functions. Using 3D culture of normal pancreatic versus PDAC-associated human fibroblasts, we identified a CAF/D-ECM phenotype that correlates with improved patient outcomes, and that includes CAFs enriched in plasma membrane-localized, active $\alpha_5\beta_1$-integrin. Mechanistically, we established that TGF$\beta$ is required for D-ECM production but dispensable for D-ECM-induced naïve fibroblast-to-CAF activation, which depends on $\alpha_v\beta_5$-integrin redistribution of pFAK-independent active $\alpha_5\beta_1$-integrin to assorted endosomes. Importantly, the development of a simultaneous multi-channel immunofluorescence approach and new algorithms for computational batch-analysis and their application to a human PDAC panel, indicated that stromal localization and levels of active SMAD2/3 and $\alpha_5\beta_1$-integrin distinguish patient-protective from patient-detrimental desmoplasia and foretell tumor recurrences, suggesting a useful new prognostic tool.

## Introduction

The mesenchymal stroma typically found in normal tissues, which is composed mainly of naïve quiescent fibroblastic cells and their secreted interstitial extracellular matrix (ECM), constitutes a natural tumor suppressive microenvironment that enforces cellular homeostasis (*Mintz and Illmensee, 1975*; *Soto and Sonnenschein, 2011*; *Xu et al., 2009*; *Bissell and Hines, 2011*; *Dolberg and Bissell, 1984*; *Petersen et al., 1992*; *Wong et al., 1992*; *Anderson et al., 2006*; *Wall et al., 2005*). By contrast, the emergence of a desmoplastic (activated) stroma, encompassing fibrotic-like modifications of local and recruited fibroblasts based on tumor interactions with the local microenvironment, has been proposed to play a major role in the development and progression of solid tumors such as pancreatic ductal adenocarcinoma (PDAC) and others (*Bijlsma and van Laarhoven, 2015*;

**eLife digest** Tumors are not entirely made out of cancerous cells. They contain many other components – referred to as tumor stroma – that may either encourage or hinder the tumor's growth. Tumor stroma includes non-cancerous cells and a framework of fibrous sugary proteins, called the extracellular matrix, which surround and signal to cells while providing physical support.

In the most common and aggressive form of pancreatic cancer, the stroma often makes up the majority of the tumor's mass. Sometimes the stroma of these pancreatic tumors can protect the cancer cells from anti-cancer drugs. Researchers have therefore been interested in finding out exactly which aspects of the tumor stroma shield and support cancer cells, and which impede their growth and progression. Answering these questions could make it possible to develop new drugs that will change a tumor-supporting stroma into one that hinders the tumor's growth and spread.

The most abundant cells in the stroma of pancreatic tumors are called cancer-associated fibroblasts. Healthy specialized fibroblasts – known as pancreatic stellate cells – help to build and maintain the 'normal' extracellular matrix and so these cells normally restrict a tumor's development. However, cancer cells can adapt healthy fibroblasts into cancer-associated fibroblasts, which produce an altered extracellular matrix that could allow the tumor to grow.

Franco-Barraza et al. have now compared healthy and cancer-associated fibroblasts from patients' pancreatic tumors. One of the main differences between these two cell types was the location of the activated form of a molecule called $\alpha_5\beta_1$-integrin. Healthy fibroblasts, in a normal extracellular matrix, have active $\alpha_5\beta_1$-integrin on the surface of the cell. However, a number of tumor-promoting signals, including some from the altered extracellular matrix, could force the active $\alpha_5\beta_1$-integrins to relocate inside the fibroblasts instead. In further experiments, where the activated integrin was retained at the cell surface, the fibroblasts were able to resist the influence of the cancer-associated extracellular matrix. Then again, if the active $\alpha_5\beta_1$-integrins were directed inside the cells, healthy cells turned into cancer-associated fibroblasts.

With this information in hand, Franco-Barraza et al. examined tumor samples from over a hundred pancreatic cancer patients using a new microscopy-based technique that distinguishes cancer cells from stroma cells. The analysis confirmed the pattern observed in the laboratory: those patients who appeared to produce more normal extracellular matrix and have active $\alpha_5\beta_1$-integrin localized mostly to the surface of the cells survived longer without the cancer returning than those patients who lacked these stroma traits. Samples from patients with kidney cancer also showed similar results and, as before, an altered extracellular matrix was linked to a worse outcome of the disease.

Together these findings suggest that if future studies uncover ways to relocate or maintain active $\alpha_5\beta_1$-integrin to the cell surface of fibroblasts they could lead to new treatments to restrict the growth of tumors in cancer patients.

Jonasch et al., 2012; Erkan et al., 2007; Xu et al., 2015; Gupta et al., 2011). Many studies suggest that a fibrotic reaction, such as that seen in pancreatitis, drives a pro-tumorigenic wound-healing response that often precedes tumor development (Whitcomb and Pogue-Geile, 2002; Binkley et al., 2004). Tumor fibrosis, also known as desmoplasia, has been reported to promote tumorigenesis, providing chemoresistance and shielding tumors from therapeutic agents (Olive et al., 2009; Ireland et al., 2016; Laklai et al., 2016; Koay et al., 2014). A mechanism of 'stromal reciprocation', involving mutual signaling between tumor and neighboring cancer-associated fibroblasts (CAFs) that promotes tumor growth, has been demonstrated for PDAC (Tape et al., 2016). On the basis of these findings, several studies have attempted complete ablation of desmoplastic stroma as a therapeutic approach to limit tumor growth, but paradoxically, this resulted in the evolution of existing tumors to a more aggressive state, and accelerated rates of tumorigenesis (Özdemir et al., 2014; Rhim et al., 2014). By contrast, the idea of chronically 'normalizing' activated stroma by reprogramming desmoplasia from a tumor-promoting to a tumor-restrictive state has been suggested to bear greater therapeutic promise (Bijlsma and van Laarhoven, 2015; Sherman et al., 2014; Klemm and Joyce, 2015; Stromnes et al., 2014; Froeling et al., 2011), and

the identification of a clinically applicable means to revert desmoplastic stroma is of considerable interest (*Stromnes et al., 2014*; *Froeling et al., 2011*; *Alexander and Cukierman, 2016*).

Activation of local fibroblastic pancreatic stellate cells and recruited naïve fibroblastic cells during desmoplasia involves their transforming growth factor-beta (TGF$\beta$)-dependent conversion to a myofibroblastic (activated) phenotype (*Desmoulière et al., 1993*; *Meng et al., 2016*; *Principe et al., 2016*; *Xu et al., 2016*). This phenotype is characterized by the induction of stress fibers and the elevated expression of alpha-smooth muscle actin ($\alpha$SMA), palladin and other actin-binding proteins, and by the production of an aligned and organized (anisotropic) ECM, with parallel fibers that are rich in discrete fibronectin splice variants (e.g., ED-A) and in type I collagen (*Mishra et al., 2007*; *Serini et al., 1998*; *Hinz, 2016*; *Rönty et al., 2006*; *Goetz et al., 2011*). Despite the strong effect of TGF$\beta$ on fibroblast activation and ECM remodeling during epithelial cancer-associated desmoplasia, knowledge of the mechanisms and downstream consequences of this activation remain limited (*Hesler et al., 2016*; *Oshima et al., 2015*). In previous studies, we demonstrated that culturing naïve fibroblasts within CAF-secreted D-ECM is sufficient to induce myofibroblastic conversion (*Amatangelo et al., 2005*). Compatible with this dynamic ECM-dependent reprogramming, specific cell-matrix receptors, such as integrins $\alpha_v\beta_5$ and $\alpha_5\beta_1$, have been identified as regulators of myofibroblastic $\alpha$SMA (*Asano et al., 2006*; *Lygoe et al., 2004*) and as participants in the maturation of specific types of cell-matrix adhesions that support anisotropic fiber formation (*Dugina et al., 2001*).

In the present work, we first asked whether D-ECM production and the ability of D-ECM to induce myofibroblastic activation can be decoupled and independently regulated. Using a patient-derived human pancreatic model (*Lee et al., 2011*) as main focus, with supporting data from human renal cancer (*Gupta et al., 2011*) and additional stromal (*Amatangelo et al., 2005*) models, we found that although TGF$\beta$ is needed for production of D-ECM, it is dispensable for subsequent D-ECM-induction of myofibroblastic activation of naïve fibroblasts. We also found that D-ECM controls $\alpha_v\beta_5$-integrin signaling, which prompts the accumulation of activated, but FAK-independent, $\alpha_5\beta_1$-integrin pools in specific intracellular vesicles. This D-ECM control of $\alpha_v\beta_5$-integrin signaling prevents the enrichment of active $\alpha_5\beta_1$-integrin at the plasma membrane (PM), where $\alpha_5\beta_1$ activity opposes myofibroblastic activation. Using a novel integrative approach combining multi-colored immunofluorescence and a new quantitative algorithm, we first validated our *in vitro* findings and then applied this process to annotated clinical samples. This defined two readily distinguishable desmoplastic phenotypes that were correlated with markedly distinct clinical outcomes. These phenotypes are based on differences in the stromal localization and levels of either activated SMAD2/3 (indicative of TGF$\beta$ signaling) or active $\alpha_5\beta_1$-integrin and FAK. These signatures help clarify the controversial role of desmoplasia in the progression of cancer. Further, insofar as reversion of D-ECM has been suggested have the potential to confer significant clinical benefit (*Stromnes et al., 2014*; *Whatcott et al., 2015*; *Neuzillet et al., 2015*), these data suggest potential therapies to stabilize patient-protective or to revert patient-detrimental stroma.

## Results

### TGF$\beta$ is necessary for CAF production of functional anisotropic D-ECM

Fibroblasts were isolated from seven PDAC surgical specimens obtained from five different individuals (with four specimens reflecting two matched tumor-normal pairs, one tumor specimen lacking a matched normal control, and two specimens pathologically designated as non-tumor/normal). These fibroblasts were characterized as naïve pancreatic stellate cells or PDAC-associated desmoplastic CAFs on the basis of assessments of the mRNA and protein expression of the myofibroblastic markers palladin and $\alpha$SMA (*Figure 1A–B*). All specimens were used in parallel for subsequent analyses. In primary culture, these fibroblasts produced characteristic ECM (*Franco-Barraza et al., 2016*). Desmoplastic CAFs produced anisotropic D-ECM with multi-layered myofibroblastic spindled nuclei and increased levels of stress fiber-localized $\alpha$SMA reminiscent of myofibroblastic cells *in vivo* (*Goetz et al., 2011*; *Provenzano et al., 2006*; *Conklin et al., 2011*; *Eyden, 2001*; *Kalluri and Zeisberg, 2006*), whereas fibroblasts derived from normal specimens did not (*Figure 1C*). Quantification of ECM fiber alignment provided a robust measure of tumor-dependent fibroblast activation. We used an arbitrary quantitative threshold of at least 55% of fibers oriented at a spread of 15° from the

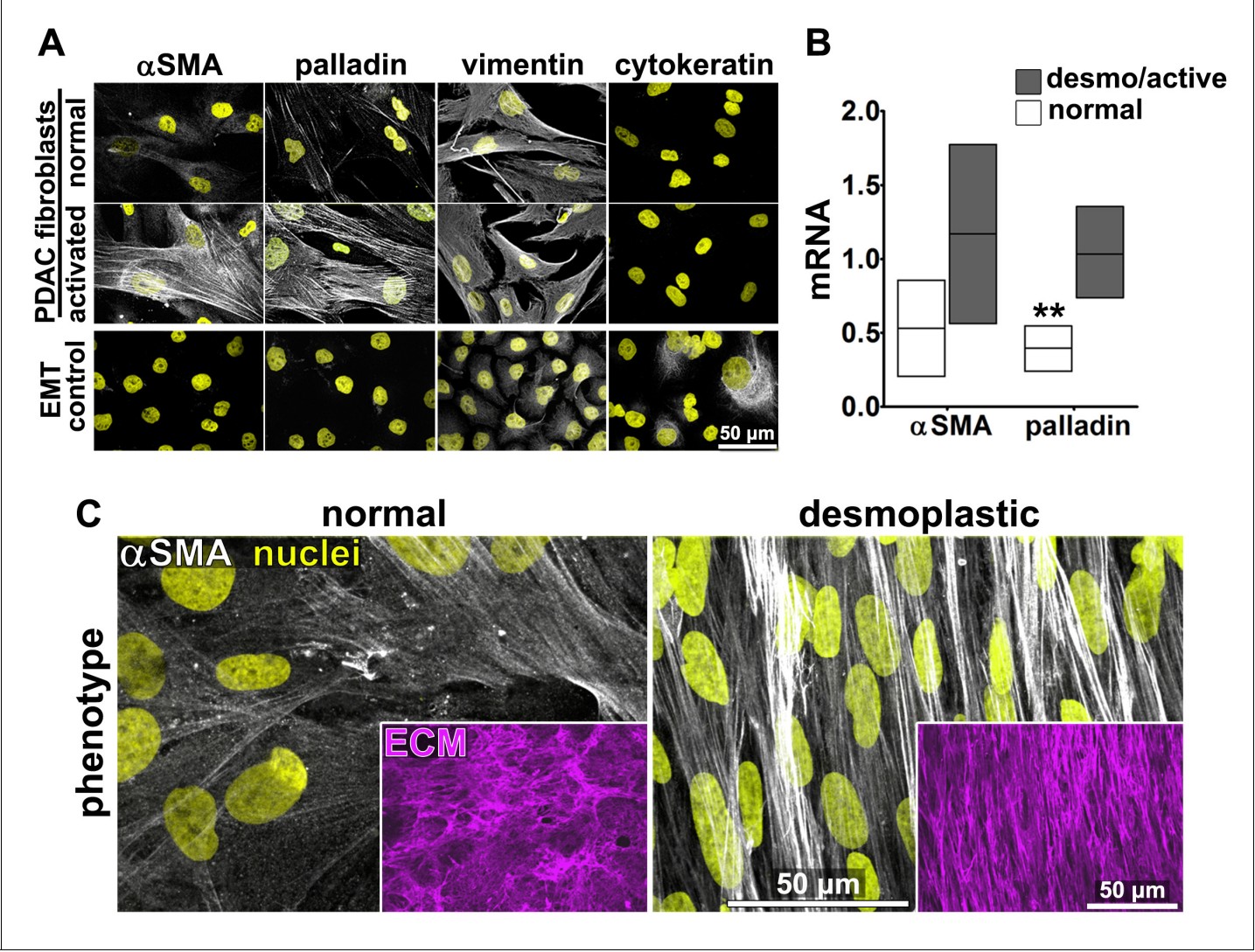

**Figure 1.** Characterization of human fibroblastic cells isolated from PDAC surgical samples. Fibroblastic cells were isolated from normal or tumoral surgical samples from PDAC patients. (**A**) Representative indirect immunofluorescent assessments of vimentin-positive and pan-cytokeratin-negative fibroblasts, isolated from PDAC surgical specimens. Harvested cells were probed for desmoplastic markers αSMA and palladin, while the pancreatic cancer cell line, Panc1, was used as an epithelial-to-mesenchymal transduced (EMT) control that is known to express both epithelial and mesenchymal markers. Assorted markers are shown in white while counterstained Hoechst-identified nuclei are shown in yellow. (**B**) The bar chart shows normal vs. desmoplastic mRNAs levels, corresponding to αSMA and palladin obtained by RT-qPCR from the indicated 3D-cultures following ECM production (obtained by confluent culturing of fibroblasts in the presence of ascorbic acid for a period lasting 8 days [*Franco-Barraza et al., 2016*]) (**p=0.0286). (**C**) Representative images of normal vs. desmoplastic phenotypes after 3D ECM production; comparison of low vs. high αSMA levels (white), heterogeneous/round vs. elongated/spindled nuclei (yellow) and disorganized/isotropic vs. parallel aligned/anisotropic ECMs (magenta) are evident in the representative images. Note that the examples shown corresponds to the matching pair of (naïve vs. desmoplastic) fibroblastic cells that were harvested from surgical samples corresponding to patient #1 and that this pair of cells was used for all examples provided in figures below, unless otherwise stated.

The following figure supplements are available for figure 1:

**Figure supplement 1.** CAFs produce anisotropic D-ECMs.

**Figure supplement 2.** TGFβ inhibition disrupts anisotropy of D-ECM devoid of preventing CAF matrix fibrillogenesis.

**Figure supplement 3.** TGFβ inhibition disrupts N-ECM production by naïve fibroblastic stellate cells.

mode angle as indicative of D-ECMs that had been produced by activated CAFs (*Figure 1—figure supplement 1*).

To first test if autocrine TGFβ signaling is essential for CAF production of TGFβ that then contributes to the formation of anisotropic D-ECM, we used an ELISA-based approach to measure levels of TGFβ present in D-ECM in comparison to levels found in normal ECM (N-ECM), or in ECM produced by CAFs treated with SB-431542, a small-molecule inhibitor of the TGFβ1 receptor. Levels of TGFβ in D-ECM were ~2 fold higher than in N-ECM or D-ECM produced in the presence of SB-431542 (D +TGFβi in *Figure 1—figure supplement 2*). Importantly, following SB-431542 treatment, CAFs produced isotropic ECMs that were phenotypically indistinguishable from intact N-ECMs (*Figure 1—figure supplement 2B–D*). This result differed from that produced by the treatment of control naïve pancreatic stellate cells with SB-431542, which produced interrupted TGFβ signaling that caused complete loss of fibrillogenesis, resulting in the absence of matrix production (*Figure 1—figure supplement 3*). Together, these results suggested that the increased TGFβ observed in D-ECM was critical for CAF production of anisotropic matrices.

Stripping matrix-producing cells from their secreted ECMs produces a residual 'extracted' matrix into which new fibroblasts (e.g., naïve fibroblastic stellate cells) or cancer cells can be seeded (*Franco-Barraza et al., 2016*). Using primary human naïve fibroblastic cells, we have previously shown that the residual three-dimensional (3D) D-ECM produced by CAFs is sufficient to induce a myofibroblastic phenotype (*Amatangelo et al., 2005*). Applying this analysis to pancreatic extracted matrices, we found that D-ECM produced by CAFs in the presence of TGFβ blockade was similar to N-ECM in being incapable of inducing *de novo* myofibroblastic activation (as reflected by increased αSMA stress fiber localization and protein levels) in naïve fibroblastic stellate cells. By contrast, untreated and control treated D-ECM effectively induced such activation (*Figure 2*, *Table 1*). Similar results were obtained using D-ECM and naïve fibroblastic stellate cells for all five PDAC patients (*Figure 2—figure supplement 1*, and *Table 2*), suggesting that myofibroblastic activation is a general phenomenon during interactions between naïve pancreatic stellate cells and CAF-produced PDAC-associated D-ECMs. A post-translational effect was implied by the fact that αSMA mRNA levels were comparable in naïve fibroblastic stellate cells cultured in N- vs D-ECM (*Figure 2—figure supplement 2A*) and because use of cycloheximide to inhibit protein translation did not alter αSMA expression levels (*Figure 2—figure supplement 2B*), whereas αSMA localization differed (*Figure 2*). We also asked whether autocrine TGFβ signaling within naïve stellate cells is necessary for their myofibroblastic activation by D-ECM (*Figure 2—figure supplement 3* and *Table 3*). Growth of these cells in the presence of SB-431542 did not block D-ECM-induced αSMA expression or localization to stress fibers.

Together, these results suggested that TGFβ inhibition during D-ECM production reduces the ability of CAFs to produce ECM that can induce myofibroblastic activation, but that once D-ECM has been deposited by CAFs, TGFβ retained within the D-ECM is subsequently dispensable for D-ECM-induced myofibroblastic activation of naïve stellate cells.

## D-ECM controls $\alpha_v\beta_5$ signaling to regulate active $\alpha_5\beta_1$-integrin during naïve-to-myofibroblastic activation

Integrins $\alpha_v\beta_5$ and $\alpha_5\beta_1$ have been reported to participate in myofibroblastic activation (*Asano et al., 2006*; *Lygoe et al., 2004*; *Dugina et al., 2001*). We first established that both of these integrin heterodimers were highly abundant, compared to other β-integrins, on the PMs of human naïve pancreatic stellate cells and of PDAC-associated CAFs (*Figure 3A*). To determine whether either or both of these heterodimers is essential for D-ECM induction of myofibroblastic conversion, naïve fibroblastic stellate cells were plated overnight on N-ECM or D-ECM in the presence of integrin-inhibitory or negative control antibodies. ALULA, a highly specific $\alpha_v\beta_5$-integrin-blocking antibody (*Su et al., 2007*), eliminated the ability of D-ECM to induce αSMA stress fiber localization (and expression) beyond levels induced by N-ECM (*Figure 3B–E* and *Table 4*). By contrast, mAb16, which specifically blocks human $\alpha_5\beta_1$-integrin activity (*Akiyama et al., 1989*), had limited effects on D-ECM induction of αSMA localization (or expression), with naïve cells undergoing robust myofibroblastic conversion (*Figure 3B–E* and *Table 4*). Intriguingly, combined application of ALULA and mAb16 eliminated the effects of $\alpha_v\beta_5$-integrin inhibition seen with ALULA alone, with naïve cells undergoing robust myofibroblastic transition similar to that in untreated and IgG controls

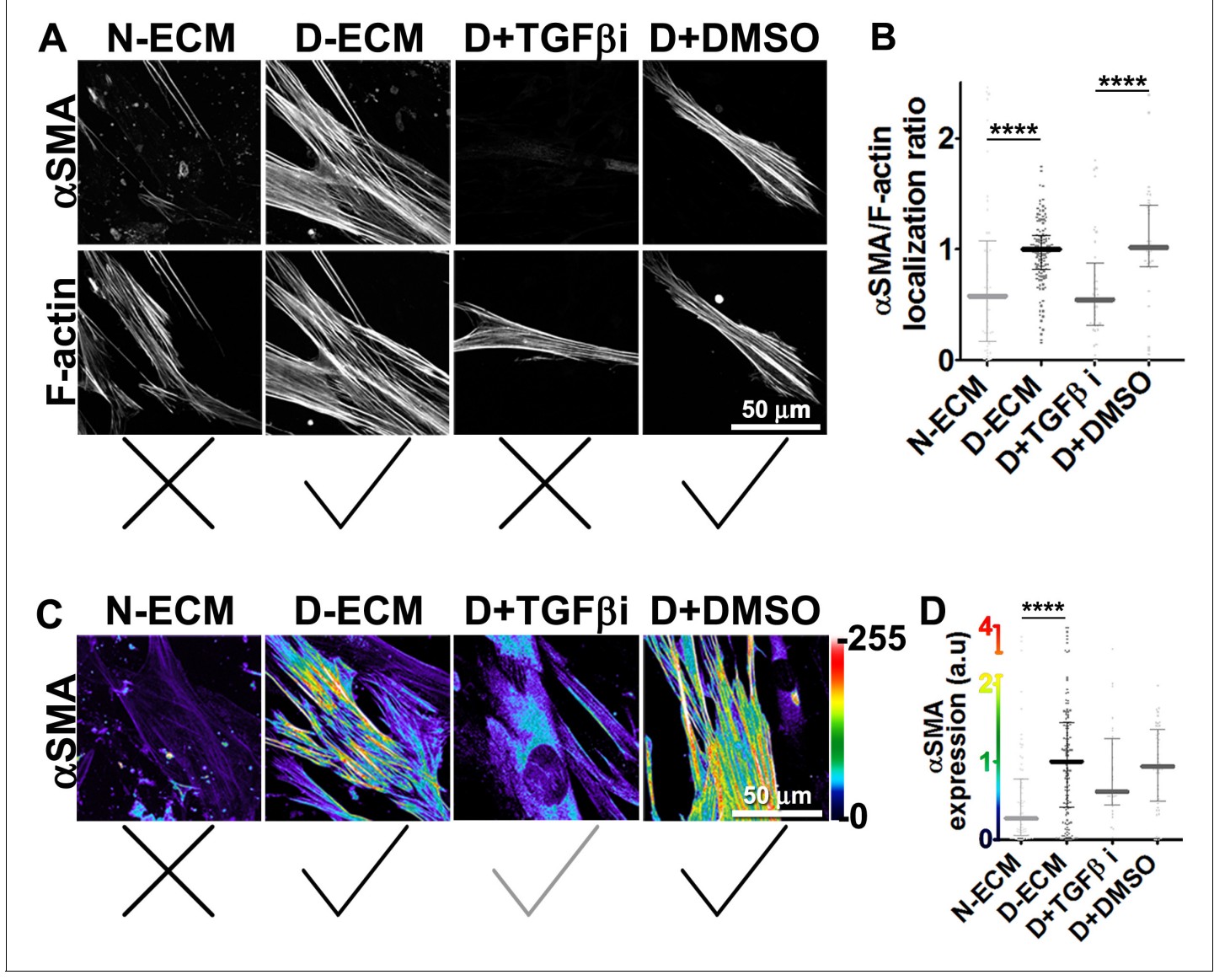

**Figure 2.** TGFβ is necessary for functional CAF-produced D-ECM. Naïve PDAC fibroblasts were cultured overnight within normal (N-ECM) vs. desmoplastic (D-ECM) ECMs that were produced in the presence or absence of TGFβ1-receptor inhibitor (D + TGFβi) or vehicle control (D + DMSO). All samples were subjected to indirect immunofluorescent labeling of αSMA and counterstained with fluorescently labeled phalloidin to detect actin stress fibers (F-actin). (A) Monochromatic images indicating double-labeled staining for αSMA and F-actin. (B) Quantification of the levels of localization of αSMA to actin stress fibers (F-actin) from (A). (C) Pseudo-colored images representing intensity-maps of αSMA levels, with an intensity color bar scale (0–255 intensity tone values) shown to the right. (D) Quantification of αSMA intensity from (C). Untreated D-ECM conditions were included in all experiments summarized in this figure and served as normalization controls (one arbitrary unit; a.u.). Checkmarks indicate conditions that induce myofibroblastic activation phenotypes. X marks indicate conditions that did not induce myofibroblastic activation. All quantifications and p-values can be found in **Table 1**.

The following figure supplements are available for figure 2:

**Figure supplement 1.** Primary CAF-produced D-ECMs induce naïve-to-myofibroblastic activation in normal pancreatic stellate cells.

**Figure supplement 2.** D-ECM imparts αSMA post-translational effects during naïve-to-myofibroblastic activation.

**Figure supplement 3.** TGFβ activity is dispensable for D-ECM-induced naïve-to-myofibroblastic activation.

**Table 1.** αSMA stress fiber localization and expression levels in naïve fibroblasts (stellate cells) cultured overnight within assorted ECMs.

| αSMA | | | N-ECM | D-ECM | D+TGFβi | D+DMSO |
|---|---|---|---|---|---|---|
| αSMA | Stress fiber localization | 25% percentile | 0.17 | 0.82 | 0.32 | 0.85 |
| | | Median | 0.58 | 1.00 | 0.55 | 1.02 |
| | | 75% percentile | 1.07 | 1.12 | 0.87 | 1.40 |
| | expression | 25% percentile | 0.05 | 0.41 | 0.52 | 0.48 |
| | | Median | 0.27 | 1.00 | 0.73 | 0.89 |
| | | 75% percentile | 0.78 | 1.50 | 1.14 | 1.19 |

Values obtained from naïve cells (pancreatic stellate cells isolated from patient #1) cultured overnight within intact D-ECMs (made from CAFs isolated from patient #1) were used for normalization and assigned an arbitrary unit of 1.00. Assorted, patient #1 derived, ECMs were intact N-ECM or intact D-ECM while experimental conditions included D-ECMs made by CAFs treated with SB-431542 (**D+TGFβi**) or DMSO (**D+DMSO**) during ECM production. Note that quantitative immunofluorescent obtained values of αSMA and F-actin were used to calculate stress fiber localization and expression of **αSMA**. P values, listed below, were calculated using the two-sided and two-tailed Mann Whitney test needed for normalized data.

N-ECM vs. D-ECM; **p<0.0001** stress fiber localization;  **p<0.0001** expression

N-ECM vs. D+TGFβi; p=0.8847 stress fiber localization; **p=0.0002** expression

N-ECM vs. D+DMSO; **p=0.0036** stress fiber localization; **p<0.0001** expression

D-ECM vs. D+TGFβi; **p<0.0001** stress fiber localization; **p=0.0198** expression

D-ECM vs. D+DMSO; p=0.3578 stress fiber localization; p=0.1208 expression

D+TGFβi vs. D+DMSO; **p<0.0001** stress fiber localization; p=0.6236 expression

(*Figure 3B–E* and *Table 4*). This raised the possibility that inhibition of $\alpha_5\beta_1$ might act downstream of inhibition of $\alpha_v\beta_5$ in receiving D-ECM signals.

To test whether these signaling interactions were specific to human pancreatic cells or also observed in other systems, we used murine skin squamous cell carcinoma associated fibroblasts, which are known to produce myofibroblastic-activating D-ECMs (mD-ECM) (*Amatangelo et al., 2005*). We cultured naïve murine skin fibroblasts within mD-ECM overnight in the presence of ALULA alone, to inhibit $\alpha_v\beta_5$-integrin, or in combination with BMA5, to inhibit the activity of murine $\alpha_5\beta_1$-integrin specifically (*Fehlner-Gardiner et al., 1996*). The results of this experiment (*Figure 3F*) paralleled those seen with pancreatic stellate cells, suggesting a general mechanism.

To explore the observation that $\alpha_v\beta_5$-integrin-induced myofibroblastic activation is dispensable upon loss of $\alpha_5\beta_1$-integrin activity, we directly tested the possibility that $\alpha_v\beta_5$-integrin negatively regulates $\alpha_5\beta_1$-integrin to prevent active $\alpha_5\beta_1$ from blocking D-ECM-induced activation (see model in *Figure 3G*). For this, we asked whether stabilizing the activity of $\alpha_5\beta_1$-integrin could phenocopy $\alpha_v\beta_5$-integrin inhibition. We found that SNAKA51, an activating antibody that stabilizes the active conformation of $\alpha_5\beta_1$-integrin (*Clark et al., 2005*), reduced the ability of D-ECM to induce αSMA stress fiber localization (and expression) in naïve stellate cells (*Figure 3* and *Table 4*). As an independent approach to simulate ALULA-dependent inhibition of $\alpha_v\beta_5$-integrin, we used Clustered Regularly Interspaced Short Palindromic Repeats (CRISPR)-editing to generate a series of naïve stellate cells in which the $\beta_5$, $\alpha_5$, $\alpha_v$ or $\beta_3$ (a negative control) integrin subunits were ablated (*Figure 3—figure supplements 1* and *2*). Cells lacking the $\beta_5$ subunit had a greatly diminished response to D-ECM, with very limited induction of αSMA stress fiber localization (and expression) (*Figure 3—figure supplement 1*). By contrast, loss of $\alpha_5$-, $\alpha_v$- or $\beta_3$-integrin profoundly reduced growth of naïve fibroblasts, yielding slow-growing cells that were also unable to respond to D-ECM. The slow growth and poor condition of the additional KO cells suggests pleiotropic roles for these integrin subunits in supporting cell viability, and thus makes it difficult to draw conclusions. Nonetheless, the results do not rule out a role for $\alpha_5\beta_1$-integrin activity, regulated by D-ECM control of $\alpha_v\beta_5$-integrin, that opposes the stimulation of the naïve-to-myofibroblastic transition (*Figure 3G*).

**Table 2.** αSMA stress fiber localization and expression levels in assorted naïve pancreatic fibroblasts (stellate cells) cultured overnight within different D-ECMs.

| (#X naïve cell) / (#Y D-ECM) | αSMA | | | | | |
| | Stress fiber localization | | | Expression | | |
| | 25% percentile | Median | 75% percentile | 25% percentile | Median | 75% percentile |
|---|---|---|---|---|---|---|
| (2)/(1) | 0.35 | **1.00** | 1.26 | 0.65 | **1.00** | 1.19 |
| (2)/(2) | 0.90 | **1.15** | 1.38 | 0.87 | **1.10** | 1.30 |
| (3)/(2) | 0.92 | **1.08** | 1.36 | 0.86 | **1.20** | 1.72 |
| (4)/(1) | 0.08 | **0.85** | 1.03 | 0.79 | **2.62** | 3.04 |
| (4)/(5) | 0.06 | **0.83** | 1.36 | 0.29 | **1.15** | 2.40 |
| (2)/(5) | 0.52 | **1.24** | 1.33 | 0.60 | **1.44** | 2.31 |
| (2)/(1 +TGFβi) | 0.00 | **0.00** | 0.04 | 0.04 | **0.07** | 0.10 |
| (2)/(5 +TGFβi) | 0.00 | **0.06** | 1.20 | 0.01 | **0.14** | 0.76 |
| (3)/(1 +TGFβi) | 0.01 | **0.06** | 0.23 | 0.01 | **0.15** | 0.28 |
| (4)/(5 +TGFβi) | 0.11 | **0.22** | 0.71 | 0.25 | **0.74** | 1.66 |

Values obtained from **naïve** cells (e.g., inactive stellate cells) isolated from patient number '#2' cultured overnight within **D-ECMs** made from CAFs isolated from patient #1 were used for normalization and assigned an arbitrary unit of 1.00. Assorted, naïve cells (patient numbers indicated) were cultured within **D-ECMs** derived from the indicated CAFs, while experimental conditions included assorted D-ECMs treated with SB-431542 (**D+TGFβi**) during ECM production. Note that quantitative immunofluorescent obtained values of αSMA and F-actin were used to calculate stress fiber localization and expression of **αSMA**. P values, listed below, were all calculated using the Mann Whitney test, compared to the normalized (2)/(1) experimental condition, to account for non-paired, two-tailed and non-Gaussian distributions of the data.

(2)/(1) vs. (2)/(2); **p=0.0836** stress fiber localization; **p=0.3825** expression
(2)/(1) vs. (3)/(2); **p=0.1680** stress fiber localization; **p=0.1736** expression
(2)/(1) vs. (4)/(1); **p=0.4266** stress fiber localization; **p=0.0755** expression
(2)/(1) vs. (4)/(5); **p=0.8927** stress fiber localization; **p=0.7509** expression
(2)/(1) vs. (2)/(5); p=0.1885 stress fiber localization; p=0.4192 expression
(2)/(1) vs. (2)/(1+TGFβi); **p<0.0001** stress fiber localization; p<0.0001 expression
(2)/(1) vs. (2)/(5+TGFβi); **p=0.0909** stress fiber localization; p=0.0040 expression
(2)/(1) vs. (3)/(1+TGFβi); **p=0.0018** stress fiber localization; **p<0.0001** expression
(2)/(1) vs. (4)/(5+TGFβi); **p=0.1018** stress fiber localization; p=0.5181 expression

Next, we asked whether manipulation of integrin activities in CAFs, as opposed to in naïve stellate cells, could influence their ability to produce functional D-ECM. In contrast to results with TGFβ blockade, neither $\alpha_v\beta_5$-integrin inhibition (with ALULA) nor stabilization of active $\alpha_5\beta_1$-integrin (with SNAKA51) altered the anisotropic fiber formation in D-ECM deposited by CAFs (***Figure 4***), or the ability of these matrices to induce myofibroblastic activation in naïve fibroblastic stellate cells (***Figure 4—figure supplement 1***). We also analyzed the ECM produced by CRISPR-edited CAFs lacking specific integrin subunits. In contrast to results with integrin inhibition, CAFs lacking $\beta_5$-integrin or $\alpha_5$-integrin subunits had decreased αSMA expression and localization of αSMA to stress fibers. Loss of $\beta_5$ expression also affected levels of D-ECM anisotropy, while CAFs lacking $\alpha_5$ failed to produce substantial matrices (matching earlier reports [***Pankov et al., 2000***; ***McDonald et al., 1987***; ***Fogerty et al., 1990***]) (***Figure 4—figure supplement 2***). In addition, loss of $\alpha_v$-integrin caused significant reduction in the ability to produce D-ECM, while loss of the negative control, $\beta_3$, had no effect on this phenotype (***Figure 4—figure supplement 3***). These results suggested that the ability of CAFs to grow and produce functional D-ECMs was selectively affected by loss of $\alpha_5$, $\alpha_v$ and $\beta_5$ integrin subunits but not by $\beta_3$ loss.

Together, these data indicated that the requirement for specific integrins in the response of naïve fibroblastic cells to D-ECMs differs from the requirement for integrins during CAF production of anisotropic D-ECMs.

**Table 3.** αSMA stress fiber localization and expression levels in naïve fibroblasts (stellate cells) cultured overnight in the presence or absence of TGFβ inhibitor within intact D-ECMs.

| | | | TGFβ-i | DMSO |
|---|---|---|---|---|
| αSMA | Stress fiber localization | 25% percentile | 0.71 | 0.72 |
| | | Median | 0.96 | 1.00 |
| | | 75% percentile | 1.29 | 1.04 |
| | expression | 25% percentile | 0.44 | 0.49 |
| | | Median | 0.62 | 0.94 |
| | | 75% percentile | 1.30 | 1.41 |

Values obtained from naïve cells cultured overnight within intact D-ECMs were used for normalization (as shown in **Table 1**) and assigned an arbitrary unit of 1.00. **TGFβ-i** is the experimental condition in which naïve pancreatic stellate cells were cultured overnight in the presence of SB-431542 within intact D-ECM. **DMSO** treatment corresponds to vehicle control. Note that quantitative immunofluorescent obtained values of αSMA and F-actin were used to calculate stress fiber localization and expression of **αSMA**.

P values, listed below, were calculated using the two-sided and two-tailed Mann Whitney test needed for normalized data.

TGFβi vs. DMSO; p=0.3508 stress fiber localization; p=0.3361 expression

TGFβi vs. N-ECM (from **Table 1**); **p=0.0110** stress fiber localization; **p=0.0010** expression

TGFβi vs. D-ECM (from **Table 1**); p=0.9132 stress fiber localization; p=0.3401 expression

DMSO vs. N-ECM (from **Table 1**); **p=0.0036** stress fiber localization; **p<0.0001** expression

DMSO vs. D-ECM (from **Table 1**); p=0.2635 stress fiber localization; p=0.7408 expression

## FAK-independent $\alpha_5\beta_1$-integrin activity blocks D-ECM-induced naïve-to-myofibroblastic activation

Although integrins often signal through focal adhesion kinase (FAK), an increasing number of FAK-independent integrin signaling activities have been observed (*Cukierman et al., 2001*; *Wu et al., 2008*; *Zoppi et al., 2008*; *Horton et al., 2016*). We investigated the role of FAK in D-ECM-induced myofibroblastic activation, treating naïve fibroblastic stellate cells plated in D-ECM with the small molecule FAK inhibitor PF573,228 (*Slack-Davis et al., 2007*). FAK inhibition strongly reduced the ability of these cells to acquire myofibroblastic traits, similar to the result seen with $\alpha_v\beta_5$-integrin inhibition (*Figure 5*). Further, naïve pancreatic stellate cells treated concomitantly with PF573,228 and the $\alpha_5\beta_1$-integrin-inhibiting mAb16 (*Akiyama et al., 1989*) underwent a myofibroblastic conversion, re-localizing and upregulating αSMA (*Figure 5*).

Emphasizing a general mechanism, treatment of naïve murine fibroblasts plated in mD-ECM with PF573,228 blocked acquisition of myofibroblastic traits (*Figure 5—figure supplement 1*). In naïve murine fibroblasts, concomitant FAK and $\alpha_5\beta_1$-integrin inhibition with combined PF573,228 and BMA5 significantly increased mD-ECM-induced myofibroblastic conversion when compared with PF573,228 treatment alone, while $\alpha_v\beta_5$-integrin co-inhibition with FAK did not (*Figure 5—figure supplement 1*). Further, FAK$^{-/-}$ murine naïve fibroblastic cells (*Ilić et al., 1995*) grown in mD-ECM had greatly reduced myofibroblastic activation, which was significantly increased if $\alpha_5\beta_1$ but not $\alpha_v\beta_5$ integrin activation was blocked (*Figure 5—figure supplement 2*). As an additional control to exclude off-target or indirect effects of drug inhibition or FAK KO background compensation, we asked whether murine fibroblasts engineered to express a dominant negative FAK Kinase-Dead mutant (FAK-KD) (*Lim et al., 2010*) behaved similarly to those with deleted or inhibited FAK, and whether they also recovered responsiveness to D-ECM following inhibition of $\alpha_5\beta_1$-integrin activity. Immortalized murine fibroblasts overexpressing FAK-KD displayed a lack of myofibroblastic response to mD-ECM compared to the response seen in isogenic hTert immortalized control cells, and again, an efficient rescue of this phenotype was imparted by $\alpha_5\beta_1$-integrin inhibition (*Figure 5—figure supplement 3*).

FAK often interacts with SRC family kinases to mediate integrin signaling. To further probe the mechanisms, we evaluated the mD-ECM responsiveness of fibroblasts cells that are genetically null for the SRC family kinases SRC, FYN and YES (*Klinghoffer et al., 1999*). We found that the ablation

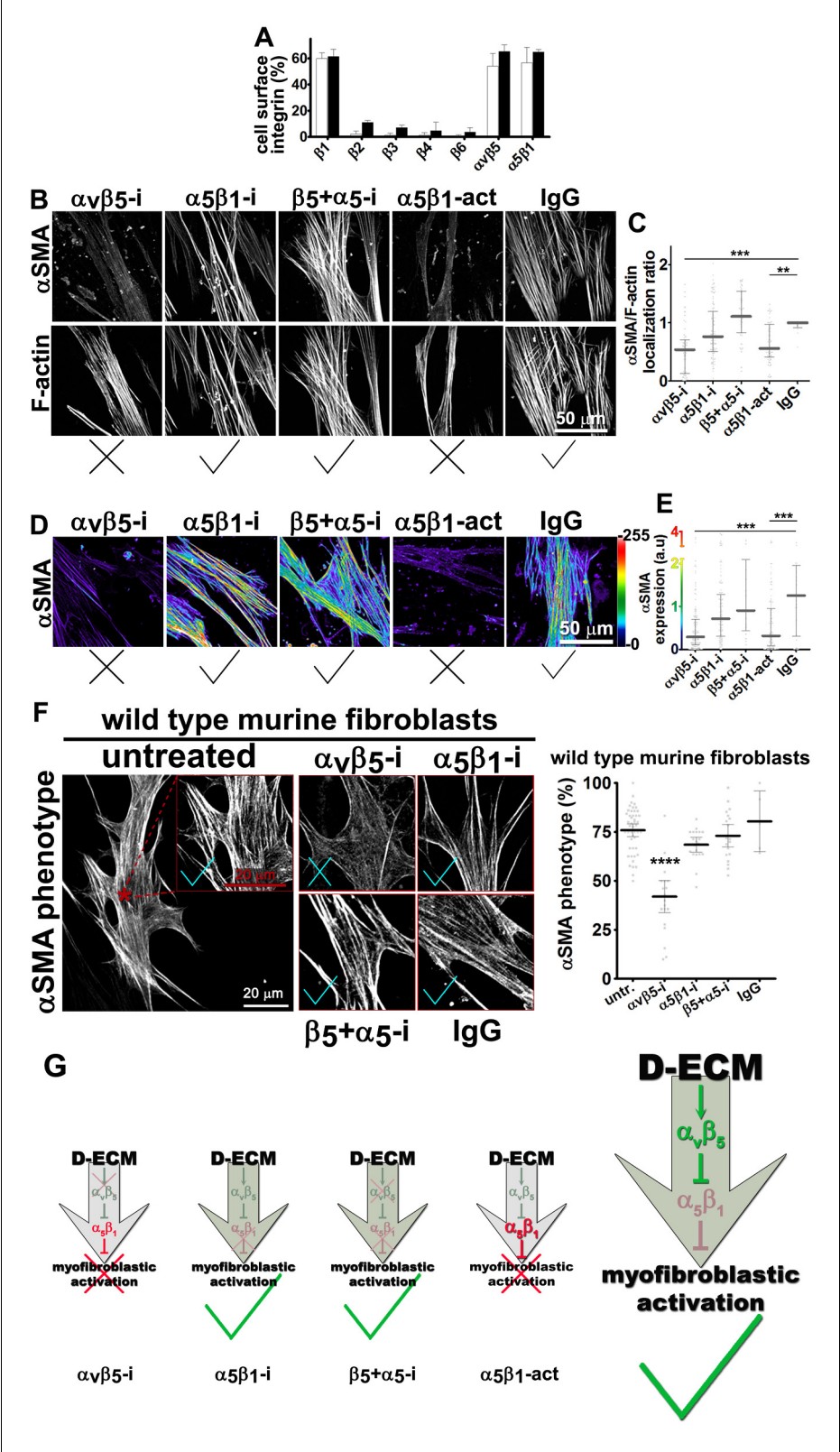

**Figure 3.** Integrin $\alpha_v\beta_5$ regulates $\alpha_5\beta_1$ activity thereby maintaining D-ECM-induced naïve-to-myofibroblastic activation. (**A**) An integrin-dependent cell adhesion array test was used to assess the PM expression of integrin heterodimers in primary fibroblasts isolated from normal (white bars) vs. matched tumor tissue (desmoplastic; dark bars). Note that no differences were apparent between the two cell types with regards to levels of $\alpha_v\beta_5$ and $\alpha_5\beta_1$ integrins. (**B**) Naïve human pancreatic fibroblastic stellate cells were re-plated onto D-ECMs overnight in the presence of functional blocking anti-$\alpha_v\beta_5$-

*Figure 3 continued on next page*

*Figure 3 continued*

integrin (ALULA [**Su et al., 2007**]; $\alpha_v\beta_5$-i), functional blocking anti-$\alpha_5\beta_1$-integrin (mAb16 [**Akiyama et al., 1989**]; $\alpha_5\beta_1$-i), combinations of both functional blocking antibodies ($\beta_5$-i + $\alpha_5$-i), functional stabilizing anti-$\alpha_5\beta_1$-integrin (SNAKA51 [**Clark et al., 2005**]; $\alpha_5\beta_1$-act), or non-immunized isotypic antibodies (IgG). Representative monochromatic images of $\alpha$SMA- and F-actin stained fibroblasts are shown. (C) Quantification of the experiment performed in (B). (D) Pseudocolored images depicting the intensity of $\alpha$SMA expression including a color bar scale (0–255 intensity tone values). (E) Quantification of (D). Note that corresponding quantifications and p-values, for results shown in (B–E) are summarized in *Table 4*. (F) Naïve murine skin fibroblasts were re-plated onto murine D-ECMs (mD-ECM) produced by murine skin squamous cell carcinoma associated CAFs (**Amatangelo et al., 2005**), and subjected to $\alpha_v\beta_5$-integrin and $\alpha_5\beta_1$-integrin inhibitors alone (ALULA: $\alpha_v\beta_5$-i [**Su et al., 2007**] and BMA5: $\alpha_5\beta_1$i) or in combination ($\beta_5$+ $\alpha_5$-i). The effects on myofibroblastic activation were measured for $\alpha$SMA as in (B). The red asterisk illustrates the area outlined in red in the magnified insert for the intact (untreated) control. The same magnification is shown for the experimental conditions in the additional panels. As a method of quantifying the percentage of cells showing myofibroblastic features, the percentage of cells that have a stress fiber localized ($\alpha$SMA) phenotype is shown (****p<0.0001). Note that inhibition of $\alpha_5\beta_1$-integrin effectively reinstituted the mD-ECM-induced phenotype that was lost by inhibition of $\alpha_v\beta_5$-integrin, just as seen above for the human PDAC system. Checkmarks identify conditions that resulted in myofibroblastic activation, while Xs identify conditions that did not result in myofibroblastic activation. (G) Model of D-ECM-induced activation of naïve fibroblasts, dependent on the activity of integrins $\alpha_v\beta_5$ and $\alpha_5\beta_1$. Inhibition of $\alpha_v\beta_5$- integrin results in release of active $\alpha_5\beta_1$-integrin, leading to blockade of D-ECM-induced myofibroblastic activation (1st arrow, red X). The activity of $\alpha_v\beta_5$- integrin is no longer needed in the absence of $\alpha_5\beta_1$-integrin activity, suggesting that $\alpha_5\beta_1$-integrin activity in not necessary for fibroblasts to undergo D-ECM-induced myofibroblastic activation (2nd arrow, green checkmark). Double inhibition of $\alpha_v\beta_5$-integrin and $\alpha_5\beta_1$-integrin results in D-ECM myofibroblastic activation, which proposes that inhibition of $\alpha_5\beta_1$-integrin can overcome or rescue the effects seen under $\alpha_v\beta_5$-integrin inhibition (3rd arrow, green checkmark). Stabilization of $\alpha_5\beta_1$-integrin in its active conformation overcomes the inhibitory/regulatory effects imparted by $\alpha_v\beta_5$-integrin, resulting in ineffective D-ECM-induced myofibroblastic activation (4th arrow, red X). Overall, the model suggests that D-ECM induces $\alpha_v\beta_5$- integrin activity, which in turn results in the regulation of active $\alpha_5\beta_1$-integrin, allowing D-ECM-induced myofibroblastic activation (large arrow to the right, green checkmark).

The following figure supplements are available for figure 3:

**Figure supplement 1.** Naïve $\beta_5$-integrin KO fibroblasts display stunted D-ECM-induced myofibroblastic activation.

**Figure supplement 2.** Expression of $\alpha_5$-, $\alpha_v$- and $\beta_3$-integrins in naïve fibroblastic stellate cells is necessary for D-ECM-induced myofibroblastic activation.

**Table 4.** $\alpha$SMA stress fiber localization and expression levels in naïve fibroblasts (stellate cells) cultured overnight within D-ECMs in the presence of integrin functional antibodies.

|  |  |  | $\alpha_v\beta_5$-i | $\alpha_5\beta_1$-i | $\beta_5$+$\alpha_5$-i | $\alpha_5\beta_1$-act | IgG |
|---|---|---|---|---|---|---|---|
| **αSMA** | **Stress fiber localization** | 25% percentile | 0.14 | 0.51 | 0.83 | 0.41 | 0.92 |
|  |  | Median | 0.54 | 0.77 | 1.11 | 0.56 | 1.00 |
|  |  | 75% percentile | 0.71 | 1.20 | 1.55 | 0.98 | 1.00 |
|  | expression | 25% percentile | 0.11 | 0.31 | 0.44 | 0.08 | 0.31 |
|  |  | Median | 0.29 | 0.71 | 0.90 | 0.32 | 1.25 |
|  |  | 75% percentile | 0.70 | 1.27 | 2.09 | 0.96 | 1.95 |

Values obtained from naïve cells cultured overnight within intact D-ECMs treated with ALULA ($\alpha_v\beta_5$-i), mAb16 ($\alpha_5\beta_1$-i), ALULA plus mAb16 ($\beta_5$+$\alpha_5$-i), SNAKA ($\alpha_5\beta_1$-act) or control pre-immune antibody (IgG). Values obtained in untreated D-ECMs (from *Table 1*) were used for normalization and assigned an arbitrary unit of 1.00. Note that the quantitative values of $\alpha$SMA and F-actin obtained by immunofluorescence were used to calculate stress fiber localization and expression of $\alpha$SMA.

P values, listed below, were calculated using the two-sided and two-tailed Mann Whitney test needed for normalized data.

$\alpha_v\beta_5$-i vs. IgG; **p=0.0001** stress fiber localization; **p=0.0002** expression

$\alpha_5\beta_1$-i vs. IgG; p=0.1311 stress fiber localization; p=0.1229 expression

$\beta_5$+$\alpha_5$-i vs. IgG; **p=0.0333** stress fiber localization; p=0.9171 expression

$\alpha_5\beta_1$-act vs. IgG; **p=0.0047** stress fiber localization; **p=0.0020** expression

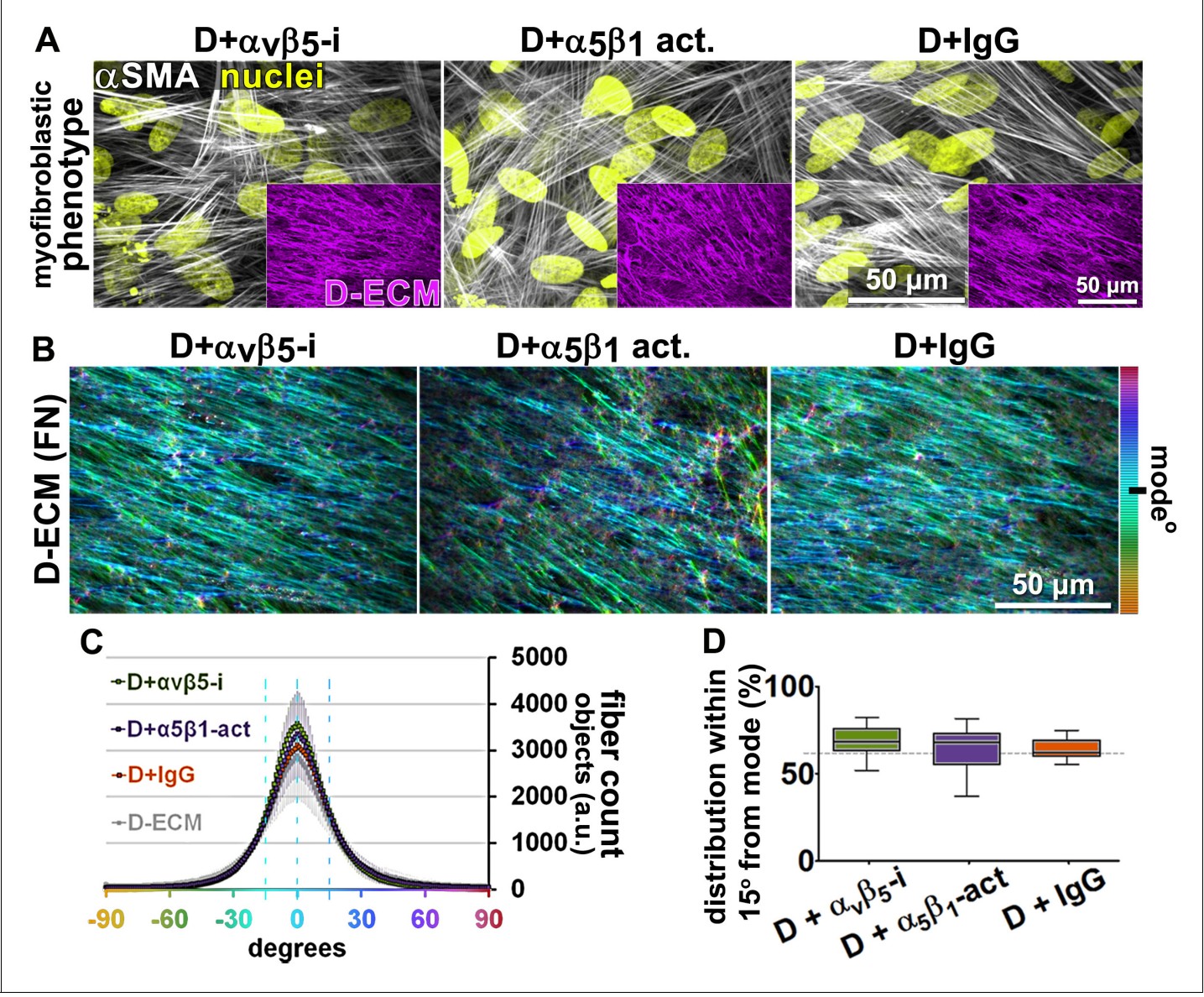

**Figure 4.** Transient $\alpha_v\beta_5$-integrin inhibition or $\alpha_5\beta_1$-integrin stabilization failed to alter CAF production of anisotropic D-ECM. (**A**) Representative indirect immunofluorescent images of 3D D-ECM producing CAFs in the presence of functional blocking anti-$\alpha_v\beta_5$-integrin (ALULA [**Su et al., 2007**]; D + $\alpha_v\beta_5$-i), active conformation stabilizing anti-$\alpha_5\beta_1$-integrin (SNAKA51 [**Clark et al., 2005**]; D + $\alpha_5\beta_1$-act.) or non-immunized isotypic antibodies (D + IgG). Spinning disk confocal monochromatic images, obtained following indirect immunofluorescence, show nuclei (Hoechst; yellow), $\alpha$SMA (white) and ECM (fibronectin; magenta). (**B**) The corresponding ECM fiber angle distributions, determined by Image-J's 'Orientation J' plugin, were normalized using hue values for a cyan mode angle visualization as shown in the bar on the right. (**C**) Corresponding curves depicting experimental-repetition-averaged variations of angle distributions normalized to 0° modes and summarizing the results. Dotted lines correspond to 15° fiber angle spreads. (**D**) Plotted data depicting summarized percentages of fibers distributed at 15° angles from the mode for each experimental condition. The dotted line denotes 55% alignment. Note how none of the treatments seem to have altered the myofibroblastic features of CAFs or their capability to produce anisotropic D-ECMs.

The following figure supplements are available for figure 4:

**Figure supplement 1.** D-ECMs produced by CAFs under transient $\alpha_v\beta_5$-integrin inhibition or stabilization of $\alpha_5\beta_1$-integrin activity are functionally intact.

**Figure supplement 2.** Loss of $\beta_5$-integrin expression in CAFs impairs D-ECM anisotropy, while expression of $\alpha_5$-integrin is imperative for effective ECM fibrillogenesis.

**Figure supplement 3.** KO of $\alpha_V$-integrin, but not $\beta3$-integrin, disrupts CAF myofibroblastic features and D-ECM production.

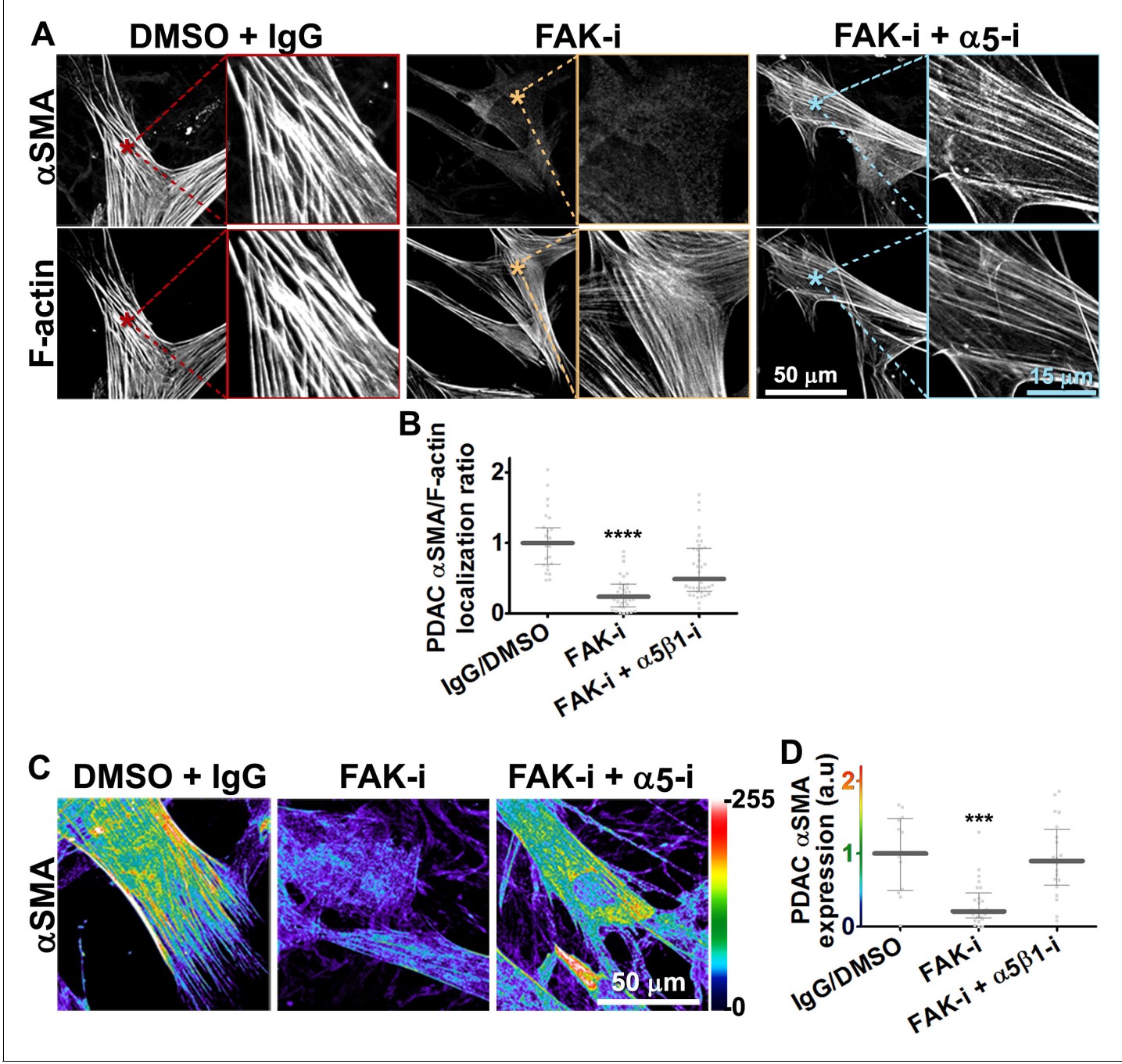

**Figure 5.** FAK-independent $\alpha_5\beta_1$-integrin activity negatively regulates PDAC D-ECM-induced naïve-to-myofibroblastic activation. Naïve fibroblasts were re-plated onto D-ECMs and challenged with either control conditions (DMSO + IgG), small molecule FAK inhibitor PF573,228 (*Slack-Davis et al., 2007*) (FAK-i) alone or FAK inhibitor in combination with $\alpha_5\beta_1$-integrin inhibitor (FAK-i + $\alpha_5$-i, mAb16 [*Akiyama et al., 1989*]), and activation of fibroblasts was tested. (**A**) Representative monochromatic images of immunofluorescently labeled αSMA and actin stress fibers (F-actin). Colored asterisks in (**A**) represent areas that are magnified in the corresponding panels to the right. (**B**) Quantification of αSMA at actin stress fibers (F-actin) from the experiment in (**A**) and normalized to DMSO + IgG control (one arbitrary unit; a.u.) (****p<0.0001). (**C**) Pseudocolored images represent intensity maps of αSMA, with an intensity color bar scale (0–255 intensity tone values) shown to the right. (**D**) Quantification of αSMA intensity from (**C**) (***p=0.0001). Note that the D-ECM-induced phenotype that was lost under FAK inhibition was rescued under $\alpha_5\beta_1$-integrin co-inhibition (just as for the two integrin co-inhibitions shown in *Figure 3*).

The following figure supplements are available for figure 5:

**Figure supplement 1.** FAK-independent $\alpha_5\beta_1$-integrin activity negatively regulates mD-ECM-induced murine naïve-to-myofibroblastic activation.

*Figure 5 continued on next page*

*Figure 5 continued*

**Figure supplement 2.** $\alpha_5\beta_1$-integrin inhibition restores mD-ECM-induced naïve-to-myofibroblastic activation in murine FAK[-/-] skin fibroblasts.

**Figure supplement 3.** $\alpha_5\beta_1$-integrin inhibition restores mD-ECM-induced naïve-to-myofibroblastic in murine FAK-KD skin fibroblasts.

**Figure supplement 4.** Ablation of SRC family kinases negates the ability of naïve fibroblasts to respond to mD-ECM.

**Figure supplement 5.** FAK inhibition in conjunction with loss of $\alpha_5$-integrin expression in human naïve fibroblasts fails to restore D-ECM-induced myofibroblastic activation.

of SRC family kinases also negated the ability of fibroblasts to respond to mD-ECM. Interestingly, in this context, inhibition of $\alpha_5\beta_1$-integrin did not restore the mD-ECM-induced myofibroblastic phenotype, indicating non-equivalent functions of FAK and SRC (*Figure 5—figure supplement 4*). Last, we explored the relationship between FAK and $\alpha_5$-integrin expression (as opposed to integrin activity). The ability of the FAK inhibitor to block D-ECM-induced myofibroblastic transition depended on an intact $\alpha_5\beta_1$ heterodimer, as PF573,228 did not rescue the mD-ECM-induced process that was disrupted in $\alpha_5$-KO naïve stellate cells (*Figure 5—figure supplement 5*).

Together, these results suggested that FAK is essential for D-ECM-induced myofibroblast activation, but dispensable for the $\alpha_5\beta_1$-integrin inhibition of this process. The data further indicate that genetic ablation of $\alpha_5\beta_1$-integrin does not recapitulate the result of inhibition of integrin activity in D-ECM responsiveness, perhaps due to additional functional requirements for this integrin in supporting fundamental cell growth.

## D-ECM regulates length of 3D-adhesion structures concomitant with $\alpha_v\beta_5$-integrin-dependent redistribution of active $\alpha_5\beta_1$-integrin to endosomal vesicles

*In vivo*, fibroblasts engage N-ECM through 3D matrix adhesion structures (3D-adhesions) that mediate matrix-dependent homeostasis (*Cukierman et al., 2001*). 3D-adhesions are elongated adhesion plaques that depend on $\alpha_5\beta_1$-integrin activity for their formation and are characterized by encompassing active $\alpha_5\beta_1$-integrin concomitant with constitutive, albeit low, levels of auto-phosphorylated, activated FAK (pFAK-Y$^{397}$) (*Cukierman et al., 2001*). We tested the idea that D-ECM might alter structure, protein composition, or signaling at 3D-adhesions. Naïve stellate cells cultured within D-ECM typically increased the length of 3D-adhesions by ~14% compared to adhesions formed in N-ECM (*Figure 6*). While inhibition of $\alpha_5\beta_1$-integrin with mAb16 was previously shown to cause 3D-adhesion loss (*Cukierman et al., 2001*), stabilization of $\alpha_5\beta_1$-integrin activity using SNAKA51 eliminated the ability of D-ECM to induce 3D-adhesion lengthening. However, $\alpha_v\beta_5$-integrin inhibition with ALULA did not (*Figure 6*). These results separate the requirements for D-ECM-induced $\alpha_v\beta_5$-integrin regulation of $\alpha$SMA (*Figure 3*) from the lack of requirement for $\alpha_v\beta_5$-integrin for adhesion reorganization in naïve fibroblastic cells.

Next, we developed a semi-quantitative indirect immunofluorescence analytic method (based on SMIA-CUKIE software; see Methods) to evaluate the total intensity of active $\alpha_5\beta_1$-integrin, versus its relative intensity, and area distributions related to 3D-adhesion sites. Plating of naïve stellate cells in D-ECM induced a 4-fold increase in active $\alpha_5\beta_1$-integrin levels compared to plating in N-ECM (*Figure 6—figure supplement 1*), but surprisingly, this did not reflect an increase in $\alpha_5\beta_1$ localized to 3D adhesions (*Figure 6—figure supplement 2A*) but rather a concentration (~2 fold) in membrane-proximal regions areas lacking adhesions. To test the idea that D-ECM might affect the recruitment or activation of FAK in 3D adhesions, potentially through control of $\alpha_5\beta_1$-integrin activation or localization, we also examined the localization of activated FAK (pFAK-Y$^{397}$) to 3D adhesions. Results normalized to levels obtained in D-ECMs (a.u. = 1.0) indicated that D-ECM induced ~2 fold more pFAK-Y$^{397}$ than N-ECM, and that this activated FAK was localized at 3D-adhesions (*Figure 6—figure supplement 2B*). By contrast, if fibroblasts are cultured in the presence of the $\alpha_v\beta_5$-integrin inhibitor ALULA, significantly lower levels of pFAK-Y$^{397}$ at 3D-adhesions and active $\alpha_5\beta_1$-integrin away from 3D-adhesions were observed in cells cultured within D-ECM (*Figure 6—figure supplement 2B–C*). A

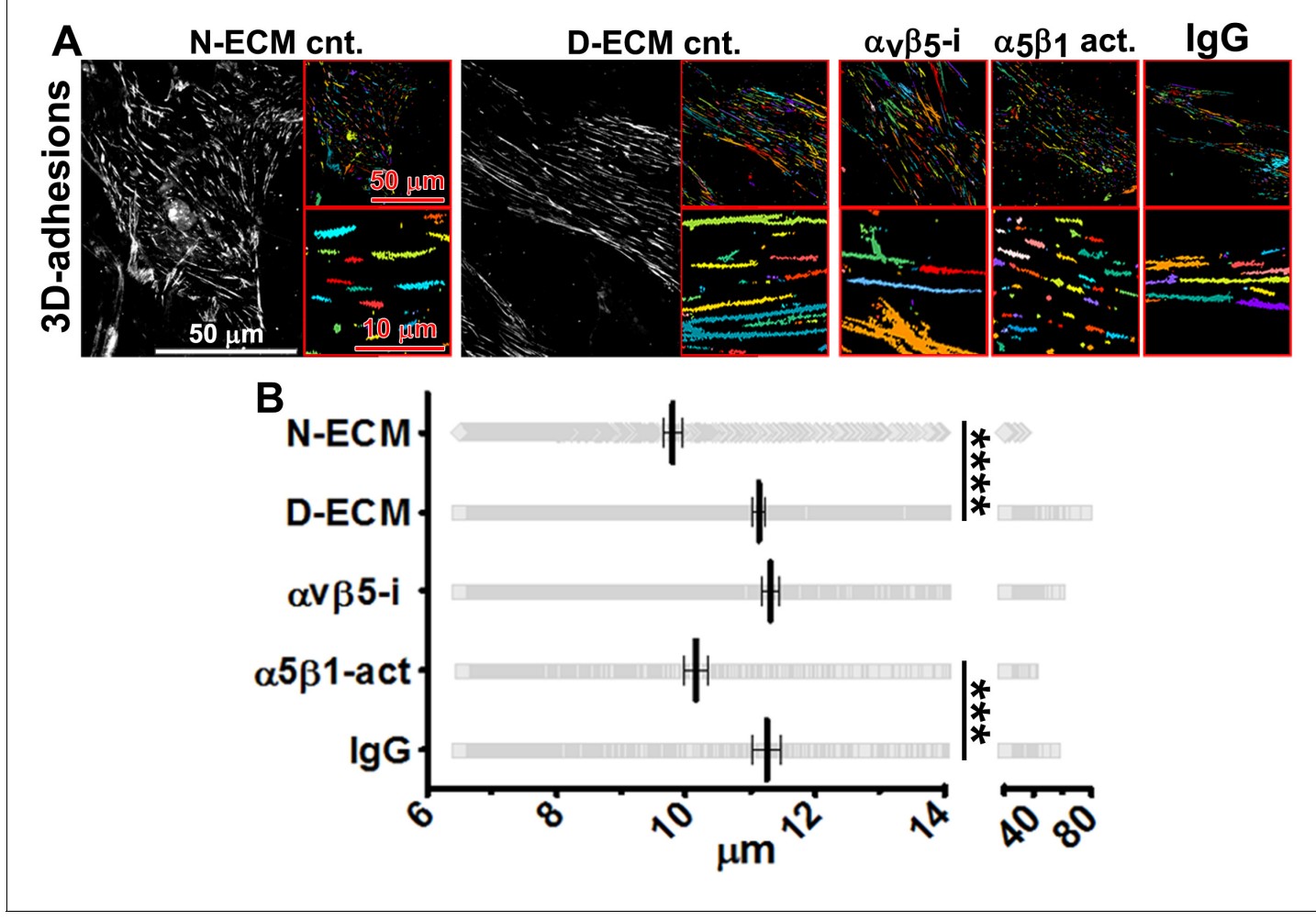

**Figure 6.** D-ECM regulates 3D-adhesion structure length, dependent on $\alpha_5\beta_1$-integrin activity. (**A**) Indirect immunofluorescent and spinning disc confocal generated images of 3D-adhesions (identified using mAb11) (*Cukierman et al., 2001*), formed by naïve fibroblastic cells cultured within N-ECM or D-ECM in the absence (cnt.) or presence of ALULA (*Su et al., 2007*) for $\alpha_v\beta_5$-integrin inhibition ($\alpha_v\beta_5$-i) or SNAKA52 (*Clark et al., 2005*) to stabilize $\alpha_5\beta_1$-integrin activity ($\alpha_5\beta_1$ act) or IgG as control. (A) The artificially colored structures represent computer-selected internally threshold objects (ITOs) of 3D-adhesion structures. (**B**) Quantification of the length of ITO generated objects from (**A**) (***p=0.0026. ****p<0.0001). Note the significant differences in 3D-adhesion length observed between N-ECM and D-ECM as well as between IgG and SNAKA51 treatments.

The following figure supplements are available for figure 6:

**Figure supplement 1.** Naïve fibroblastic cells increase overall levels of active $\alpha_5\beta_1$-integrin in response to D-ECM.

**Figure supplement 2.** D-ECMs induce an increase in pFAK levels at 3D-adhesions and redistribution of increased active $\alpha_5\beta_1$-integrin to locations away from 3D-adhesions.

**Figure supplement 3.** D-ECM-induced increase in active $\alpha_5\beta_1$-integrin is reversed by inhibition of $\alpha_v\beta_5$-integrin.

**Figure supplement 4.** D-ECM induces intracellular accumulation of active $\alpha_5\beta_1$-integrin.

**Figure supplement 5.** D-ECM-induced increase in intracellular $\alpha_5\beta_1$-integrin activity is regulated by $\alpha_v\beta_5$-integrin.

summary of results regarding the levels and locations of active $\alpha_5\beta_1$-integrin is shown in *Figure 6—figure supplement 3*.

To better refine the location of $\alpha_5\beta_1$-integrin activation, and taking advantage of the fact that the epitope recognized by SNAKA51 is typically extracellular with PM-localized integrin (*Clark et al., 2005*), we performed SMIA-CUKIE analysis comparing SNAKA51 levels in permeabilized versus non-permeabilized cells plated on D-ECM versus N-ECM. Non-permeabilized cells had a 5-fold reduction of detectable D-ECM-induced active (SNAKA51-positive) $\alpha_5\beta_1$-integrin relative to permeabilized cells (*Figure 6—figure supplement 4*), indicating that the activated integrin may be rapidly internalized. As a control and to demonstrate the specificity of the active integrin conformation detection, we demonstrated that D-ECM treatment in the presence of the mAb16 inhibitory antibody eliminated all detectable active $\alpha_5\beta_1$-integrin. In addition, inhibition of $\alpha_v\beta_5$-integrin, by treatment with ALULA, effectively eliminated the intracellular pool while it considerably increased the amounts of surface-exposed active $\alpha_5\beta_1$-integrin induced by growth in D-ECM (*Figure 6—figure supplement 5*).

Generally similar results were obtained in analysis of pancreatic human naïve stellate cells deficient in discrete integrin subunits. Naïve $\beta_5$-integrin KO fibroblastic stellate cells plated within D-ECM had 2.7- and 2.5-fold decreases in total $\alpha_5\beta_1$-integrin activity and pFAK-Y$^{397}$ levels, respectively, when compared to control KO cells (*Figure 7*). Experiments were also conducted using naïve $\alpha_5$, $\alpha_v$ and $\beta_3$-integrin KO stellate cells as controls. As expected, no detectable $\alpha_5\beta_1$-integrin activity was observed in $\alpha_5$-KO, while only modest decreases in the activity of $\alpha_5\beta_1$-integrin was observed in $\alpha_v$ and $\beta_3$-integrin KO cells grown in intact D-ECM. Interestingly, while a modest pFAK-Y$^{397}$ downregulation was observed in $\alpha_5$-KO, there were no appreciable changes in pFAK-Y$^{397}$ in $\alpha_v$-KO cells, whereas a small increase in pFAK-Y$^{397}$ was induced by D-ECM in $\beta_3$-integrin KO fibroblastic stellate cells (*Figure 7—figure supplement 1*). Although results with $\alpha_v$- and $\beta_3$-integrin KO did not exclude these integrins from any role in response to D-ECM, these experiments indicated that $\beta_5$-integrin was an effective regulator of both active $\alpha_5\beta_1$-integrin and pFAK-Y$^{397}$ levels in the response of naïve stellate cells to D-ECM. These results suggest that the observed requirements in naïve fibroblasts for D-ECM-regulated $\alpha_v\beta_5$-integrin induction of $\alpha$SMA (*Figure 3*) are concomitant with $\alpha_v\beta_5$-integrin control of redistributions and levels of active $\alpha_5\beta_1$-integrin and pFAK-Y$^{397}$.

We next examined whether stabilizing $\alpha_5\beta_1$-integrin activity with SNAKA51 altered its D-ECM-regulated intracellular relocation. For this, naïve pancreatic stellate cells were cultured overnight in D-ECM in the presence of Alexa 660 fluorophore pre-labeled SNAKA51 or of Alexa 660 pre-labeled isotype antibody control. Cells were then fixed with or without permeabilization and visualized with additional SNAKA51 pre-labeled with a different fluorophore (Alexa 488). SNAKA51 stabilization of $\alpha_5\beta_1$-integrin activity reduced intracellular, while increasing PM, pools of active $\alpha_5\beta_1$-integrin when compared to a non-specific IgG treatment control (*Figure 8A*). As an independent approach to confirm this result, double immunogold labeling of 3D-adhesions with mAb11 and active $\alpha_5\beta_1$-integrin with SNAKA51 was analyzed by transmitted electron microscopy of naïve fibroblastic stellate cells cultured within N-ECM, or of fibroblasts cultured in D-ECM while being treated with the $\alpha_v\beta_5$-integrin inhibiting antibody ALULA or a control antibody. D-ECM induced the intracellular enrichment of the active integrin in discrete punctate structures, which probably reflected endosomes that did not include recognizable clathrin-coated pits. Conversely, inhibition of $\alpha_v\beta_5$-integrin reduced the accumulation of active $\alpha_5\beta_1$-integrin intracellular pools, with most signal localized to the PM (*Figure 8B*). Supporting this mechanism in an independent model, the relocation of active $\alpha_5\beta_1$-integrin from intracellular to PM locations following culture within intact D-ECM was also observed in naïve $\beta_5$-integrin KO fibroblastic stellate cells that were compared to control naïve fibroblasts (*Figure 8—figure supplement 1*).

We directly tested whether the observed intracellular increase in active $\alpha_5\beta_1$-integrin corresponded to enrichment in discrete endosomal vesicles using a double labeling indirect immunofluorescence approach in naïve control and $\beta_5$-integrin KO pancreatic stellate cells plated into D-ECM. We compared the localization of activated $\alpha_5\beta_1$-integrin to that of proteins restricted to the early (EEA1 and Rab5) and late (Rab7 and Rab11) endosomal compartments, as well as that of a multivesicular endosomal marker (the tetraspanin CD81). D-ECM induced the partial localization of active $\alpha_5\beta_1$-integrin to Rab7-, Rab11-, and most clearly, to CD81-positive endosomes in control KO fibroblasts (*Figure 9*). By contrast, $\beta5$-KO naïve fibroblasts did not similarly relocalize active $\alpha_5\beta_1$-integrin to these intracellular locations following plating in D-ECM.

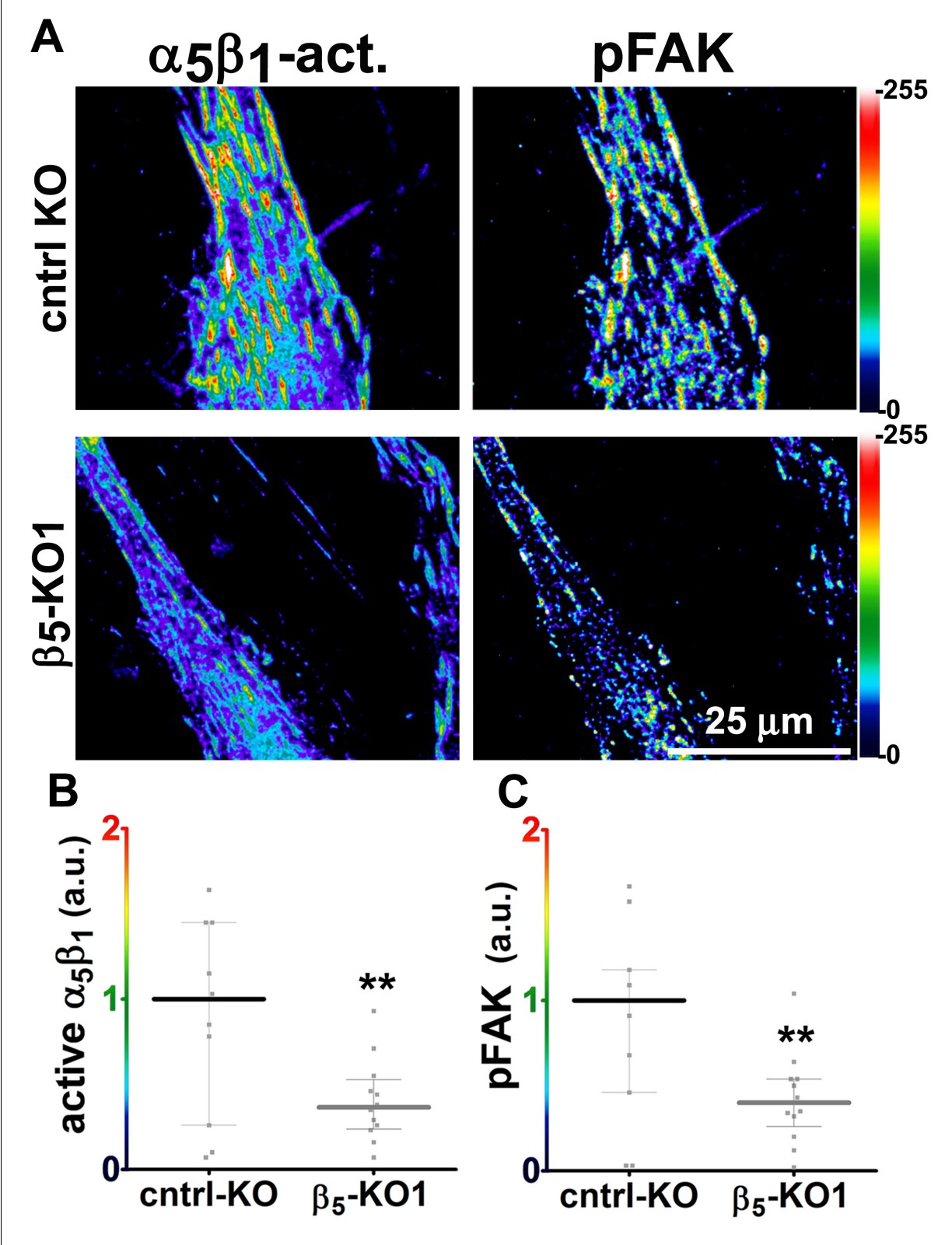

**Figure 7.** Loss of β5-integrin expression in naïve fibroblasts effectively reduces D-ECM-induced levels of active $\alpha_5\beta_1$-integrin and pFAK-$Y^{397}$. (A) Pseudocolored images showing the intensities of representative indirect immunofluorescence images indicating active $\alpha_5\beta_1$-integrin ($\alpha_5\beta_1$-act.) or pFAK from control KO (cntrl KO) or $\beta_5$-integrin KO ($\beta_5$-KO1) naïve fibroblasts. The fibroblasts were cultured overnight within D-ECMs. Intensity scale bars are
*Figure 7 continued on next page*

Figure 7 continued

shown to the right. (B and C) Quantification of total active $\alpha_5\beta_1$-integrin (**p=0.0420) (B) and pFAK-Y$^{397}$ (**p=0.0246) (C) levels of cells from (A). Note that both activities that were induced by D-ECM in naïve fibroblasts are lost in $\beta_5$-integrin KO naïve fibroblastic stellate cells.

The following figure supplement is available for figure 7:

**Figure supplement 1.** Loss of $\alpha_V$- or $\beta_3$-integrins does not significantly reduce overall levels of active $\alpha_5\beta_1$-integrin that is induced by D-ECM in naïve fibroblasts.

## The increase in levels of active $\alpha_5\beta_1$-integrin that is also evident in D-ECM-producing CAFs is $\alpha_v\beta_5$-integrin independent

We complemented the preceding analysis of naïve pancreatic integrin KO fibroblasts using the SMIA-CUKIE algorithm to evaluate active $\alpha_5\beta_1$-integrin and pFAK-Y$^{397}$ levels in 3D ECM-producing CAFs lacking specific integrin subunits. $\beta_5$-integrin KO CAFs, which have deficiencies in myofibroblastic expression of $\alpha$SMA and in the formation of anisotropic D-ECM (see *Figure 4—figure supplement 2*), had no discernible differences in levels of active $\alpha_5\beta_1$-integrin when compared to control KO CAFs. However, $\beta_5$-integrin KO CAFs cells had notable downregulation of pFAK-Y$^{397}$ expression, suggesting a disturbance in overall integrin signaling that resulted from the loss of this cell-matrix receptor (*Figure 10*). In contrast to the observed concomitant regulation of stress-fiber-localized $\alpha$SMA and of the increased active levels of $\alpha_5\beta_1$-integrin and FAK in naïve fibroblastic cells responding to D-ECM, a specific mechanistic decoupling between myofibroblastic features, such as $\alpha$SMA expression and anisotropic D-ECM production, and the regulation of active $\alpha_5\beta_1$-integrin in CAFs was observed in this analysis of $\beta_5$-integrin KO CAFs. This interpretation was further supported by comparative analysis of 3D ECM production in CAFs lacking other integrin subunits. Loss of $\alpha_v$-integrin in CAFs, which led to failure to produce 3D ECMs and to low $\alpha$SMA expression, did not affect levels of active $\alpha_5\beta_1$-integrin, whereas loss of the $\beta_3$-integrin subunit, which is associated with cells competent for induction of myofibroblastic features, significantly lowered active $\alpha_5\beta_1$-integrin levels (*Figure 10—figure supplement 1*).

The observed differences between the requirement in CAFs for $\alpha_v\beta_5$ integrin expression during the concomitant regulation of $\alpha$SMA levels and anisotropic D-ECM production, versus the dispensable nature of $\alpha_v\beta_5$ for the maintenance of high active $\alpha_5\beta_1$-integrin levels in these cells, suggests a functional decoupling between the processes.

## Fibroblasts isolated from RCC patients also display D-ECM- and integrin-dependent naïve-to-myofibroblastic activation

To determine whether the relationships between D-ECM, TGF$\beta$ and integrins described above pertain to additional human cancers of clinical relevance, we turned to a human renal cell carcinoma (RCC) stroma model (*Gupta et al., 2011*; *Goetz et al., 2011*). RCC stroma is of particular interest because it encompasses a microenvironment that is highly angiogenic and that physically intercalates with cancer cells (*Lohi et al., 1998*), thus distinguishing it from the desmoplasia seen in PDAC. We first demonstrated that renal (r) fibroblasts include an overrepresentation of $\alpha_5\beta_1$ and $\alpha_v\beta_5$ integrins relative to alternative $\beta$-integrin heterodimers at the plasma membrane (*Figure 11A*). Naïve r-fibroblasts expressed low levels of $\alpha$SMA (*Figure 11B–C*) and produced isotropic rN-ECM when compared to rCAF-derived matrices, which produced anisotropic rD-ECM (*Figure 11—figure supplement 1*). As with PDAC CAFs, treating rCAFs with a TGF$\beta$ inhibitor (SB-431542) during ECM production caused cells to produce isotropic matrices (*Figure 11—figure supplement 1*). Overnight plating of naïve r-fibroblasts in rD-ECM induced the formation of myofibroblastic features such as an increase in stress-fiber-localized $\alpha$SMA, as compared to cells plated in rN-ECM. Further, the isotropic 3D matrices produced by TGF$\beta$R1-inhibited rCAFs failed to induce naïve-to-myofibroblastic activation, whereas TGF$\beta$ signaling activity was dispensable in naïve fibroblasts undergoing myofibroblastic activation in response to intact rD-ECM induction.

In addition, as with the pancreatic model, regulation of active $\alpha_5\beta_1$-integrin by $\alpha_v\beta_5$-integrin was needed for naïve fibroblasts to undergo rD-ECM-induced myofibroblastic activation (*Figure 11—figure supplement 2* and *Table 5*). Accordingly, treatment of naïve r-fibroblasts seeded within rD-ECM

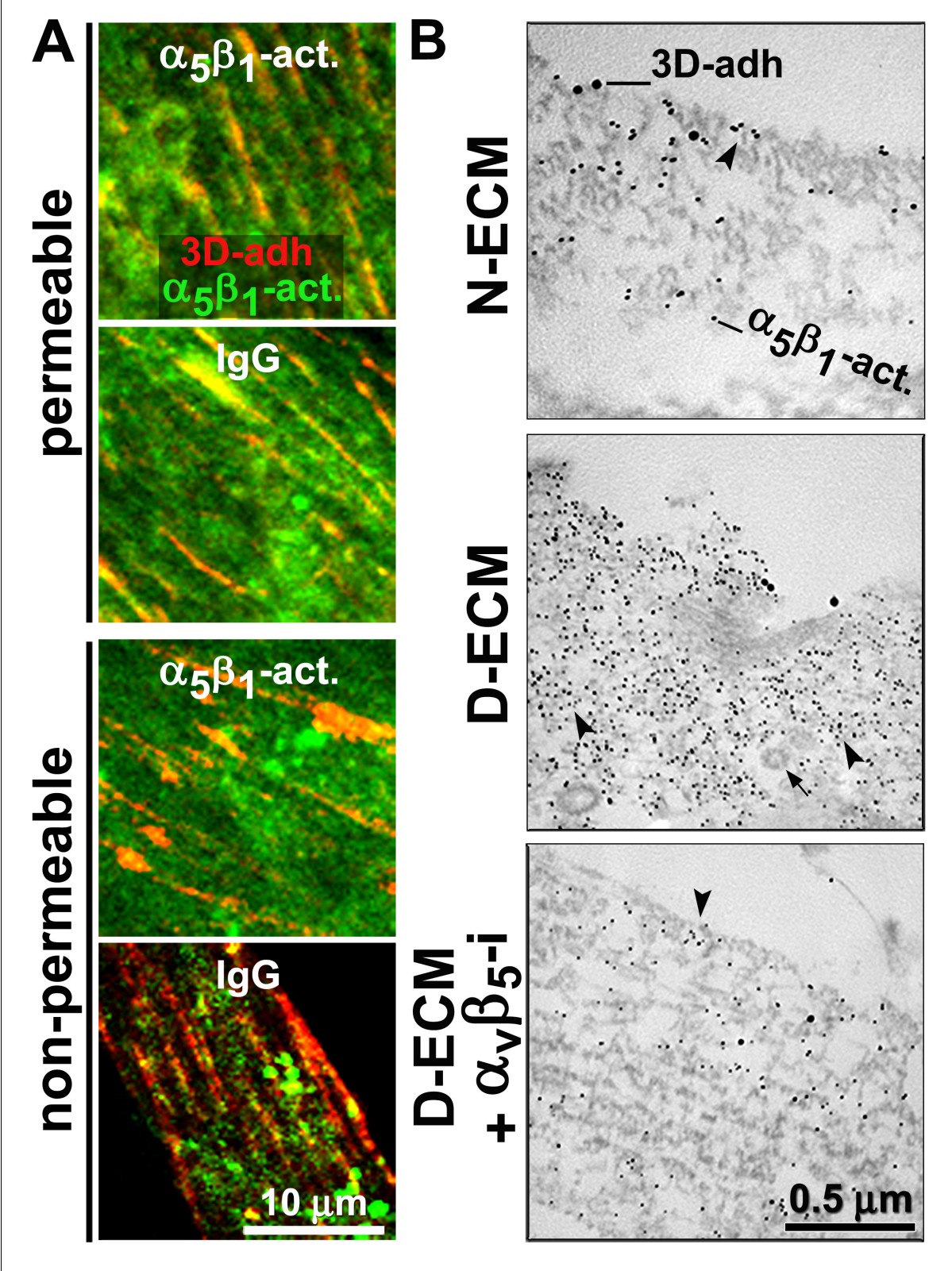

**Figure 8.** D-ECM prompts the internalization or relocation of $\alpha_5\beta_1$-integrin activity in an $\alpha_v\beta_5$-integrin-dependent manner. (**A**) Representative indirect immunofluorescent images corresponding to overnight 'chase' incubations with pre-labeled anti-active-$\alpha_5\beta_1$-integrin antibodies (SNAKA51 [*Clark et al., 2005*]) or IgG controls (blue, –not shown), followed by *de novo* detected active $\alpha_5\beta_1$-integrin labeling after fixation ($\alpha_5\beta_1$-act. in green) relative to 3D-adhesion structures (3D-adh. in red) under permeable vs. non-permeable conditions. Note how SNAKA51 treatment but not IgG prompts

*Figure 8 continued on next page*

Figure 8 continued

the relocation of integrin activity to the PM while there is practically no change between permeable and non-permeable active $\alpha_5\beta_1$-integrin levels. (B) Transmitted electron microscopy images of double immunogold-labeled 3D-adhesions (–3D-adh. large particles, [*Cukierman et al., 2001*]) vs. active $\alpha_5\beta_1$-integrin (-$\alpha_5\beta_1$-act. small particles, [*Clark et al., 2005*]), detected in naïve cells cultured within N–ECM vs. D-ECM in the presence or absence of $\alpha_v\beta_5$-integrin blockage using ALULA (*Su et al., 2007*) ($\alpha v\beta$5-i). Both reduction and relocation of active $\alpha_5\beta_1$-integrin pools are observed. Arrowheads point at random immunogold particles as examples, while the closed arrow indicates the location of a clathrin-coated vesicle.
The following figure supplement is available for figure 8:

**Figure supplement 1.** KO of $\beta_5$-integrin in naïve fibroblasts results in a redistribution of D-ECM-induced active $\alpha_5\beta_1$-integrin from intracellular pools back to the plasma membrane.

in the presence of small molecule FAK inhibitor PF573,228 (*Slack-Davis et al., 2007*), blocked acquisition of myofibroblastic traits (*Figure 11—figure supplement 3*). In naïve r-fibroblasts, concomitant

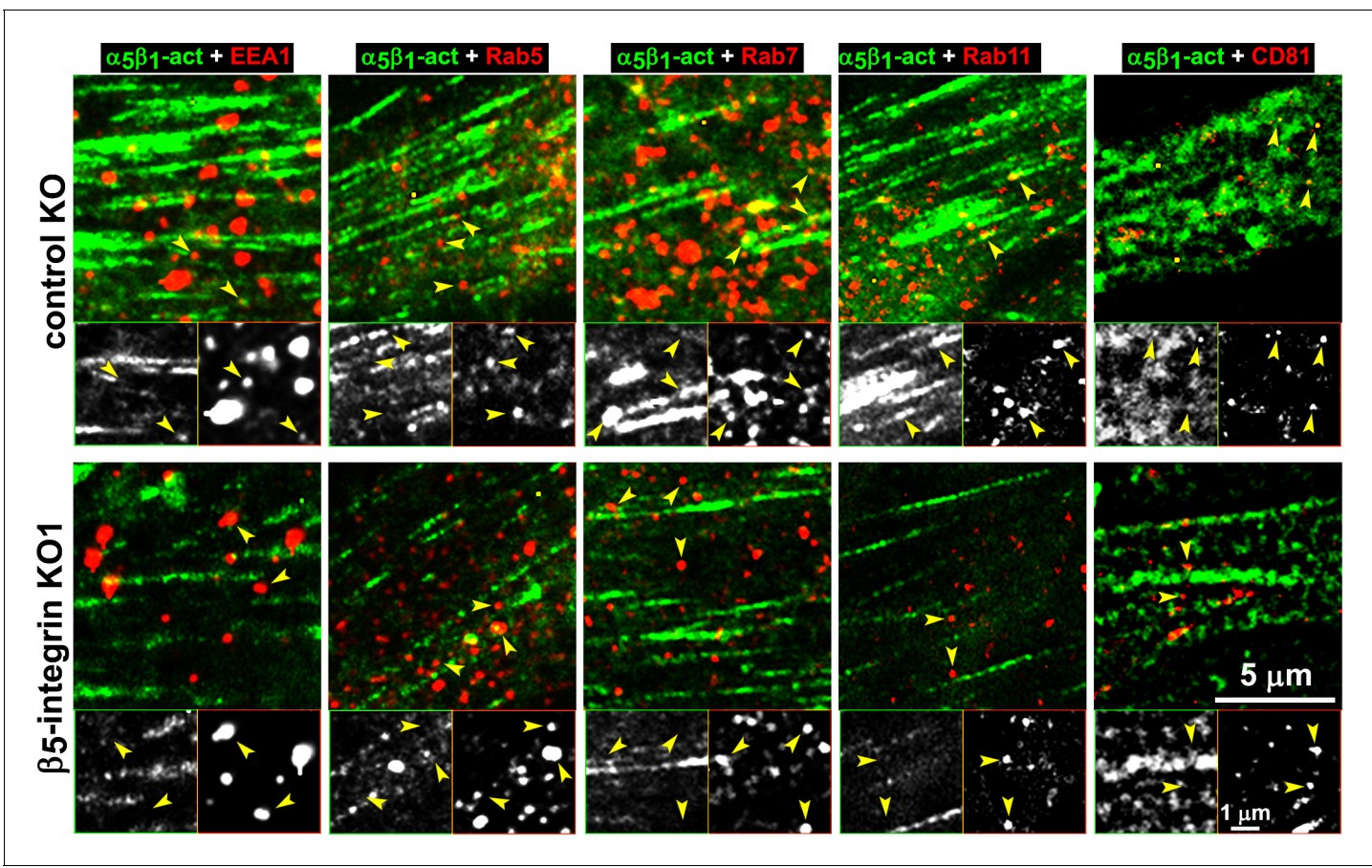

**Figure 9.** Loss of $\beta$5-integrin expression in naïve fibroblasts effectively reduces D-ECM-induced relocation of active $\alpha_5\beta_1$-integrin to late and multivesicular endosomes. Control KO (cntrl-KO) or $\beta_5$-integrin knock-out ($\beta_5$-KO1) naïve fibroblastic stellate cells were cultured overnight in desmoplastic-ECMs (D-ECM), and were subjected to indirect immunofluorescence using SNAKA51 to detect active $\alpha_5\beta_1$-integrin ($\alpha_5\beta_1$-act in green) in combination with one of the following endosomal markers shown in red: anti-EEA-1 (for early endosome), anti-Rab5 (for clathrin-mediated endocytosis early endosome), anti-Rab7 (late endosome to be degraded, recycled or rerouted), anti-Rab11 (late endosome to be recycled), or anti-CD81 (multivesicular endosomes). Top panels: confocal images were captured for each double-stained condition to identify the localization of active $\alpha_5\beta_1$-integrin in relation to the assorted types of endosomes shown in red. Yellow arrowheads point to assorted endosomes and are identically placed in the slightly zoomed monochromatic inserts shown below, which allow better appreciation of the relative locations of the markers vs. those of active $\alpha_5\beta_1$-integrin. Note that the partial co-localization of active $\alpha_5\beta_1$-integrin with Rab7, Rab11 and especially with CD81 is lost in the naïve $\beta_5$-KO compared to control KO naïve fibroblasts in response to D-ECM.

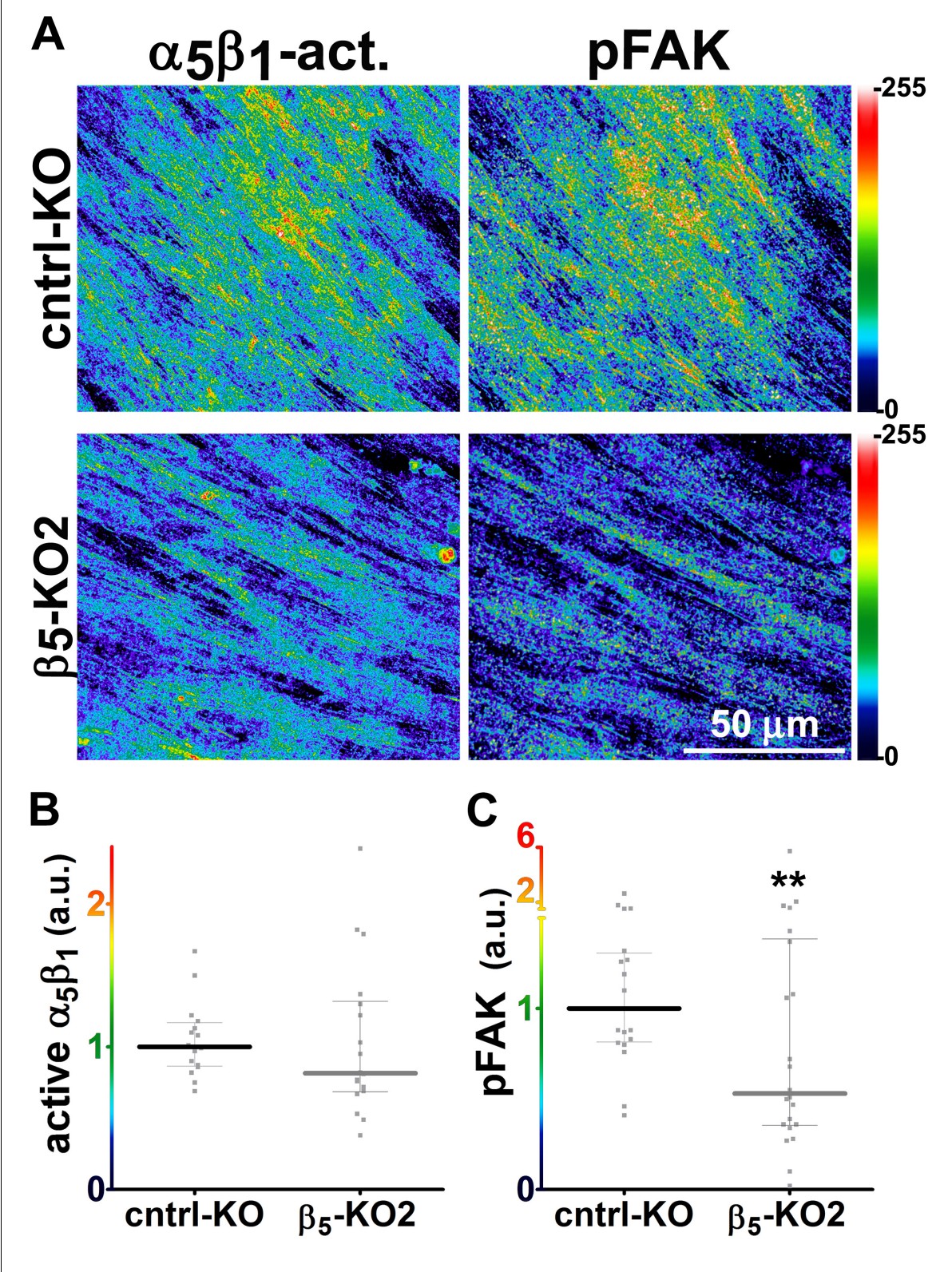

**Figure 10.** Loss of $\beta_5$-integrin in D-ECM-producing CAFs fails to significantly reduce overall levels of active $\alpha_5\beta_1$-integrin. Representative indirect immunofluorescence images showing active $\alpha_5\beta_1$-integrin (with SNAKA51 [*Clark et al., 2005*] [$\alpha_5\beta_1$-act.]) or pFAK in control KO (cntrl-KO) or $\beta_5$-integrin KO ($\beta$5-KO1) CAFs at the completion of the 3D matrix production process (see Materials and methods). (**B**-**C**) Graphs depicting levels of active $\alpha_5\beta_1$-

*Figure 10 continued on next page*

*Figure 10 continued*

integrin (B) or pFAK (C;**p=0.0465) in control and $\beta_5$-integrin KO CAFs from (A). Note that active $\alpha_5\beta_1$-integrin levels in CAFs were not significantly changed in response to $\beta_5$-integrin loss.

The following figure supplement is available for figure 10:

**Figure supplement 1.** Loss of $\beta_3$- but not of $\alpha_V$-integrin subunits in D-ECM-producing CAFs alters levels of active $\alpha_5\beta_1$-integrin.

FAK and $\alpha_5\beta_1$-integrin inhibition with combined PF573,228 and mAb16 significantly increased rD-ECM-induced myofibroblastic activation compared to PF573,228 treatment alone. Again, effects similar to those observed in the PDAC system were seen: $\alpha_v\beta_5$-integrin co-inhibition with FAK failed to rescue rD-ECM naïve-to-myofibroblastic conversion (*Figure 11—figure supplement 3*). Also, pFAK-Y$^{397}$ levels measured in 3D-adhesions of naïve r-fibroblasts induced by rD-ECM were effectively reduced in the presence of $\alpha_V\beta_5$-integrin inhibitor (ALULA [*Su et al., 2007*]), rendering pFAK-Y$^{397}$ levels similar to N-ECM levels (*Figure 11—figure supplement 4*). Culturing naïve r-fibroblasts in rD-ECM triggered lengthening of 3D-adhesion structures compared to those in rN-ECM, and this was effectively prevented by stabilization of $\alpha_5\beta_1$-integrin activity using SNAKA51 (*Clark et al., 2005*) (*Figure 11—figure supplement 5*). Last, plating naïve r-fibroblasts in rD-ECM increased levels of active $\alpha_5\beta_1$-integrin that was localized away from 3D-adhesion structures compared to those in rN-ECM. These levels were regulated by $\alpha_v\beta_5$-integrin, as treatment with ALULA restricted the enrichment of $\alpha_5\beta_1$-integrin activity in areas lacking adhesions (*Figure 11—figure supplement 6*).

Together, these results imply that the mechanisms identified for PDAC D-ECM production and activity were also relevant in RCC-associated fibroblastic stroma.

## *In vivo* stromal activity and distribution of SMAD2/3, $\alpha_5\beta_1$-integrin, and FAK predict patient survival and neoplastic recurrence

Reciprocal signaling between tumor and stromal cells occurs *in vivo* (*Tape et al., 2016*). Such bidirectional signaling is not simulated in the *in vitro* stroma 3D model used for the analysis described above. To extend our findings, we assessed whether the results we established *in vitro* were reflected *in vivo* in the original surgical samples used to harvest fibroblasts. We investigated whether the integrin redistribution phenotype that was commonly observed in both naïve fibroblasts in response to D-ECM and CAFs during D-ECM production *in vitro*, was also evident in the *in vivo* tumor-associated stroma. First, optimizing conditions *in vitro*, we used a simultaneous multi-channel immunofluorescence (SMI) approach, coupled with SMIA-CUKIE, to perform quantitative parallel analysis of seven relevant biomarkers in CAFs or control normal pancreatic stellate cells at the endpoint of production of D-ECM or N-ECM, respectively. These included a master mix (see Materials and methods) of epithelial and tumor cell markers (cyan; note, absent from *in vitro* specimens), the stromal marker vimentin (magenta), a nucleus-detecting DNA intercalating agent (Draq5, yellow), a 3D-adhesion marker (mAb11 [*Cukierman et al., 2001*], red) and SNAKA51 (*Clark et al., 2005*) to detect levels and localizations of active $\alpha_5\beta_1$-integrin (green), anti-pFAK-Y$^{397}$ (orange) and anti-pSMAD2/3 (indicative of TGF$\beta$ pathway activation [*Tsukazaki et al., 1998*]; blue).

This method reiterated our observation that during D-ECM production *in vitro*, CAFs maintain high levels of active $\alpha_5\beta_1$-integrin, with much of the active pool not localized to 3D-adhesions, as well as high levels of pFAK-Y$^{397}$ that are localized at 3D-adhesions. TGF$\beta$ activation also increased during the production of D-ECM, represented by augmented total and nuclear pSMAD2/3 levels, when compared to naïve/normal cultures in which active $\alpha_5\beta_1$-integrin mostly localizes at 3D-adhesion sites during N-ECM matrix production (*Figure 12*). We also conducted a parallel experiment using a classic indirect immunofluorescence approach to include a two-by-two marker comparison between active $\alpha_5\beta_1$-integrin and 3D-adhesions, pFAK or pSMAD2/3. This experiment assessed the precision of the SMI approach, and evaluated the extent of phenotypic similarities between the pancreatic and renal matrix-producing CAFs and normal fibroblastic systems *in vitro*. While normal pancreatic and renal 3D cultures included active $\alpha_5\beta_1$-integrin that is mostly localized at 3D-adhesion

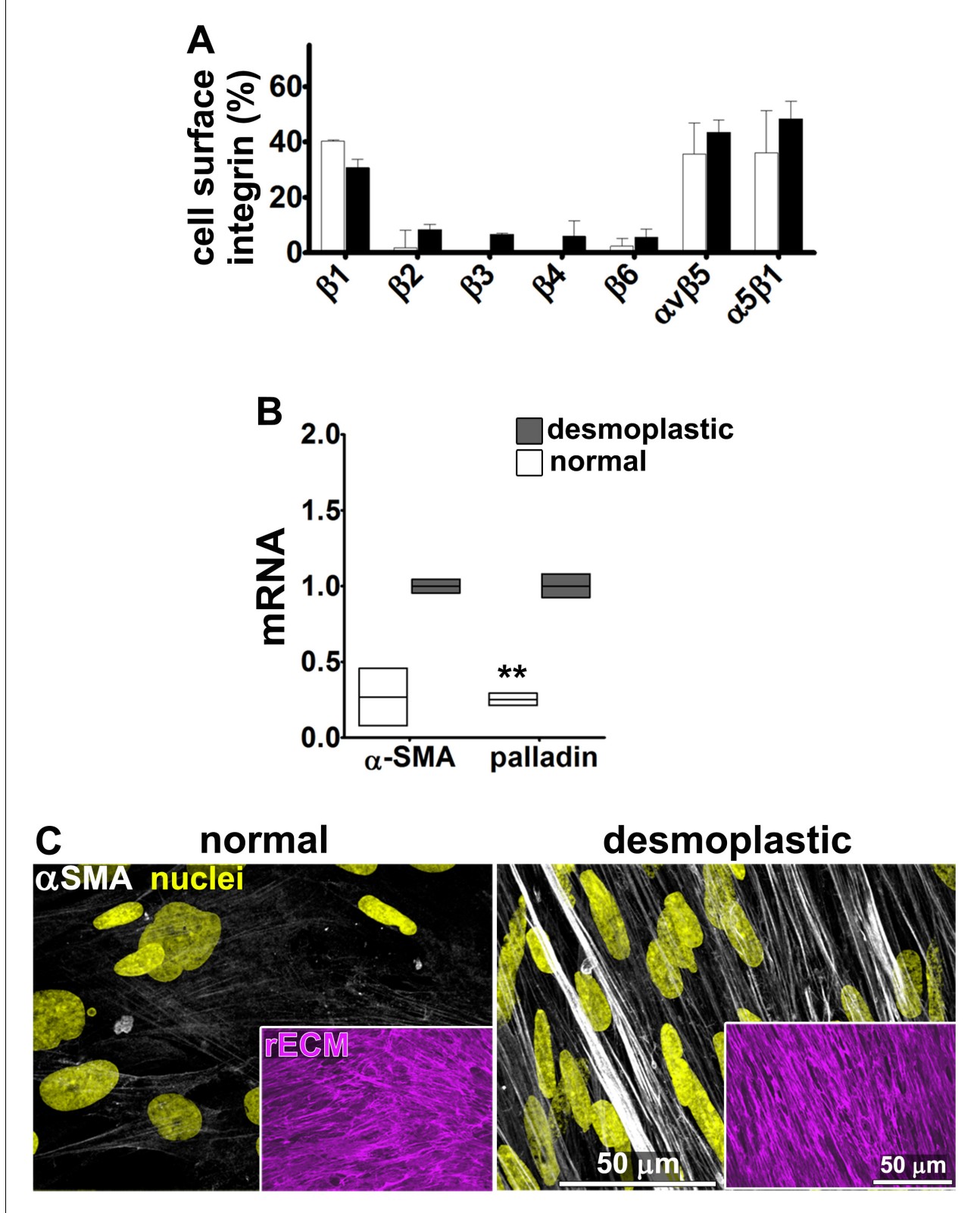

**Figure 11.** Renal fibroblasts present a similar profile to pancreatic fibroblasts during matrix production. Fibroblasts were isolated from RCC surgical pathologically normal or tumoral samples, and their ECM-producing phenotypes were assessed after seven days of matrix production. (**A**) An integrin-dependent cell adhesion array test was used to assess the PM expression of integrin heterodimers in primary fibroblasts isolated from normal (white bars) vs. tumoral (dark bars) tissues. Note that no differences were apparent between the two cell types with regards to levels of $\alpha_v\beta_5$ and $\alpha_5\beta_1$

*Figure 11 continued on next page*

*Figure 11 continued*

integrins. (B) Normal vs. desmoplastic mRNAs levels, corresponding to αSMA and palladin (used as an additional myofibroblastic marker as before) were obtained via RT-qPCR from the indicated 3D-cultures following renal ECM (rECM) production, which was achieved via confluent culturing of fibroblasts in the presence of ascorbic acid for a period lasting 8 days (*Franco-Barraza et al., 2016*) (\*\*p=0.0286). (C) Representative images of normal vs. desmoplastic phenotypes, subsequent to 3D rECM production, are shown; comparison of low vs. high αSMA levels (white), heterogeneous/round vs. elongated/spindled nuclei (yellow) and disorganized/isotropic vs. parallel aligned/anisotropic rECMs (magenta) are evident in the representative images.

The following figure supplements are available for figure 11:

**Figure supplement 1.** rD-ECM production by rCAFs is TGFβ-dependent.

**Figure supplement 2.** rD-ECM-induced renal naïve-to-myofibroblastic activation is reminiscent of pancreatic D-ECM fibroblastic stellate cell activation.

**Figure supplement 3.** FAK-independent $\alpha_5\beta_1$-integrin activity negatively regulates rD-ECM-induced naïve-to-myofibroblastic activation.

**Figure supplement 4.** rD-ECM-induced increase in pFAK levels localized at 3D-adhesions are regulated by $\alpha_v\beta_5$–integrin in naïve renal fibroblasts.

**Figure supplement 5.** rD-ECM regulates 3D-adhesion structure length, dependent on $\alpha_5\beta_1$-integrin activity.

**Figure supplement 6.** Naïve renal fibroblastic cells increase overall levels of active $\alpha_5\beta_1$-integrin in response to rD-ECM.

locations and included relatively low pFAK and pSMAD2/3 levels, PDAC and RCC CAF 3D matrix-producing cultures again exhibited high active $\alpha_5\beta_1$-integrin levels that were evident at locations

**Table 5.** αSMA stress fiber localization and expression levels in naïve renal fibroblasts cultured overnight within assorted renal ECMs.

| | | rN-ECM | rD-ECM | rD+TGFβi | rD+DMSO | TGFβi | DMSO | αvβ5-i | α5β1-i | 5+α5-i | α5β1-act | IgG |
|---|---|---|---|---|---|---|---|---|---|---|---|---|
| αSMA stress fiber localization | 25% percentile | 0.05 | 0.88 | 0.28 | 0.689 | 0.87 | 0.91 | 0.46 | 0.54 | 0.63 | 0.74 | 0.96 |
| | Median | 0.29 | 1.00 | 0.72 | 0.89 | 1.03 | 1.05 | 0.67 | 0.89 | 1.00 | 0.93 | 1.06 |
| | 75% percentile | 0.67 | 1.10 | 0.97 | 1.03 | 1.17 | 1.16 | 0.96 | 1.06 | 1.33 | 1.54 | 2.19 |

Values obtained from naïve renal fibroblastic cells cultured overnight within intact rD-ECMs (made from rCAFs) were used for normalization and assigned an arbitrary unit of 1.00. Assorted ECMs were intact r**N-ECM** or intact r**D-ECM,** while experimental conditions included D-ECMs made by CAFs treated with SB-431542 (r**D+TGFβi**) or DMSO (r**D+DMSO**) during ECM production. Experimental conditions during replating of naïve cells within rD-ECM also included treatment with SB-431542 (**TGFβi**) or vehicle (**DMSO**) as well as treatment with ALULA ($\alpha_v\beta_5$-**i**), mAb16 ($\alpha_5\beta_1$-**i**), ALULA plus mAb16 ($\beta_5+\alpha_5$-**i**), SNAKA ($\alpha_5\beta_1$-**act**) or control pre-immune antibody (**IgG**). Note that quantitative values for αSMA and F-actin immunofluorescence were used to calculate stress fiber localization and expression of **αSMA**.

P values, listed below, were calculated using the two-sided and two-tailed Mann Whitney test needed for normalized data.

rN-ECM vs. rD-ECM; **p<0.0001** stress fiber localization
rN-ECM vs. rD+TGFβi; **p=0.0221** stress fiber localization
rN-ECM vs. rD+DMSO; **p<0.0001** stress fiber localization
rD-ECM vs. rD+TGFβi; **p=0.0058** stress fiber localization
rD-ECM vs. rD+DMSO; p=0.1657 stress fiber localization
rD+TGFβi vs. rD+DMSO; **p=0.0630** stress fiber localization
TGFβi vs. DMSO; p=0.7996 stress fiber localization
TGFβi vs. rN-ECM; **p<0.0001** stress fiber localization
TGFβi vs. rD-ECM; p=0.6560 stress fiber localization
DMSO vs. rN-ECM; **p<0.0001** stress fiber localization
DMSO vs. rD-ECM; p=0.5042 stress fiber localization
$\alpha_v\beta_5$-i vs. IgG; **p=0.0001** stress fiber localization
$\alpha_5\beta_1$-i vs. IgG; **p=0.0008** stress fiber localization
$\beta_5+\alpha_5$-i vs. IgG; ***p=0.0880*** stress fiber localization
$\alpha_5\beta_1$-act vs. IgG; ***p=0.0665*** stress fiber localization

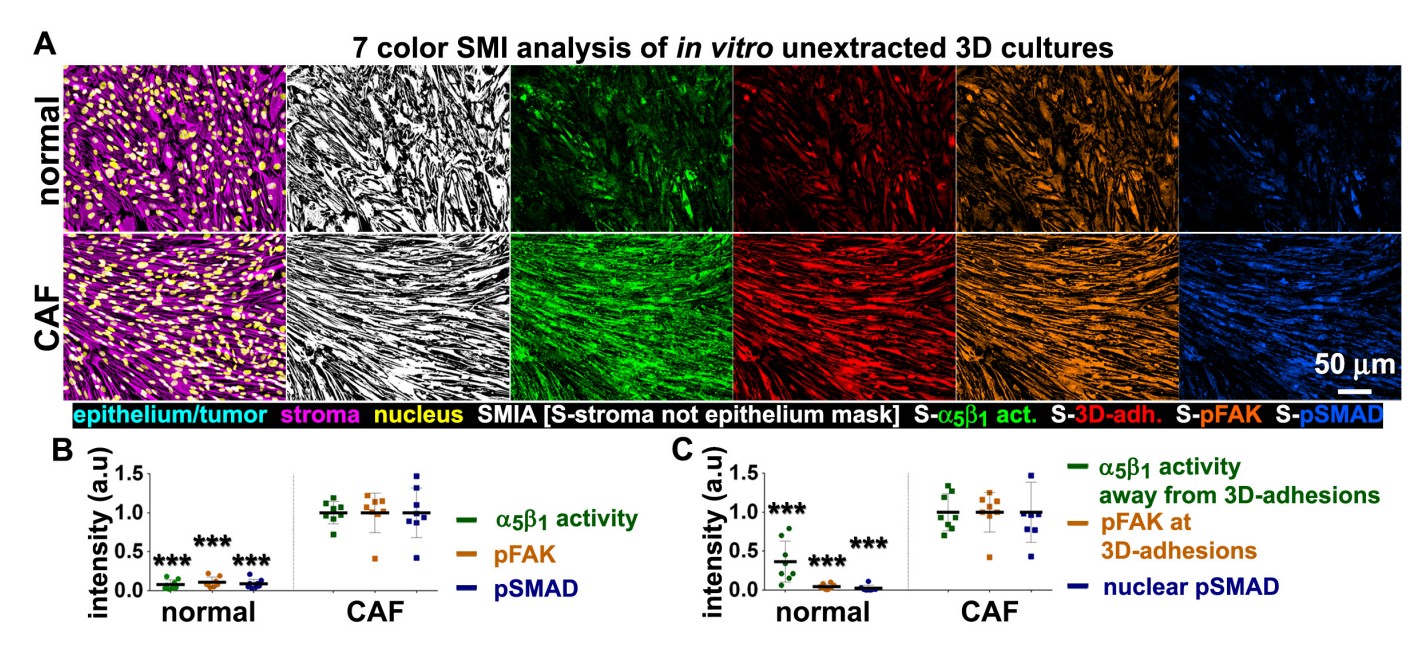

**Figure 12.** *In vitro* characterization of normal and tumor-associated 3D matrix-producing fibroblastic cultures provide a set of prognostic markers to be tested *in vivo*. (**A**) SMI approach image outputs of *in vitro* 3D cultures of naïve fibroblastic stellate cells (normal) and desmoplastic CAFs (tumor associated) during ECM production. Leftmost panels demonstrate the positive staining of vimentin (magenta), and the lack of cytokeratin (cyan), indicating the purity of the fibroblasts isolated; nuclei are marked in yellow. White masks (SMI approach; SMIA) in the next panel represent vimentin-positive/cytokeratin-negative (in this case vimentin-positive only as there is no cytokeratin present) areas as recognized by the software following threshold values provided by the user. Next panels represent the assorted markers localized at pixels corresponding to SMI-selected masks and conforming to: active $\alpha_5\beta_1$-integrin (S-$\alpha_5\beta_1$ act.; in green), 3D-adhesions (S-3D-adh.; in red), pFAK-$Y^{397}$ (S-pFAK; in orange), and pSMAD2/3 (S-pSMAD; in blue). (**B**) Graphs summarizing SMIA-CUKIE-generated data outputs representing median intensity levels of active $\alpha_5\beta_1$-integrin (green bullets), pFAK-$Y^{397}$ (orange bullets) and pSMAD2/3 (blue bullets) from data conditions as in (**A**) (***p=0.0002). (**C**) Graphs summarizing SMIA-CUKIE-generated data from marker intersections indicating mean intensity levels of $\alpha_5\beta_1$-integrin activity localized away from 3D-adhesions (green bullets) (***p=0.0008), pFAK-$Y^{397}$ at 3D-adhesions (orange bullets) (***p=0.0002), and nuclear pSMAD$_{2/3}$ (blue bullets) (***p=0.0002).

The following figure supplements are available for figure 12:

**Figure supplement 1.** *In vitro* characterization of pancreatic and renal normal and CAF unextracted 3D fibroblastic cultures.

**Figure supplement 2.** *In vitro* uncovered traits correspond to *in vivo* PDAC and RCC stromal phenotypes.

**Figure supplement 3.** Quantification of *in vivo* SMI approach validates *in vitro* findings.

away from 3D-adhesions, concomitant with increased pFAK-$Y^{397}$, and nuclei that were enriched with pSMAD2/3 (*Figure 12—figure supplement 1*).

Having validated the seven-color SMI approach *in vitro*, we used the same technique, combined with SMIA-CUKIE analyses, on formalin-fixed paraffin embedded (FFPE) samples matching the original tissues from the five PDAC and four RCC patients used to generate the 3D stroma models analyzed *in vitro* (*Figure 12—figure supplement 2*). This analysis indicated clear increases in total $\alpha_5\beta_1$-integrin activity, pFAK-$Y^{397}$, and pSMAD2/3 in the stroma of PDAC and RCC compared to normal pancreatic and renal parenchyma (*Figure 12—figure supplement 3A*). When the same comparisons were performed for $\alpha_5\beta_1$-integrin activity specifically localized away from stromal 3D-adhesion-positive pixels, the results again indicated clear increases (*Figure 12—figure supplement 3B*).

To determine whether this signature of CAF activation had prognostic value, we analyzed 128 PDAC and 126 RCC surgical specimens, using tissue microarrays (TMAs; *Table 6*). To establish baseline factors that are prognostic for survival, we initially evaluated the annotated clinical data for specimens on the TMAs (https://www.foxchase.org/sites/fccc/files/assets/cukierman_Franco-Barraza

**Table 6.** PDAC and RCC cohorts included in TMAs.

| PDAC | | RCC | |
|---|---|---|---|
| Samples | Number | Samples | Number |
| Tumor | 128 (128 OS and 102 DSS) | Tumor | 126 (116 OS and 115 DSS) |
| Gender | cases (%) | Gender | cases (%) |
| Female | 76 (59.40) | Female | 34 (27) |
| Male | 52 (40.60) | Male | 90 (71.4) |
| N/A | 0 (1.34) | N/A | 2 (1.6) |
| Average age years (range) | | Average age years (range) | |
| 67.51 (37–91) | | 60.67 (23–82) | |
| TNM stage | | TNM stage | |
| N | cases (%) | N | cases (%) |
| 0 | 33 (25.80) | 0 | 99 (78.57) |
| 1 | 91 (71.10) | 1 | 2 (1.59) |
| 2 | 0 (0) | 2 | 8 (6.35) |
| 3 | 0 (0) | 3 | 0 (0) |
| N/A | 4 (3.10) | N/A | 17 (13.49) |
| T | cases (%) | T | cases (%) |
| 0 | 0 (0) | 0 | 0 (0) |
| 1 | 10 (7.81) | 1 | 38 (30.16) |
| 2 | 34 (26.56) | 2 | 33 (26.19) |
| 3 | 66 (51.56) | 3 | 44 (34.92) |
| 4 | 13 (10.15) | 4 | 0 (0) |
| N/A | 5 (3.91) | N/A | 11 (8.73) |
| M | cases (%) | M | cases (%) |
| 0 | 109 (85.16) | 0 | 85 (67.46) |
| 1 | 6 (4.70) | 1 | 34 (26.98) |
| N/A | 13 (10.15) | N/A | 7 (5.56) |
| Overall stage | cases (%) | Overall stage | cases (%) |
| I | 17 (13.28) | I | 32 (25.4) |
| II | 38 (29.69) | II | 26 (20.63) |
| III | 53 (41.41) | III | 25 (19.84) |
| IV | 14 (10.94) | IV | 35 (27.8) |
| N/A | 6 (4.69) | N/A | 8 (6.35) |

N/A: Not available.

%20SMIA-CUKIE-Dec-2016.xlsx) using univariate (Uni) and multivariate (MVA) analyses. For PDAC specimens, association between pathological N stage , overall pathological stage and positive nodes with overall survival (OS) indicated a significant increased risk of death correlated with increasing levels of these variables, with Uni hazard ratios (HRs) of 2.5 for pathological N (p=0.0002; 95% CI 1.5–4.2), 1.3 for pathological stage (p=0.01; 95% CI 1.1–1.6) and 1.15 for positive nodes (p=0.0001; 95% CI 1.1–1.2). Similar analyses looking at the association between clinical variables and recurrence-free survival (RFS) resulted in the identification of positive correlations for pathological N (HR = 2.1; p=0.009; 95% CI 1.2–3.7), pathological stage (HR = 1.4; p=0.01; 95% CI 1.0–2.0) and positive nodes (HR = 1.1; p=0.005; 95% CI 1.0–1.2). Further, MVA analyses for OS in PDAC specimens showed an association for N stage: HR = 1.9; p=0.06; 95% CI 1.0–3.6.

We then asked whether the levels and localization of the stromal biomarker signature developed here yielded clinically useful prognostic biomarkers for PDAC, testing the idea that stromal pSMAD2/3, indicative of TGFβ signaling, would be associated with poor outcomes and active $\alpha_5\beta_1$-integrin at 3D adhesions would be associated with better survival. For this, we conducted high-throughput SMI image acquisition and SMIA-CUKIE analysis of TMA PDAC samples, using univariate CART methodology to integrate the reporting levels of numerical outputs and intersections of the seven biomarkers of interest with OS or RFS. We observed significantly shorter OS in tumor surgical samples that had high stromal pSMAD2/3 values, whether values were quantified as mean or median stromal pSMAD2/3 intensity levels (*Figure 13*). Even though many patients did not show clinical improvement following surgery (so that our cohort presented a bias towards fast recurrence), high stromal pSMAD2/3 levels also correlated with shorter RFS (*Figure 13—figure supplement 1*). In addition, higher mean quantified levels of stromal $\alpha_5\beta_1$-integrin activity also correlated significantly with longer RFS (*Figure 13—figure supplement 2A*). Interestingly, longer times to recurrence following surgery, quantified as the percentage area coverage relative to stromal occupied areas or as integrated intensities, were significantly associated with increased levels of integrin activity localized to stromal 3D-adhesion positive locations (*Figure 13—figure supplement 2B*). These results suggest that stromal activation of TGFβ, represented by high stromal pSMAD2/3 levels and indicative of capacity for active production of D-ECM production, is a prognostic stromal trait for poor outcome, while increased stromal levels of active $\alpha_5\beta_1$-integrin at 3D adhesion positive areas may constitute a patient-protective PDAC-associated desmoplastic phenotype.

To investigate whether these patterns are also seen in stroma associated with an additional epithelial cancer, we analyzed the RCC cohort (*Table 6*). Initial Uni and MVA analyses on the clinical data tested whether, in spite of the bias generated by the retrospective use of surgical cases, the cohort in question showed significant associations between pathological stages and OS and Disease-Specific Survival (DSS). Importantly, using CART analyses, we determined that RCC patients with high stromal levels of pFAK-Y[397] localized at 3D-adhesions, higher nuclear pSMAD2/3, or increased levels of active $\alpha_5\beta_1$-integrin, measured as mean, median or total stromal intensity levels as well as stromal percentage area coverages, had significantly shorter OS and DSS (*Figure 13—figure supplements 3* and *4*).

Together, these results strongly support the idea that the markers of an active desmoplastic phenotype, as defined herein, are useful in predicting patient outcomes.

## Discussion

Understanding the activity of and clinically exploiting tumor-associated stroma have for a long time posed challenges, given strong evidence for both tumor-limiting and tumor-promoting stromal functionality (*Mintz and Illmensee, 1975*; *Bissell and Hines, 2011*; *Langhans, 1879*; *Paget, 1889*; *Dvorak, 1986*). The data in this study are useful by providing insight into stromal function in several ways. First, by addressing prior studies that suggest that changes in tumor-associated ECM are necessary for TGFβ-dependent myofibroblastic activation (*Desmoulière et al., 1993*; *Serini et al., 1998*), they dissect the interaction of TGFβ with N-ECM versus D-ECM. These results indicate that TGFβ contributes to N-ECM assembly, and that it is essential solely for the phenotypic remodeling required for conversion to anisotropic D-ECM, but not for secretion of ECM components per se in the context of D-ECM production. This fact is supported by the observation that CAFs require TGFβ signaling for the myofibroblastic production of anisotropic D-ECM and for expression of αSMA, while the myofibroblastic features that are acquired by normal/naïve fibroblasts in response to D-ECM do not entail this signaling pathway. Second, our work addressed two cell-ECM receptor integrin heterodimers that have previously been shown to influence myofibroblastic activation, $\alpha_v\beta_5$ and $\alpha_5\beta_1$ (*Asano et al., 2006*; *Lygoe et al., 2004*; *Dugina et al., 2001*). Our study shows that both of these integrin heterodimers are abundant on the surface of both naïve fibroblasts and CAFs, but have distinct roles in the ability of naïve fibroblasts to respond to and of CAFs to assemble anisotropic D-ECM. In particular, our work integrated multiple approaches to emphasize an important role for $\alpha_v\beta_5$-integrin in the response of naïve fibroblasts to D-ECM and for CAFs in producing anisotropic matrix. We also found that while $\alpha_5\beta_1$ counteracted the response of naïve fibroblasts to D-ECM, it was, not surprisingly, essential for matrix fibrillogenesis. Interestingly, in the context of extracted D-ECM imparting naïve-to-myofibroblastic activation, cross-signaling between the two

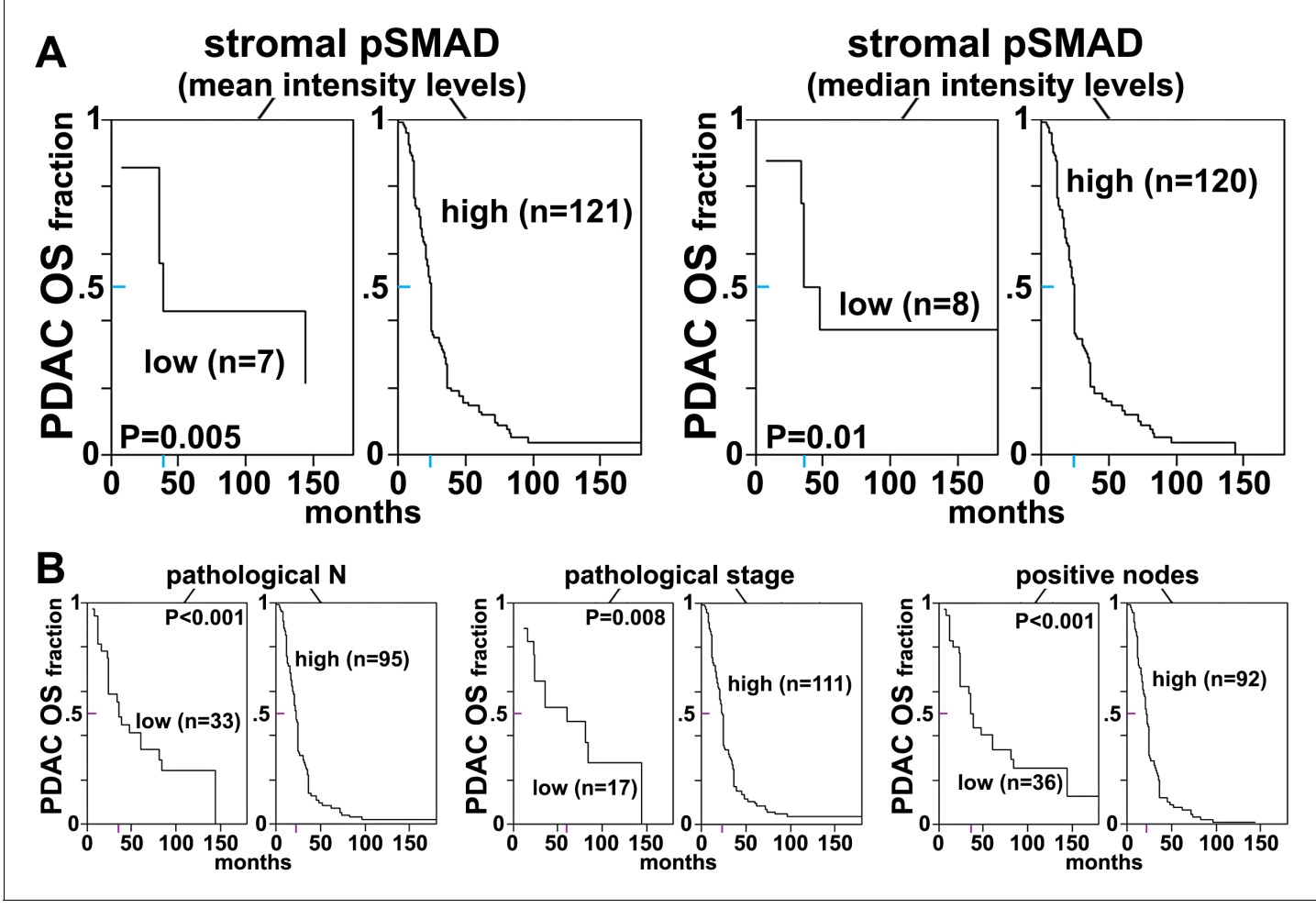

**Figure 13.** Stromal pSMAD2/3 levels are predictive of poor overall survival in PDAC patients. (**A**) CART-generated survival curves depicting overall survival (OS) as a function of stromal pSMAD2/3 expression in PDAC patients. Left and right curves were obtained using mean and median intensity levels, generated by SMIA-CUKIE, of stromal pSMAD2/3 related to OS, respectively. (**B**) Survival curves depicting pathological N, stage, and node status with PDAC patient OS. Colored tick-lines crossing Y axes indicate 0.5 survival marks and correspond to X axes locations that mark the median survival times obtained from the assorted curves. P values are shown. All patient data as well as SMIA-CUKIE generated data corresponding to human cohort constructed TMAs (shown in **Table 6**) can be found in the online table at the following publically available link: https://www.foxchase.org/sites/fccc/files/assets/cukierman_Franco-Barraza%20SMIA-CUKIE-Dec-2016.xlsx. Note that since our cohorts comprised of samples obtained from surgeries, early neoplastic stages were overrepresented.

The following figure supplements are available for figure 13:

**Figure supplement 1.** Stromal pSMAD2/3 levels are predictive of short recurrence-free survival in PDAC patients.

**Figure supplement 2.** Stromal active $\alpha_5\beta_1$-integrin levels localized at 3D-adhesions correlate with longer recurrence-free survival in PDAC patients.

**Figure supplement 3.** Stromal levels and locations of p-SMAD2/3, active $\alpha_5\beta_1$-integrin, and pFAK correlate with short overall survival in RCC patients.

**Figure supplement 4.** Stromal levels and locations of p-SMAD2/3, active $\alpha_5\beta_1$-integrin, and pFAK correlate with short disease specific survival in RCC patients.

integrins was apparent, with $\alpha_v\beta_5$ regulating the internalization of part of the elevated pool of activated $\alpha_5\beta_1$ to intracellular endosomal compartments (**Figure 14**). Third, we also defined specific ECM responses as independent of, or dependent on, the canonical integrin effector FAK. Fourth, importantly, we demonstrated that the signaling relationships identified in this study were valid in

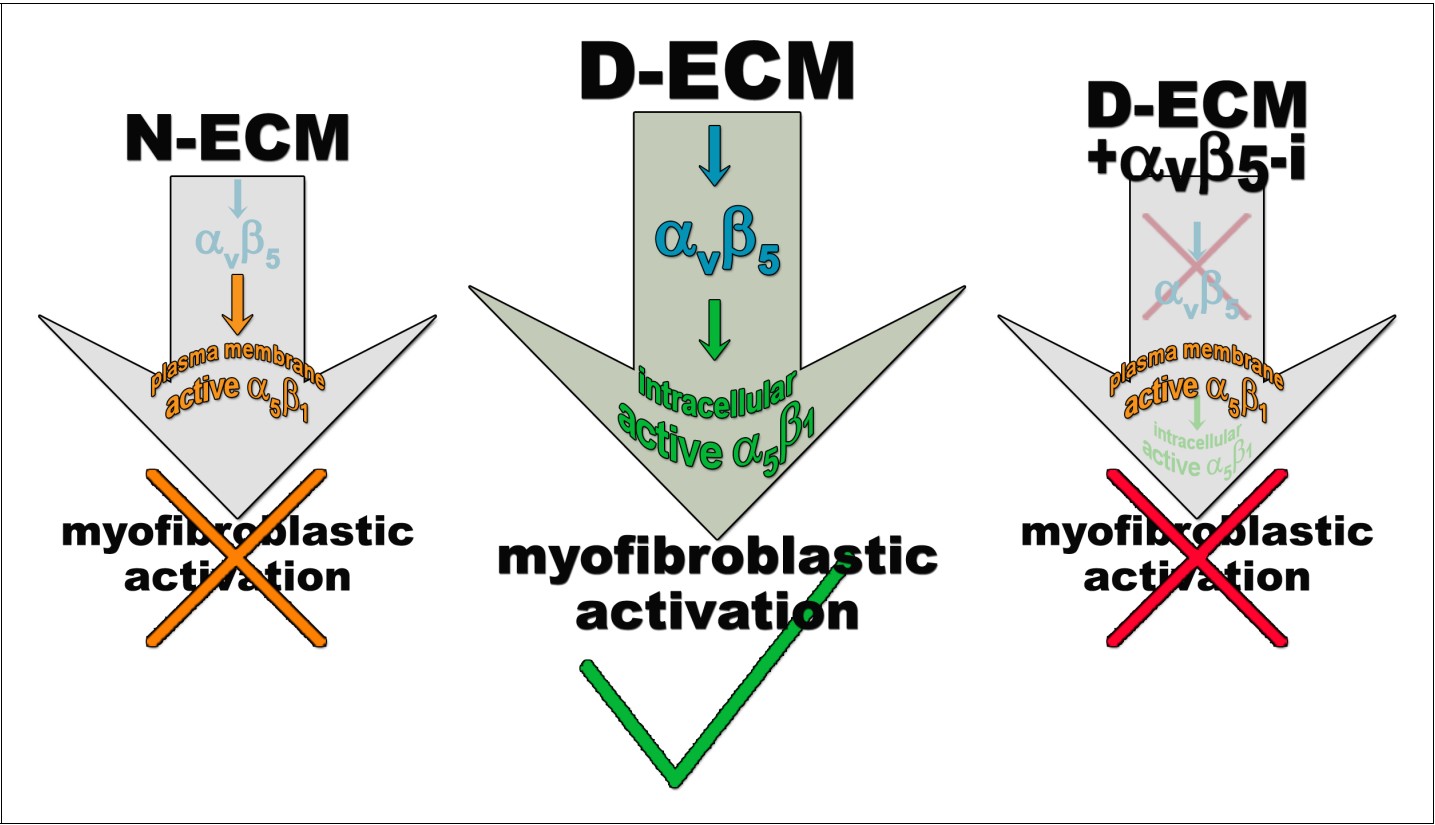

**Figure 14.** Model depicting D-ECM-induced $\alpha_V\beta_5$ regulation of active $\alpha_5\beta_1$-integrin localization during naïve-to-myofibroblastic activation. Essentially, naïve fibroblasts grown in N-ECMs do not trigger a surplus in active $\alpha_V\beta_5$ and $\alpha_5\beta_1$-integrin conformations. Consequently, the non-pathological N-ECM spares regulation of $\alpha_V\beta_5$-integrin activity and assures physiological accumulation of low active $\alpha_5\beta_1$-integrin levels at the PM, rendering fibroblasts inactive. D-ECMs induce an excess of active $\alpha_5\beta_1$-integrin and regulate $\alpha_V\beta_5$-integrin-mediated relocation of active $\alpha_5\beta_1$-integrin from the PM to intracellular (e.g., late endosomal) pools, resulting in myofibroblastic activation. Last, this D-ECM activity is dependent on the activity of $\alpha_V\beta_5$-integrin, as inhibition of $\alpha_V\beta_5$-integrin maintains active $\alpha_5\beta_1$-integrin at the PM (similarly to N-ECM regulation of active $\alpha_5\beta_1$-integrin locations), which averts D-ECM-induced myofibroblastic activation. The summarized data provide a possible explanation for the early model presented in *Figure 3G*, which called for a D-ECM control of $\alpha_V\beta_5$ regulation of active $\alpha_5\beta_1$ and suggests that the alluded to regulation constitutes a re-localization of active $\alpha_5\beta_1$ to intracellular endosomal compartments, allowing D-ECM-induced naïve-to-myofibroblastic activation.

multiple model systems, including human primary PDAC and RCC, as well as additional murine models. Fifth, we used high-content imaging and data analysis to demonstrate that interactions analyzed *in vitro* can be used to define an *in vivo* stromal signature that has prognostic value, addressing a significant clinical goal.

There is strong evidence for desmoplastic stromal reciprocation, involving altered ECM (e.g., D-ECM) and adjacent tumor and/or stromal cells during PDAC development and progression (*Xu et al., 2009*; *Tape et al., 2016*; *Alexander and Cukierman, 2016*; *Garcia et al., 2014*; *Roskelley and Bissell, 1995*). Several prior studies have searched for proteomic signatures of normal vs. tumor-affected stroma (*Webber et al., 2016*), in order to identify stromal traits that are associated with favorable or unfavorable clinical outcomes (*Erkan et al., 2008*; *Moffitt et al., 2015*). Some of these signatures have been proposed to serve as clinically relevant tumor sub-type indicators (*Choi et al., 2014*). The desmoplastic 3D models used in this study allowed us to functionally dissect both mechanisms of D-ECM production by CAFs, including the specific induction of anisotropic ECMs, and D-ECM-induced responses such as myofibroblastic activation of naïve fibroblasts, mechanistically informing the significance of such signatures.

A role for TGF$\beta$ in regulating the production of anisotropic ECM had been previously suggested (*Khan and Marshall, 2016*). Remodeling that causes isotropic ECMs to become increasingly anisotropic, together with increases in matrix tension and stiffness, are all features of chronic fibrosis that

physically enhance the accessibility and activation of TGF$\beta$ by myofibroblastic integrins (*Klingberg et al., 2014*). Another interesting study demonstrated that substrate stiffness influences the ability of a cell induced *in vitro* to undergo myofibroblastic activation, and can positively or negatively influence the degree to which a cell is able to accurately acquire the characteristics (*Achterberg et al., 2014*) of a CAF *in vivo*. In accord with this line of thinking, one may argue that specific ECM topographies, and in particular the overall physical properties of cell substrates, play key pathophysiological roles in promoting positive feedback loops involving myofibroblastic cells and D-ECMs in fibrous tumor-associated microenvironments, such as PDAC-associated desmoplasia (*Alexander and Cukierman, 2016*). Our results indicated that while TGF$\beta$ is necessary for normal fibroblasts to assemble N-ECM, it is essential solely for the phenotypic remodeling of anisotropic D-ECM by CAFs, and not for its production. Negating CAF capability to produce anisotropic D-ECM restricts tumor growth *in vivo* (*Goreczny et al., 2016*), while increased levels of anisotropic D-ECM correlate with poor patient survival and cancer relapse (*Goetz et al., 2011*; *Conklin et al., 2011*; *Bredfeldt et al., 2014*). Inhibition of the TGF$\beta$ type I receptor, which reduces the activity of a downstream canonical effector of TGF$\beta$, SMAD2/3, is known to hinder matrix fibrillogenesis (*Callahan et al., 2002*). In our work, elevated fibroblast-expressed pSMAD2/3 and nuclear translocation of this protein in CAFs producing D-ECM correlated with short time to recurrence and reduced survival in both PDAC and RCC patient cohorts. Results from our study also suggest that while TGF$\beta$-targeted therapeutic interventions may revert D-ECMs to their naturally tumor-suppressive isotropic phenotype, the same approach could ablate homeostatic N-ECM. Consideration of this possibility is important because total ablation of TGF$\beta$ induces multifocal inflammatory disease (*Shull et al., 1992*) and because clinical trials are currently assessing TGF$\beta$-inhibitory drugs (*Herbertz et al., 2015*).

While the understudied $\alpha_v\beta_1$ heterodimer was recently recognized as a regulator of latent TGF$\beta$ in fibrosis (*Reed et al., 2015*), $\alpha_v\beta_5$ and $\alpha_5\beta_1$ integrins are involved in myofibroblastic activation (*Asano et al., 2006*; *Lygoe et al., 2004*; *Dugina et al., 2001*). Both $\alpha_v\beta_5$ and $\alpha_5\beta_1$ integrin receptors were abundant in primary fibroblasts collected from patients. Our results demonstrated that in response to D-ECM, $\alpha_v\beta_5$-integrin activity supports the relocalization of active $\alpha_5\beta_1$-integrin to intracellular endosomal pools and that this vesicular accumulation is needed for the maintenance of naïve-to-myofibroblastic activation. In addition, genetic loss (but not transient inhibition) of $\alpha_5\beta_1$, which resulted in slow-growing naïve cells, altogether prevented *de novo* D-ECM induction of naïve-to-myofibroblastic transition. Interestingly, in other studies, $\alpha_5\beta_1$-positive elongated 'super mature' adhesion structures (*Dugina et al., 2001*; *Hinz et al., 2003*), which control tension-dependent recruitment and retention of $\alpha$SMA at stress fibers, were found to play a functional role in myofibroblastic activation (*Goffin et al., 2006*). Together with our findings, these data suggest that $\alpha_5\beta_1$ participates in the regulation of $\alpha$SMA expression, whereas the $\alpha_v\beta_5$-regulated relocation of activated $\alpha_5\beta_1$-integrin participates in $\alpha$SMA localization to stress fibers. Other studies have shown that inhibition of $\alpha_v\beta_5$-integrin activates $\beta_1$-integrin, counteracting ECM-dependent myofibroblastic $\alpha$SMA stress fiber localization (*Wang et al., 2012*). However, the nature of the specific $\beta_1$-integrin heterodimer and the mechanistic details of this negative myofibroblastic regulation were previously unknown. We further defined the $\alpha_5\beta_1$ activity that regulates $\alpha$SMA localization as independent of FAK. Interestingly, a previously defined $\alpha_5\beta_1$ integrin activity needed to form 3D-adhesions in both normal cell-derived ECMs and *in vivo* is also FAK-independent (*Cukierman et al., 2001*).

Intriguingly, our results indicate that D-ECM induces $\alpha_v\beta_5$-integrin-dependent redistribution of active $\alpha_5\beta_1$-integrin to intracellular pools, within structures that include CD81-positive multivesicular (*Escola et al., 1998*) and late Rab7/Rab11 endosomes. We found that using SNAKA51 to stabilize $\alpha_5\beta_1$-integrin activity resulted in reduced internalization, a finding that differs from results reported previously which indicated that treating cells with SNAKA51 internalized active integrin (*Mana et al., 2016*). It is possible that these differences reflect differences in analytic time points. *Mana et al. (2016)* looked at a time point of 20 min after treatment, whereas we analyzed data after overnight incubation with SNAKA51, which may have led to compensatory enrichment of activity at the PM in our study, and thus further investigation is required. However, it is interesting that engagement of the integrin-binding ligands RGD, osteopontin and cilengitide with $\alpha_v\beta_3$ and $\alpha_5\beta_1$ results in the association of $\alpha_5\beta_1$ with a Rab-coupling protein which regulates its localization concomitant with EGFR (*Caswell et al., 2008*). It is therefore possible that the use of ALULA (*Clark et al.,*

*2005*) in this study triggered activation of the Rab cellular trafficking machinery, in a manner restricted by $\alpha_v\beta_5$ activity during tumor-associated desmoplasia.

The importance of identifying reliable biomarkers for early PDAC diagnostics (*Laeseke et al., 2015*) and improved RCC prognostic predictions is clear. Recently, increased stromal density was deemed to be a better prognostic outcome in a retrospective study looking at pancreatic cancer patients who underwent pancreaticoduodenectomy followed by adjuvant treatment, while assessing overall levels of stromal $\alpha$SMA was found to be inconclusive (*Bever et al., 2015*). In the future, it will be interesting to test whether the increased stromal density, perhaps in the context of specific relocalization of $\alpha$SMA, corresponds to this study's 'patient-protective' phenotype. This example, together with additional studies that suggest both tumor-protective and -detrimental roles for highly heterogeneous desmoplasia (*Özdemir et al., 2014*; *Rhim et al., 2014*; *Sherman et al., 2014*; *Bever et al., 2015*; *Dekker et al., 2015*; *Sasson et al., 2003*), underscore the need for better stromal categorization. The new software developed in this study to facilitate analysis of the discrete localizations of the various markers specifically in stroma (freely downloadable from https://github.com/cukie/SMIA_CUKIE) allows analysis of batch-sorted monochromatic images in bulk, and generation of qualitative and quantitative outputs to evaluate biomarkers located at any given mask and/or marker intersection. In this study, we used this software to identify predictive biomarker signatures for PDAC and RCC patients. Stromal increases in pSMAD2/3 and pFAK correlated with worse outcomes but, in spite of the bias introduced by the use of resectable surgical cases, we identified a patient-protective stroma phenotype in PDAC, in which increased active $\alpha_5\beta_1$-integrin selectively localized at stromal 3D-adhesions is associated with improved outcomes. By contrast, intensities of active $\alpha_5\beta_1$-integrin activity correlated with increasingly unfavorable OS as well as DSS in RCC patients. Hence, locations and intensities of stromal $\alpha_5\beta_1$-integrin activity concomitant with constitutive active pFAK-Y$^{397}$ could represent novel desmoplasia-dependent outcome-predicting stromal patterns. As our *in vitro* studies suggested a complex function for $\alpha_5\beta_1$-integrin activity, and given the differences between the molecular characteristics of distinct tumor subtypes, we emphasize that individual tumor classes will require careful study before these biomarkers could be used as predictive clinical indicators.

In summary, our study provides new insights into the intracellular and extracellular signaling processes that support desmoplastic formation, with mechanistic studies facilitated by the use of *in vitro* fibroblast-derived 3D ECMs. Specifically, the results suggest that $\alpha_v\beta_5$-integrin prevents FAK-independent $\alpha_5\beta_1$-integrin activity from impairing D-ECM-induced naïve-to-myofibroblastic activation by relocating this activity to multivesicular and other late endosomes. $\beta_5$-integrin's contribution to the ability of CAFs to produce functional anisotropic D-ECM, in which active $\alpha_5\beta_1$-integrin levels are not altered, distinguishes between naïve-to-myofibroblastic activation and myofibroblastic CAFs. Importantly, this study supports the idea that stromal normalization, to produce a form that plays a tumor-suppressive role (*Mintz and Illmensee, 1975*; *Soto and Sonnenschein, 2011*; *Xu et al., 2009*; *Bissell and Hines, 2011*; *Alexander and Cukierman, 2016*) may be a favorable approach, especially as some recent attempts aimed at completely abolishing stroma have proven harmful to patients (*Bijlsma and van Laarhoven, 2015*; *Özdemir et al., 2014*; *Rhim et al., 2014*). This work also suggests specific protein targets that may be useful for stroma-normalizing therapies. For example, these data suggest that future therapeutic strategies might involve combinatorial treatments to inhibit TGF$\beta$ and $\alpha_v\beta_5$-integrin. Finally, the study also suggests that monitoring the level and localization of stromal pSMAD2/3, as well as those of $\alpha_5\beta_1$-integrin activity together with those of stromal pFAK, could assist in stratifying individuals that would benefit from $\alpha_v\beta_5$-integrin and other chronic fibrosis-directed therapeutics.

## Materials and methods

### Ethics statement

Human tissues were collected under exemption-approval of the Center's Institutional Review Board after patients signed a written informed consent agreeing to donate samples to research. To protect patients' identities, samples were coded and distributed by the Institutional Biosample Repository Facility.

## Reagents and cell lines

All reagents (see below) were from Sigma-Aldrich (St. Louis, MO) unless listed otherwise. All primary and immortalized fibroblasts were cultured at 37°C under 5% $CO_2$ using Dulbecco's Modified Eagle's Medium (Mediatech (Manassas, VA)) supplemented with 10% or 15% (murine or human cells, respectively) Premium-Select Fetal Bovine Serum (Atlanta Biologicals (Lawrenceville, GA)), 2 mM L-glutamine and 100 u/ml-μg/ml penicillin-streptomycin. Panc1, FAK$^{+/+}$, FAK$^{-/-}$, SYF, and littermate wild-type control cells were from the American Tissue Culture Collection (Manassas, VA). FAK-KD and hTert controls were a gift from David Schlaepfer (UCSF, San Francisco, CA).

A request to annotate the human fibroblastic cell lines that were generated in this study was issued to the Research Resource Identifiers for their quarterly year annotations. Also, these cells were submitted for CellCheckTM 16 Marker STR Profile and Inter-species Contamination Test for human cell line authentication to IDEXX Bioresearch (Columbia, MO). Markers evaluated to authenticate the diversity of cell origins (patients) were: AMEL, CSF1PO, D13S317, D16S539, D18S51, D21S11, D3S1358, D5S818, D7S820, D8S1179, FGA, Penta_D, Penta_E, TH01, TPOX, and vWA. Interspecies contamination testing included comparison of human samples versus mouse, rat, Chinese hamster and African green monkey DNA. Mycoplasma contamination status was negative. This study did not utilize cell lines listed in the Database of Cross-Contaminated or Misidentified Cell Lines (Version 7.2) from the International Cell Line Authentication Committee (ICLAC).

## Isolation and characterization of fibroblastic cells

Pan-keratin (AE1/AE3 -Dako (Carpinteria, CA)) negative and vimentin (EPR3776 -Abcam (Cambridge, MA)) positive fibroblastic cells were isolated from fresh surgical normal and tumoral tissue samples using the enzymatic tissue digestion approach as published (*Franco-Barraza et al., 2016*). Fresh surgical tissues from normal or tumoral pancreas and from kidney or murine skin (see below) were collected into conical tubes containing Dulbecco's phosphate-buffered saline with antibiotics at 4°C. Samples were minced and subjected to overnight gelatinase digestion. Digested tissues were subjected to 10 min 200 g centrifugation followed by three sequential size-exclusion filtrations using sterile nylon mesh strainers of pore sizes 500 μm, 100 μm and 40 μm. The resulting cells were characterized as naive or CAFs according to methods reported previously (*Gupta et al., 2011*; *Goetz et al., 2011*; *Amatangelo et al., 2005*). Cells were used for no longer than 12 passages while maintained within their own derived ECMs (if cells were not to be maintained in 3D ECMs, only 4–6 passages would have been used).

Fibroblasts used in the study were harvested from five different patients for PDAC (including two matched pairs) and four for RCC models, while the fibroblasts used for the murine model of skin squamous-cell-carcinoma-associated desmoplasia were from our published study in which the D-ECM-induced naïve-to-myofibroblastic transition was first described (*Amatangelo et al., 2005*). Note that PDAC fibroblastic cells from patient #1 were used *in vitro* throughout the study unless stated otherwise.

## Preparation of fibroblast-derived 3D ECMs

We obtained cell-derived ECMs using our published protocols (*Franco-Barraza et al., 2016*). Confluent fibroblastic cultures were maintained for 7 days in the presence of daily added and freshly prepared ascorbic acid at a concentration of 50 ìg/ml. The resulting 'unextracted' 8 day long multilayered 3D cultures were used as naïve or CAF-matrix—producing cultures in this study and were processed for phenotypic characterization and ECM alignment (isotropy vs. anisotropy assessments). Cell extraction, to obtain N- and D-ECMs, was achieved using 0.5% (v/v) Triton X-100 complemented with freshly added 20 mM $NH_4OH$. Extracted N- and D-ECMs were stored sealed, using Parafilm, in culturing plates with Dulbecco's phosphate-buffered saline lacking $CaCl_2$ and $MgSO_4$ at 4°C. The resulting extracted matrices were used for D-ECM-induction of naïve-to-myofibroblastic activation. For the production of 3D matrices using inhibitors, the following were added during the 7 days of ECM production: TGF$\beta$1R small molecule inhibitor (SB431542) (25 μM); DMSO; conformation-dependent anti-active $\alpha_5$-integrin functional antibody (*Clark et al., 2005*) (45 μg/ml SNAKA51 [a gift from Martin Humphries, Manchester University, UK]); functional-blocking anti-$\alpha_v\beta_5$-integrin (*Clark et al., 2005*) (60 μg/ml ALULA [a gift from Drs Dean Sheppard (UCSF, San Francisco, CA),

and Shelia Violette (Biogen, Cambridge, MA]); or species-matched non-immunized isotypic antibodies.

## RT-qPCR

The Ambion PureLink kit (Life Technologies) was used, according to manufacturer's instructions, to extract total cellular RNA from the various experimental conditions and tested for RNA quality using a Bioanalyzer (Agilent). Contaminating genomic DNA was removed using Turbo DNA free from Ambion. RNA concentrations were determined with a spectrophotometer (NanoDrop; Thermo Fisher Scientific). RNA was reverse transcribed (RT) using Moloney murine leukemia virus reverse transcriptase (Ambion) and a mixture of anchored oligo-dT and random decamers (IDT). Two reverse-transcription reactions were performed for each experimental duplicate sample using 100 ng and 25 ng of input RNA. Taqman assays were used in combination with Taqman Universal Master Mix and run on a 7900 HT sequence detection system (Applied Biosystems). Cycling conditions were 95°C, 15 min, followed by 40 (two-step) cycles (95°C, 15 s; 60°C, 60 s). Ct (cycle threshold) values were converted to quantities (in arbitrary units) using a standard curve (four points, four-fold dilutions) established with a calibrator sample. Values were averaged per sample and standard deviations were from a minimum of two independent PCRs. Polymerase (RNA) II (DNA directed) polypeptide F (POLR2F) was used as internal control. Identification numbers of commercial assays (Life Technologies) and sequences of the primers and probe for the POLR2F assay are: Hs00363100_m1 (PALLD1), Hs00426835_g1 (ACTA2), TGCCATGAAGGAACTCAAGG (forward), TCATAGCTCCCATCTGGCAG (reverse), and 6fam-CCCCATCATCATTCGCCGTTACC-bqh1 (probe).

## D-ECM prompted naïve-to-myofibroblastic activation

Human naïve or assorted murine fibroblasts were pre-incubated for 45 min with functional antibodies, small molecule inhibitors or controls, prior to overnight culturing within assorted ECMs. Experiments were carried out in the presence or absence of 45 µg/ml SNAKA51 (*Clark et al., 2005*), 60 µg/ml ALULA (*Su et al., 2007*), 250 µg/m mAb16 (*Akiyama et al., 1989*) (a gift from Kenneth Yamada [NIDCR/NIH, Bethesda, MD]), BMA5 (*Fehlner-Gardiner et al., 1996*) diluted as instructed by the manufacturer (Millipore, Temecula, CA), IgG control, DMSO, or small molecule inhibitor of FAK (PF573,228 [*Slack-Davis et al., 2007*]). ECM-induced phenotypes were assessed by RT-qPCR, western blot or immunofluorescence as described in the corresponding Methods sections.

## Array to detect constitutive plasma-membrane-localized levels of assorted β-integrin heterodimers

An integrin-mediated cell adhesion assay was conducted to detect cell surface levels of heterodimeric integrin receptors on the assorted fibroblastic cells. The array, ECM531 from Millipore, is based on the use of selected monoclonal antibodies against assorted $\beta$-integrin heterodimers and testing amounts of cell adhesion to each particular antibody. Cells are incubated onto an array of wells coated with the various antibodies and then washed, before adherent cells are counted. The number of cells adhering to a specific antibody is directly related to the amount of the specific integrin receptor expressed at the cell surface. Numbers of adherent cells' surfaces are evaluated by 540–570 nm absorbance of cells stained using a staining reagent provided with the kit according to the manufacturer'sinstructions.

## TGFβ ELISA

Quantification of TGF$\beta$ stored/deposited into unextracted fibroblast 3D cultures was performed using the DuoSet human TGF$\beta$1 ELISA kit, as instructed by the manufacturer (R and D Systems, #DY240-05). All wash buffers and reagents were provided in the DuoSet Ancillary Reagent Kit 1 (R and D systems, #DY007). Briefly, whole cell lysates (or conditioned media as controls) were obtained from fibroblasts and loaded onto plates that were pre-coated with capture antibody overnight at RT. After 2 hr incubation of lysates, concentrated media and calibration curve standards, wells were rinsed three times with wash buffer and then incubated for 2 hr at RT with detection antibody. Next, after three more washes, streptavidin-HRP was incubated in the wells for 20 min at RT. After another three washes, substrate solution was added to the wells, and after 15 min, stop solution was added to halt the HRP-substrate reaction. Next, OD at 450 nm and 570 nm was obtained using the Synergy

H1 plate reader (BioTek). The OD at 570 nm was subtracted from the OD at 450 nm to correct for plate imperfections. Then the standard curve was constructed, and the concentration of TGF$\beta$ (pg/mL) in experimental samples was determined. Last, TGF$\beta$ concentration was normalized to the amount of protein loaded into each well and then normalized to control TAF levels, with the modal value referred to as one a.u. (arbitrary unit).

## Electron microscopy

Au$_8$-SNAKA51 and Au$_{15}$-mAb11 gold-antibody complexes were conjugated as described in *Handley et al. (1981)*. Cells from functional assays were fixed with 2.0% paraformaldehyde in PBS, pH 7.4, at 4°C overnight. Following antibody incubation, cells were fixed with 2.0% glutaraldehyde washed and post-fixed in 2.0% osmium tetroxide for 1 hr at room temperature. Dehydration through a graded ethanol series and propylene oxide was followed by embedding samples in EMbed-812 (Electron Microscopy Sciences [Hatfield, PA]). Sections were stained with 2% alcoholic uranyl acetate and scanned with a FEI Tecnai electron microscope, while images were acquired using a Hamamatsu digital camera operated via AMT Advantage image capture software (AMT 542.544; 18 Sep 2009).

## Indirect immunofluorescence (*in vitro*)

The method for indirect immunofluorescence was as published previously (*Gupta et al., 2011*; *Amatangelo et al., 2005*; *Franco-Barraza et al., 2016*; *Cukierman et al., 2001*), only Triton was omitted for non-permeable conditions. Samples were either fixed/permeated or solely fixed as published (*Franco-Barraza et al., 2016*). Following 60 min blocking, using Odyssey Blocking Buffer (LI-COR Biosciences, Lincoln, NE) containing 1% donkey serum, the following primary antibodies were incubated in combinations as indicated by the specific experiments for a period of 60 min: anti-pan-cytokeratin (clones: AE1/AE3) (AB_2631307) from Dako (Carpinteria, CA), rabbit monoclonal anti-Vimentin EPR3776 (AB_10004971) (1:200) from Abcam (Cambridge, MA), rabbit polyclonal anti-human fibronectin (AB_476961) (1:200) and 1 A4 mouse anti-αSMA (AB_476701) (1:300) were from Sigma-Aldrich (St. Louis, MO), mouse SNAKA51 Ab (45 µg/ml) was a gift from M Humphries (*Clark et al., 2005*), rat mAb11 (45 µg/ml) from K Yamada (*Akiyama et al., 1989*), rabbit polyclonal anti-palladin (AB_2158782) (1:200) (Proteintech, Chicago, IL), rabbit monoclonal anti-phospho FAK [Y397] (AB_2533701) (1:200) (Invitrogen, Camarillo, CA), rabbit monoclonal anti-phospho Smad2 [S465/467]/Smad3 [S423/425] (AB_2631089) (1:200) (Cell Signaling, Danvers, MA), rabbit monoclonal anti-Rab5 (AB_10828212) (1:150) (Cell Signaling, Danvers, MA), rabbit monoclonal anti-Rab7 (AB_10831367) (1:25) (Cell Signaling, Danvers, MA), rabbit monoclonal anti-Rab11 (AB_10693925) (1:25) (Cell Signaling, Danvers, MA), rabbit monoclonal anti-EEA1 (1:150) (AB_10828484) (Cell Signaling, Danvers, MA), rabbit monoclonal anti-CD81 (AB_2275892) (1:35) (Santa Cruz Biotechnologies, Dallas, Texas) and species-matched non-immunogenic mouse IgG (Abcam) and Rat IgG (Jackson ImmunoResearch Inc., West Grove, PA) were used at equal concentrations to their corresponding matching isotypes. Following three rinses of 5 min with PBS-Tween20 (0.05%), secondary antibodies were incubated for 60 min using donkey F(ab')two fragments (1:100) cross-linked to assorted fluorophores (Jackson ImmunoResearch Laboratories Inc.). When mentioned, SYBR Green (1:10,000 Invitrogen [Eugene, OR]) or Draq-5 (1:10,000 Pierce Biotechnology [Rockford, lL]) and/or fluorescent Phalloidin (2.5 µl/100 µl, Invitrogen [Eugene, OR]) were included. Following washes with PBS and a final rinse with double-distilled water, samples were mounted using Prolong Gold anti-fading reagent from Life Technologies (Carlsbad, CA). When needed, anti-αSMA and/or SNAKA51 primary antibodies were pre-linked to fluorescent dye crystals using the Mix-n-Stain$^{TM}$ kit following the manufacturer's recommendations (Biotium, Hayward, CA).

## Confocal image acquisition

Confocal spinning disk Ultraview (Perkin-Elmer Life Sciences, Boston, MA) images were acquired with a 60X (1.45 PlanApo TIRF) oil immersion objective, under identical exposure conditions per channel using Volocity 6.3.0 (SCR_002668) (Perkin-Elmer Life Sciences, Boston, MA). Maximum reconstruction projections were obtained using MetaMorph 7.8.1.0 (SCR_002368) (Molecular Devices, Downingtown, PA) as described previously (*Amatangelo et al., 2005*). For monochromatic *in vitro* image analyses using SMIA-CUKIE, please refer to SMI methods below.

## ECM fiber orientation analysis

Fibronectin channel monochromatic images were analyzed via ImageJ's (SCR_003070) OrientationJ plugin (SCR_014796) (http://bigwww.epfl.ch/demo/orientation/) as described previously (*Franco-Barraza et al., 2016*; *Rezakhaniha et al., 2012*). Numerical outputs were normalized by setting mode angles to 0°, and correcting angle spreads to fluctuate between −90° and 90°. Angle spreads for each experimental condition, corresponding to a minimum of three experimental repetitions and five image acquisitions per condition were plotted and their standard deviations calculated using Excel spreadsheets. Percentage of fibers oriented between −15° and 15° was determined for each normalized image-obtained data. See statistics section below.

## Digital imaging analyses (*in vitro*)

16 bit TIFF files corresponding to the maximum projections of assorted monochromatic reconstituted z-stacks were analyzed using MetaMorh Offline software. On the basis of a noise/signal ratio, gray-scale levels of reference images (control) were optimized to reduce background to a minimum while avoiding signal saturation (at the maximum point). An inclusive threshold for positive signal was set for control images. These levels and threshold values were used throughout the entire experiment and served as normalizing controls for each experiment.

Intensity levels of αSMA or active α5-integrin were determined by threshold-masked images, evaluating total pixel intensities and integrated optical densities, using corresponding functions from the software's integrated morphometry analysis (IMA) algorithm. The percentage of αSMA colocalization at F-actin stress fiber structures was calculated by querying αSMA-positive area coverages that overlapped (i.e., co-localized) with phalloidin-positive stress fibers areas.

For experiments questioning levels in murine cells, an 'activated fibroblastic phenotype' was determined by counting the percentage of cells presenting stress-fiber-localized αSMA. Output data from each analysis were normalized to corresponding control conditions and expressed as fold variations related to control.

For 3D-adhesion length calculations: calibrated images (one pixel = 0.11 µm) were used while their gray-scale levels were optimized to reveal 3D-adhesion (i.e., mAb11) positive-stained structures. These were recognized as 'objects' by the software via threshold settings (identical for all images) and measured through the 'dimension' function of IMA. Objects with lengths ≥6.5 µm, signifying *bona fide* 3D-adhesion structures (*Cukierman et al., 2001*), were included in the analysis.

The more intricate analyses of multi-labeled 3D-adhesions were performed using the 'line-scan' function of the software. Channel-overlaid 8bit images were subjected to a transversal analysis (12 µm length) to identify peaks of median gray levels of the stated particular markers along the line-scanned structure being analyzed.

*In vitro* SMIA-CUKIE (see '**SMI approach**' below for additional information) was used to query active $\alpha_5$-integrin localization (i.e., at or away from 3D-adhesions *in vitro* and where stroma-positive and tumoral-negative locations were identified *in vivo*). Areas corresponding to 3D-adhesion-positive pixels (i.e., mAb11) were assigned as *masks* areas, while SNAKA-51-positive pixels were designated active $\alpha_5$-integrin, pFAK$^{Y397}$ as PFAK, and pSmad2/3 S$^{465,467}$/S$^{423,467}$ as PSMAD or *marker* positive areas. *Marker* values corresponding to median, mean, standard deviation, total intensity, integrated intensity and area coverages relative to the *mask,* or relative to the image at *mask,* or away from *mask* locations were obtained as Excel spreadsheet outputs. See below for *in vivo* SMIA-CUKIE analyses using surgical FFPE samples.

## Simultaneous multi-channel immunofluorescent (SMI) approach and SMIA-CUKIE software-based analyses

### Conjugation of Q-dot to primary antibodies

Q-dot antibody labeling kits were from Molecular Probes-ThermoFisher Scientific. SiteClick labeling constitutes a three-step procedure lasting a few days and requires that the desired antibody's carbohydrate domain is modified to allow an azide molecule to be attached to it; In consequence, DIBO-modified nanocrystals are linked to the antibody in question. Each conjugation was carried out strictly as instructed by the protocols provided by the manufacturer. Briefly, approximately 100 µg of antibody was used in each case. Unless there were excessive carrier proteins present in the antibody, there was no need for an antibody pre-purification step. The labeled antibodies were

estimated to be at 1 µmolar concentration and stored in sterile conditions at 4°C in light-protected boxes. Sample incubations were all done at a final concentration of ~20 nM or 1:50 dilution (see below).

## Immunolabeling of FFPE tissue sections

Slides were exposed to a short-wave UV lamp for about 30 min in a light-protected box to quench auto-fluorescence. They were then kept in a light-protected (i.e., dark) box until used. Sections were deparaffinized in xylene and rehydrated in ascending graded alcohol to water dilutions. Sections were then treated with Digest-All (Invitrogen) and permeabilized in 0.5% TritonX-100. After treating them with blocking buffer as in *in vitro* work (see above), samples were first incubated with the above-mentioned Q-dot pre-labeled antibodies (i.e., Q655 SNAKA51, Q565 mAb11) overnight at 4°C. Sections were washed as in *in vitro* work, and incubated with a mix of mouse monoclonal 'cock-tail' containing anti-pan-cytokeratin (clones AE1/AE3, DAKO), anti-EpCam (MOC-31EpCam, Novus Biologicals (Littleton, CO)), and anti-CD-70 (113–16 Biolegend (San Diego, CA) [*Ryan et al., 2010*]) to detect epithelial/tumoral locations, together with rabbit monoclonal anti-vimentin (EPR3776, Abcam) antibodies for mesenchymal (stromal) components for 2 hr at RT. Note that pre-incubated Q-dot labeled antibodies could no longer be recognized by secondary antibodies, thus allowing for indirect immunofluorescent detection of tumoral and stromal masks. Secondary antibodies were as used as in *in vitro* work and included donkey anti-mouse Cy2 and donkey anti-rabbit Cy3. Nuclei were stain using Draq-5 (as *in vitro*). Sections were quickly dehydrated in graded alcohol and clari-fied in Toluene before mounting in Cytoseal-60. Slides were cured overnight at RT before the imaging.

## Image acquisitions of FFPE fluorescently labeled sections

FFPE sections of biological samples are known to give strong broad autofluorescence, thus obscuring the specific (labeled) fluorescent signal. To overcome this problem, images were collected using Caliper's multispectral imaging system (PerkinElmer), which utilizes a unique imaging module with Tunable Liquid Crystal. Two different systems namely Nuance-FX (for 40x objective) and Vectra (for 20x objective and high throughput) were used depending on the acquisition needs. A wavelength-based spectral library for each system and each tissue type (i.e., pancreas and kidney) was created by staining control sections with individual fluorophores or by mock treating samples to include the specific autofluorescence spectra. Once a spectral library was constructed for each organ type, it was saved and used for the subsequent image acquisition and analysis. All excitations were achieved via a high-intensity mercury lamp using the following filters (emission-excitation): for Nuance, DAPI (450-720), FITC (500-720), TRITC (580-720), CY5 (680-720); for Vectra, DAPI (440-680), FITC (500-680), TRITC (570-690), CY5 (680-720). For emissions collection, 'DAPI' filter (wavelength range 450–720) was used for all Q-dot labeled markers, while masks used the conventional FITC, TRITC, and CY5 filters. After collecting all image cubes (including all channels), images were unmixed to obtain 16-bit greyscale individual stains monochromatic files. Using Photoshop's 'Levels' and 'Batch Conversion' functions, the images were processed in bulk to render identically scaled 8-bit monochromatic images for each channel. The resulting images were sampled to set identical threshold values for each channel, which were used to feed the values needed for analyses in SMIA-CUKIE to signify positive-labeled pixels.

## SMIA-CUKIE usage and outputs

SMIA-CUKIE (SCR_014795) was written for the bulk analysis of high-throughput-acquired monochromatic images, corresponding to simultaneously labeled channels like the ones used in this study. As examples, masks in *Figure 12* and *Figure 12—figure supplement 2* were generated using SMIA-CUKIE and demonstrate how the software isolates stromal locations while omitting positive tumoral areas, based on mask value thresholds that are provided by the user.

Images were sorted into 'Batch Folders', each containing the five monochromatic images corresponding to the original (unmixed) sample. The newly written software, available at https://github.com/cukie/SMIA_CUKIE, was created to bulk process and analyze batches of monochromatic images providing localization (masks), intensities, and similar quantifying values (markers), including co-localizations of multichannel monochromatic immunofluorescent (or IHC, etc.) images. The software can

identify intersection areas between an unlimited amount of masks, while queried marker values and locations can also be estimated for numerous interrogations. The software requires the identification of common nomenclatures to recognize each type of monochromatic image (i.e vim for vimentin etc.). It also needs information regarding the available number of masks and markers deposited in the batch folders containing the images. In this work, three masks were used corresponding to nuclei, epithelium/tumor and stroma, as well as two markers corresponding to adhesion structures and active $\alpha_5\beta_1$ integrin. For each of the masks and markers, the software requires a numeric threshold (0-255), which indicates the value of pixel intensity that is to be considered positive for each channel (corresponding to each mask and marker). The software then allows choosing amongst all possible query combinations. Alternatively, it provides an option for the user to write the desired tests to be queried. For example, when active $\alpha_5\beta_1$ integrin values were requested at *bona fide* stromal locations, the software was instructed to look for 'SNAKA under vimentin, NOTepi/tumor'. In this example, 'SNAKA' was the nomenclature we used for the active integrin channel while vimentin and epi/tumor served as masks. After running this function, the software rendered an Excel file containing values of area coverage for the marker at the requested mask intersection (stroma), as well as total area coverage related to the image. Similarly, values included mean, median, standard deviation, total and integrated intensities. In addition, the software can be instructed to provide image outputs corresponding to requested mask locations, as well as markers shown solely at the corresponding mask intersections (as shown in *Figure 12* and *Figure 12 –figure supplement 2*). Data values obtained in this work are available online at https://www.foxchase.org/sites/fccc/files/assets/cukierman_Franco-Barraza%20SMIA-CUKIE-Dec-2016.xlsx. Finally, values were used for statistical analyses as described in the corresponding section.

## CRISPR/CAS9-mediated knockout of $\beta_5$ and other integrins in fibroblasts

### Immortalization of human fibroblasts

Before performing CRISPR/Cas9-mediated KO of $\beta5$ integrin, fibroblasts were immortalized with human telomerase (hTERT) (*Counter et al., 1998*), using a retroviral infection with the pBABE-neo-hTERT vector, which was a gift from Robert Weinberg (Addgene plasmid # 1774). First, in order to produce functional retrovirus, packaging cells Phoenix-Amphotropic (φNX) (ATCC # CRL-3213) at 50% confluence in 10 cm culture dishes were transfected with 10 ug pBABE-neo-hTERT vector using 10 uL Lipofectamine (Invitrogen, #18324) and 40 uL of Plus Reagent (Invitrogen, # 11514015) in 6 mL serum/antibiotics free Opti-MEM (Gibco, # 31985062) culture media overnight at 37°C in a cell culture incubator. The following morning, the medium was replaced with 10 mL fresh Opti-MEM (serum/antibiotics free) and incubated 24 hr at 32°C. In the morning of days 2, 3 and 4 post-transfection, the conditioned medium containing retrovirus was collected and filtered through a 0.45 um syringe filter (Millipore, #SLHV013SL) and used for subsequent retroviral transfections of fibroblasts. For the viral infection, fibroblasts were cultured in 10 mL of conditioned medium containing the retrovirus (supplemented with 4 ug/mL Polybrene (Santa Cruz, #sc-134220)) for 8 hr at 37°C. Then cells were washed and incubated with fresh culture medium (DMEM 10% FBS, 1% L-Glut, 1% P/S) and incubated at 37°C overnight. Beginning the next day, this infection procedure was repeated two additional times, for a total of three retroviral infections. Fibroblasts were then selected with G-418 at a concentration of 750 µg/mL until cells grew back to ~90% confluence. Cells were then expanded and tested for overexpression of hTERT, and lack of p16, by western blotting to confirm hTERT overexpression. These cells were used for CRIPSR/CAS9mediated knockout of $\beta5$ integrin.

### gRNA design

To knockout the $\beta5$-integrin-subunit-encoding gene from fibroblasts, CRISPR/CAS9 gene editing was performed to introduce a frameshift mutation that disrupts the reading frame causing a premature stop codon to be read, ultimately halting translation and knocking out this gene.

First, gRNAs specific for $\beta5$ integrin were designed by targeting exon 3. The first 200 base pairs of exon three were inserted into MIT Optimized CRISPR Design website: http://crispr.mit.edu/ and the top two scoring gRNAs were selected, based on the presence of a PAM site (a CAS9 recognition site) and limited off-target binding.

The two independent gRNA sequences were as follows:

gRNA1: ACCGAGAGGTGATGGACCGT
gRNA 2: CACCGAGAGGTGATGGACCG

For a non-targeting gRNA control, the following gRNA against eGFP was designed: CATGTGA TCGCGCTTCTCGT.

## Generation of lentiviral KO vector

Once the gRNA sequences were designed and ordered (Integrated DNA Technologies), they were cloned into the LentiCRISPR v2 vector (LentiCRISPR v2 was a gift from Feng Zhang; Addgene plasmid # 52961) (*Sanjana et al., 2014*). This is a dual-expression vector, expressing the CRISPR/CAS9 protein as well as the cloned gRNA sequence driven by the human U6 promoter. Briefly, DNA oligos representing the gRNA sequences are listed below, with overhangs compatible with the Esp3I restriction enzyme (**bold, underlined)** and an additional G added to the beginning of each gRNA, with a complementary C on the reverse oligo (***bold, italics***), for efficient transcription driven by the human U6 promoter.

| | |
|---|---|
| Integrin $\beta_5$ gRNA 1.1 | **CACCG**ACCGAGAGGTGATGGACCGT |
| Integrin $\beta_5$ gRNA 1.2 | **AAAC**ACGGTCCATCACCTCTCGGTC |
| Integrin $\beta_5$ gRNA 2.1 | **CACCG**CACCGAGAGGTGATGGACCG |
| Integrin $\beta_5$ gRNA 2.2 | **AAAC**CGGTCCATCACCTCTCGGTGC |
| eGFP gRNA 1.1 | **CACCG**CATGTGATCGCGCTTCTCGT |
| eGFP gRNA 1.2 | **AAAC**ACGAGAAGCGCGATCACATGC |
| Integrin $\alpha_5$ gRNA 1.1 | **CACCG**CTCAGTGGAGTTTTACCGGC |
| Integrin $\alpha_5$ gRNA 1.2 | **AAAC**GCCGGTAAAACTCCACTGAGC |
| Integrin $\alpha_5$ gRNA 2.1 | **CACCG**TCAGTGGAGTTTTACCGGCC |
| Integrin $\alpha_5$ gRNA 2.2 | **AAAC**GGCCGGTAAAACTCCACTGAC |
| Integrin $\alpha_v$ gRNA 1.1 | **CACCG**ATTCAATTGGCTGGCACCGGCGG |
| Integrin $\alpha_v$ gRNA 1.2 | **AAAC**CCGCCGGTGCCAGCCAATTGAATC |
| Integrin $\alpha_v$ gRNA 2.1 | **CACCG**TGACTGGTCTTCTACCCGCCGG |
| Integrin $\alpha_v$ gRNA 2.2 | **AAAC**CCGGCGGGTAGAAGACCAGTCACC |
| Integrin $\beta_3$ gRNA 1.1 | **CACC**GCCCAACATCTGTACCACGCG |
| Integrin $\beta_3$ gRNA 1.2 | **AAAC**CGCGTGGTACAGATGTTGGGC |
| Integrin $\beta_3$ gRNA 2.1 | **CACC**GACCTCGCGTGGTACAGATGT |
| Integrin $\beta_3$ gRNA 2.2 | **AAAC**ACATCTGTACCACGCGAGGTC |

Next, the gRNA oligos stocks were diluted to 100 uM, and 1 uL of each gRNA pair was added to a T4 PNK reaction mixture (NEB, #M0201S) for a 10 uL reaction. The reaction was allowed to run in a thermal cycler to phosphorylate and anneal the oligos, according to the following program:

1. 37°C for 30 min
2. 95°C for 5 min
3. Decrease 5°C every minute until 25°C

To prepare the vector for cloning, 5 ug of LentiCrispr v2 vector were simultaneously cut with Fast Digest Esp3I (ThermoFisher, #FD0454) and dephosphorylated with Fast AP (ThermoFisher, #EF0651) for 30 min at 37°C and subsequently run on a 1.5% agarose gel and purified for cloning using the GeneJet Gel Extraction Kit (Thermofisher, #K0691).

To clone the annealed gRNA oligos into the vector, the oligos were diluted 1:200 in RNAase-free water (ThermoFisher, #4387937), and added to the quick ligation reaction mixture (NEB, #M2200S), along with 1 μL of the digested and dephosphorylated vector, and the reaction was allowed to proceed for 10 min at room temperature.

For bacterial transformation, 2 μL of the reaction was mixed with 50 μL of competent Stbl3 strain of *Escherichia coli* for 30 min on ice. The bacteria were then heat shocked for 45 s at 42°C and

immediately transferred to ice for 2 min. Then, 50 μL of the transformed bacteria were spread onto an LB/agar dish containing 100 μg/mL ampicillin and incubated at 37°C overnight.

The next day, single colonies were screened by colony PCR using the U6 promoter forward primer: GAGGGCCTATTTCCCATGATT and the corresponding reverse gRNA primers (gRNA x.2). Positive clones were selected to expand for plasmid purification and sequencing.

## CRISPR lentivirus production

For functional lentiviral production, viruses were generated in 293T cells using the cloned LentiCrispr v2 plasmids, and two packaging plasmids: psPAX2 (psPAX2 was a gift from Didier Trono; Addgene plasmid # 12260) and VSVg. Briefly, 10 μg of cloned LentiCrispr V2, 5 μg of psPAX2, and 2 μg of VSVg were mixed in 1 mL serum-free/antibiotic-free DMEM in an 1.5 mL Eppendorf tube. 30 μL of X-treme Gene 9 (Roche, #06365787001) was added to the DNA mixture and gently mixed with the pipette tip. The mixture was allowed to sit for 45 min at room temperature. Next, the mixture was added drop-wise to a T75 flask containing 5 mL serum-free/antibiotic-free DMEM and 293T cells at ~85% confluence and slowly rocked for 30 s to evenly mix the DNA transfection mixture. The next morning, the serum-free media was removed from the 293T cell flasks, and 10 mL of fresh DMEM containing 10% FBS and 1% penicillin/streptomycin were added to each flask. Two days and 4 days post-transfection, the medium was collected and filtered through a 0.45 μM syringe filter (Millipore, #SLHV013SL) and then used immediately for viral infection of target cells, or stored at −80°C until needed.

## CRISPR lentiviral infection

Target cells (naïve or desmoplastic fibroblasts) were seeded at ~40% confluence in 2 mL complete fibroblast media in a six-well plate. The following day, the target cells were infected with 2 mL of lentivirus for each corresponding CRISPR construct (eGFP or integrin gRNAs) in the presence of 10 ug/mL Polybrene (Santa Cruz, #sc-134220). As a control for the infection, cells were infected with a lentivirus overexpressing GFP, and the appearance of GFP-positive cells signified a successful infection. 24 hr later, the medium was replenished with fresh complete medium for each cell type. After 72 hr, puromycin selection (1 ug/mL for naïve fibroblasts, 2 ug/mL for desmoplastic fibroblasts) of the infected cells began. The selection process lasted between 7 and 10 days; cells were expanded, and the efficiency of CRISPR/CAS9 knockout was assessed by western blotting. The cell lines with the greatest degree of target protein knockout were used for subsequent experiments.

## SDS-PAGE/western blotting

Briefly, various normal fibroblasts and CAFs were grown to confluency in six-well plates and were lysed in standard RIPA buffer. Lysates were then homogenized by sonication, allowed to rest on ice for 15 min, and centrifuged for 10 min at maximum speed in a microcentrifuge. Next, samples were diluted in 2x loading buffer (Bio-Rad, Hercules, CA), boiled for 3 min, and loaded onto 4–20% gradient gels (Bio-rad, Hercules, CA) and run at 70 volts for 2 hr. Next, gels were subjected to semi-dry transfer to PVDF membranes (Millipore, Billerica, MA) and blocked for 1 hr at RT in 5% milk in 0.1% TBST. Membranes were then incubated in the one of the following primary antibodies overnight at 4°C: rabbit polyclonal anti-$\alpha_v$-integrin (AB_2631308) (1:5000) (Mybiosource, San Diego, CA), rabbit monoclonal anti-$\alpha_5$-integrin (AB_2631309) (1:1000) (Abcam, Cambridge, MA), rabbit polyclonal anti-$\beta_5$-integrin (AB_10806204) (1:1000) () (Millipore, Billerica, MA) or rabbit polyclonal anti-$\beta_3$-integrin (1:1000) (AB_91119) (Millipore, Billerica, MA). The following day, membranes were washed in 0.1% TBST, 5 times for 5 min each. Next, membranes were incubated for 2 hr in anti-rabbit IgG-HRP secondary antibodies (Sigma, St. Louis, MO) in 5% milk in 0.1% TBST. Blots were then washed again five times/5 min each. Finally, Immobilon Western Chemiluminescent HRP substrate (Millipore, Billerica, MA) was added to the blots and protein bands were captured by film or Protein Simple Digital Imaging System (Protein Simple, San Jose, CA).

## Bioinformatics and statistics

Two-tailed non-parametric Mann-Whitney tests were used to query experimental significances for all *in vitro* data, comparing control conditions to experimental ones independently. All *in vitro* results were presented as medians with interquartile range. Each PDAC-related *in vitro* experiment was

performed at least three times, using one patient matched CAF and normal fibroblasts. For verification, these series of experiments were also conducted, using one additional pair of normal and activated fibroblasts from a non-related patient, two normal fibroblasts isolated from two additional non-related patients, and one tumor tissue which provided one additional non-related activated (desmoplastic) fibroblastic cell. All cells characterized as normal fibroblasts were verified for their phenotype and challenged with D-ECMs produced by CAFs isolated from three of the above describedpatients. These included a total of five different patient sources which included seven surgical samples. The RCC-related *in vitro* assessments were performed employing a combination of two patients-derived CAFs and normal fibroblasts from another two different sources. All the image analyses were conducted in at least ten images containing no less than three cells per region per experimental condition for each above-mentioned repetition.

Regarding statistics pertinent to the TMAs, to identify clinical and laboratory variables related to patient survival (OS, DSS, and RFS), univariate and multivariable analyses were performed by constructing decision trees using the CART methodology. Clinical and laboratory variables were considered as predictors of survival time and are listed in data uploaded to https://www.foxchase.org/sites/fccc/files/assets/cukierman_Franco-Barraza%20SMIA-CUKIE-Dec-2016.xlsx. The unified CART framework that embeds recursive binary partitioning into the theory of permutation tests was used. Significance-testing procedures were applied to determine whether no significant association between any of the clinical variables and the response could be stated, or whether the recursion would need to stop. Also, log-rank tests and univariate and multivariable Cox proportional hazards (PH) were used to correlate clinical and laboratory variables with survival; hazard ratios (with 95% confidence intervals) were calculated, as appropriate, for various comparisons. Goodness-of-fit of the PH model was evaluated based on Schoenfeld residuals. In order to account for potential batch effects, a 'batch' variable was included as an adjustment variable in all the analyses that were pertinent to the PDAC cohort. All tests were two-sided and used a significance level of 5% to test each hypothesis. Significant p-values in all figures were denoted by asterisks as follows: ****p<0.0001 extremely significant, ***p=0.0001–0.01 very significant, **p=0.01–0.05 significant and *p=0.05–0.10 marginally significant. Statistical analyses were performed using GraphPad Prism software, version 6.05 for Windows (La Jolla, CA) and the R packages *survival* and *parity* (www.r-project.org).

## Acknowledgements

We thank Wafik S El-Deiry for providing critiques on the written manuscript and A Carson for proofreading. We also thank K Yamada (NIH), S Violette (Biogene Inc), D Sheppard (UCSF), M Humphries (UM-UK), and D Schlaepfer (UCSD) for providing reagents.

## Additional information

### Funding

| Funder | Grant reference number | Author |
| --- | --- | --- |
| National Cancer Institute | R01 CA113451 | Janusz Franco-Barraza<br>Ralph Francescone<br>Tiffany Luong<br>Neelima Shah<br>Raj Madhani<br>Robert G Uzzo<br>John P Hoffman<br>Edna Cukierman |
| National Cancer Institute | Core Grant CA06927 | Janusz Franco-Barraza<br>Ralph Francescone<br>Tiffany Luong<br>Neelima Shah<br>Raj Madhani<br>Gil Cukierman<br>Essel Dulaimi<br>Karthik Devarajan<br>Brian L Egleston<br>Emmanuelle Nicolas<br>R Katherine Alpaugh |

| Funder | Grant reference number | Author |
| --- | --- | --- |
| | | Ruchi Malik<br>Robert G Uzzo<br>John P Hoffman<br>Erica A Golemis<br>Edna Cukierman |
| The Commonwealth of Pennsylvania | | Janusz Franco-Barraza<br>Ralph Francescone<br>Tiffany Luong<br>Neelima Shah<br>Raj Madhani<br>Gil Cukierman<br>Essel Dulaimi<br>Karthik Devarajan<br>Brian L Egleston<br>Emmanuelle Nicolas<br>R Katherine Alpaugh<br>Ruchi Malik<br>Robert G Uzzo<br>John P Hoffman<br>Erica A Golemis<br>Edna Cukierman |
| U.S. Department of Defense | Idea Award with Special Focus in Military Personnel-Affected Pancreatic Cancer W81XH-15-1-0170 | Ralph Francescone<br>Tiffany Luong<br>Neelima Shah<br>Karthik Devarajan<br>Edna Cukierman |
| National Cancer Institute | Ruth L. Kirschstein National Research Service Award T32 training grant CA009035 | Ralph Francescone |
| Fox Chase Cancer Center | Nodal Grant | Ruchi Malik<br>Edna Cukierman |
| Marvin S. Greenberg Fund in Support of Pancreatic Cancer Research | | John P Hoffman<br>Edna Cukierman |
| National Institutes of Health | CA191425 | Erica A Golemis |
| Bucks County Board of Associates | | Edna Cukierman |

The funders had no role in study design, data collection and interpretation, or the decision to submit the work for publication.

## Author contributions

JF-B, Data curation, Formal analysis, Supervision, Validation, Investigation, Visualization, Methodology, Writing—original draft, Writing—review and editing; RF, Resources, Data curation, Formal analysis, Validation, Investigation, Methodology, Writing—review and editing; TL, RMal, Data curation, Formal analysis, Validation, Methodology; NS, Validation, Investigation, Visualization, Methodology; RMad, Data curation, Validation, Investigation, Methodology; GC, Software, Visualization, Methodology; ED, Resources, Validation; KD, Data curation, Software, Formal analysis, Validation, Visualization, Methodology; BLE, Data curation, Software, Validation, Visualization, Methodology; EN, Formal analysis, Methodology; RKA, Resources, Methodology; RGU, JPH, Resources, Writing—original draft; EAG, Conceptualization, Visualization, Writing—review and editing; EC, Conceptualization, Data curation, Software, Formal analysis, Supervision, Funding acquisition, Visualization, Writing—original draft, Project administration, Writing—review and editing

## Author ORCIDs

Janusz Franco-Barraza, http://orcid.org/0000-0003-3652-5311
Edna Cukierman, http://orcid.org/0000-0002-1452-9576

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
