## [Decision Letter]

Thank you for submitting your article "Matrix-regulated integrin αvβ5 maintains α5β1-dependent desmoplastic traits prognostic of neoplastic recurrence" for consideration by *eLife*. Your article has been favorably evaluated by Harry Dietz (Senior Editor) and three reviewers, one of whom, Johanna Ivaska, is a member of our Board of Reviewing Editors. The reviewers have opted to remain anonymous.

The reviewers have discussed the reviews with one another and the Reviewing Editor has drafted this decision to help you prepare a revised submission.

Summary:

A timely and critical question in cancer biology concerns the dual role of the fibroblasts in restraining disease in some instances (i.e. normal fibroblasts) while promoting disease progression in other instances (i.e. activate carcinoma-associated fibroblasts). In particular, recent studies evaluating pancreatic cancer have highlighted this issue by further confirming that normal fibroblasts can restrain disease; findings that have been reasonably well described in other solid tumors. Further, strong consensus evidence in pancreatic cancer and numerous other solid tumors indicates a disease promoting role for activated myofibroblasts in the stroma. In the current work the authors undertake focused studies to evaluate this dual role of fibroblasts and elucidate fundamental mechanisms driving these distinct behaviors. Franco-Barraza and colleagues have investigated the underlying mechanisms of PDAC desmoplasia by generating 3D in vitro cell derived matrixes from normal fibroblasts (to generate N-ECM) and PDAC CAFs to generate D-ECM (desmoplastic ECM). They find that TGFb signalling is essential for D-ECM production but dispensable for the ability of the D-ECM to covert naïve fibroblasts to activated myofibroblasts. They find that in response to PDAC-associated D-ECM, αvβ5-integrin re-distributes active α5β1-integrin away from pFAK-Y397 positive 3D-adhesions and, thus, promotes myofibroblastic activation whereas inhibition of avb5 under these conditions was protective against myofibroblasts activation by increasing active a5b1 location to the cell surface. They also have data from human PDAC and RCC investigating whether levels of FAK and α5β1 integrin can distinguish tumor suppressive v tumor promoting stroma.

Essential revisions:

The reviewers found your work to be particularly relevant to the field right now as there is some controversy about the role of fibroblasts in disease progression and response to therapy. They felt that the data presented in the submitted manuscript are very timely and may help clarify the dynamics of cancer stroma crosstalk in pancreatic carcinoma. However, all of the reviewers found the manuscript to be extremely dense and especially the Results section very hard to follow. Their enthusiasm was somewhat hampered by the lack of clarity in several instances (please see specific points below) that make portions of the manuscript difficult to read/interpret. Furthermore, they were critical about the clinical relevance and strength of the human data and the representability of the limited number of fibroblasts for PDAC as a whole. In addition, to the specific points raised below the authors are strongly encouraged to significantly edit and rewrite the manuscript to improve clarity of presentation, understanding what has been done and the logic behind the experiments.

1) PDAC is a highly heterogeneous disease, but the authors have studied fibroblasts from only 3 tumors, and they seem to have presented data from only 1 tumor (although they may have repeated with fibroblasts from the other 2 tumors these data aren't shown). The authors are encouraged to re-write the manuscript to show very clearly the data from the fibroblasts from the 3 different patients. This would make a more understandable story but also give clearer insights into the heterogeneity they might be seeing (i.e. are there differences from the 3 tumours when they have characterised the fibroblasts in depth?).

2) The human data, particularly from PDAC patients, need to be clarified. It is unclear why 10 PDAC patients (and 10/11 RCC patients) are excluded. However, the most obvious problem is that since there was no effect of stromal active α5β1 levels on OS, the authors have compared RFS – but apparently half of the cohort had a RFS of 0 (even though many have a OS of years!). Combined with the seemingly arbitrary cut-off for low vs. high (rather than above and below median which might be considered standard), this effectively leaves 3 patients with low stromal active α5β1 at 3D-adh. The authors then claim RFS benefits of about 11 months with high stromal active α5β1 at 3D-adh, but the median of those patients is about 11 months, compared with the survival of the 3 'low' patients at 10, 10 and 19 months.

In addition, despite assessing levels of α5β1 away from 3D-adh, and pFAK and pSmad, the relationship to OS or RFS is not shown or discussed, leading the reader to presume the data were negative. Far from supporting the conclusions of the in vitro work, this section raises questions as to the relevance to cancer.

It also appears highly likely that the patient samples used in Figure 13 etc. represent a subset of pancreatic cancer patients (i.e. the ~20% that are candidates for surgical resection). Thus, they may not fully represent the patient population and some of the most aggressive forms of the disease. Some discussion of this seems appropriate.

3) The TGF-β experiments described starting in Figure 1 are interesting and insightful but also raise a number of important questions that are not adequately addressed or discussed in the manuscript in its current form. For instance, what is really changing when TGF-β signaling is blocked? Are the activated myofibroblasts in general reverting to a more phenotypically quiescent state, is ECM secretion changing resulting in fundamentally different ECM composition, are the CAFs simply less contractile or polarized etc.? While this reviewer fully understands that each of these factors cannot be examined in detail in the current work, a more encompassing discussion of these potential shifts would improve the manuscript. Likewise, if data exists from evaluating these cultures over time it would strengthen the manuscript. Further, the data suggests that the differential influence of the N v D ECM is largely due to the differences in ECM alignment. Can a case be made that this is the dominant factor or one of the critical factors?

4) How effective are the employed antibodies at reducing integrin activity in each instance? The consistently intermediate (trending yet non-significant) levels of SMA following inhibition of alpha5-beta1 are a bit perplexing and generate some concern the that inhibition is simply weak. As such the CRISPR data is appreciated, however it would appear to be much more insightful to also have an approach to knockdown alpha5-beta1 levels.

5) Currently, very little mechanistic insight is provided into the underlying mechanism of altered integrin traffic. While defining the exact mechanism of how a5v5 inhibition drives increased a5b1 localization to 3D adhesion is probably too much to ask a few obvious things could be done. i) Given the high level of expertise in imaging demonstrated in the manuscript the authors should be able to describe the intracellular localization of the active a5b1 with more accuracy (are these early endosomes, recycling endosomes…). ii) There is a wealth of literature in cancer cells showing that inhibition of avb3 integrin with osteopontin, cilengitide or cRGD drives a5b1 recycling from intracellular endosomes to the membrane. Could a similar mechanism be involved in the ALULA-driven increase of a5b1 in 3D adhesions? This should be at least discussed.

---

## [Author Response]

*Essential revisions:*

*The reviewers found your work to be particularly relevant to the field right now as there is some controversy about the role of fibroblasts in disease progression and response to therapy. They felt that the data presented in the submitted manuscript are very timely and may help clarify the dynamics of cancer stroma crosstalk in pancreatic carcinoma. However, all of the reviewers found the manuscript to be extremely dense and especially the Results section very hard to follow. Their enthusiasm was somewhat hampered by the lack of clarity in several instances (please see specific points below) that make portions of the manuscript difficult to read/interpret.*

We acknowledge the initially submitted manuscript was extremely dense, reflecting the large amount of data included (which has now expanded further, as outlined below), and the complexity of the analysis. We have completely rewritten the manuscript, paying particular attention to the Results section, and hope that the text is now much more clear.

*Furthermore, they were critical about the clinical relevance and strength of the human data.*

This is a very important point. Due to the efforts of our clinical collaborators, we were able to double the number of patients represented in the PDAC cohort (see specific answers to points made by each reviewer below for particulars). The inclusion of these additional specimens, presented in a new Figure (Figure 13 and Figure 13—figure supplement 1–Figure 13—figure supplement 2), now allows us to draw statistically robust conclusions that support our overall hypothesis.

*And the representability of the limited number of fibroblasts for PDAC as a whole.*

Our data now includes quantitative assessments of cells harvested from 5 PDAC patients. These include 2 patients for whom we obtained matching normal fibroblastic cells and cancer associated fibroblasts (CAF), as well as one additional patient-derived set of CAFs and two unpaired normal pancreatic stellate cells. This made up for 7 different populations of primary cells being assessed (in contrast to the previous focus placed mostly in only one pair in the original submission). We tested all normal cells for their response to desmoplastic-ECM (D-ECM) and tested all D-ECMs for their phenotype and function, and for each, evaluated the ability of a TGFbeta receptor inhibitor during matrix production to reverse their ability to produce functional D-ECM (see new Table 2 and new Figure 2—figure supplement 1). These added analyses buttressed the idea that the signaling we observe is generalizable for describing interactions between naïve pancreatic stellate cells and CAF-produced, PDAC-associated D-ECMs.

*In addition, to the specific points raised below the authors are strongly encouraged to significantly edit and rewrite the manuscript to improve clarity of presentation, understanding what has been done and the logic behind the experiments.*

As noted above, the entire manuscript has been rewritten to increase clarity.

1) PDAC is a highly heterogeneous disease, but the authors have studied fibroblasts from only 3 tumors, and they seem to have presented data from only 1 tumor (although they may have repeated with fibroblasts from the other 2 tumors these data aren't shown). The authors are encouraged to re-write the manuscript to show very clearly the data from the fibroblasts from the 3 different patients. This would make a more understandable story but also give clearer insights into the heterogeneity they might be seeing (i.e. are there differences from the 3 tumours when they have characterised the fibroblasts in depth?).

We thank the reviewers for this comment. We have now included quantitative data comparing phenotypes from 3 sets of CAFs and 4 sets of naïve fibroblasts for multiple reported parameters, clearly noting that all specimens are analyzed. Statistical analysis of phenotypes indicates limited issues with heterogeneity, although certainly, the differences observed in this study between signaling pathways required for CAF functions versus regulation of naïve fibroblast-to-myofibroblastic activation may explain some *in vivo* observed stroma heterogeneity. These tests included measurement of matrices produced in the presence or absence of inhibition of the TGFbeta pathway for their ability to induce myofibroblastic features (including levels and localization of α-smooth muscle actin) in naïve fibroblastic stellate cells. Results from the multiple cell-matrix combinations (new Table 2 and new Figure 2—figure supplement 1)are now bolstered by results from other model systems (revised Figure 3 and Figure 11 together with Figure 11—figure supplement 1–Figure 11—figure supplement 6 as well as revised Table 5) and supported by statistically significant analysis of a larger number of clinical PDAC samples that show similar signaling relationships (new Figure 13 and new Figure 13—figure supplement 1–Figure 13—figure supplement 2).

*2) The human data, particularly from PDAC patients, need to be clarified. It is unclear why 10 PDAC patients (and 10/11 RCC patients) are excluded.*

A) A small number of patients were excluded from the cohorts analyzed in the initial submission because we did not have annotation for their OS, RFS or DSS. The numbers of patients used for each analysis are now clearly annotated in a new Table 6. In addition, please note that we were able to extend analysis to include additional PDAC patients, so that the total number of specimens analyzed in this revision now totals 128 tumors (versus 65 in the original submission), with new results presented in Figure 13 and Figure 13—figure supplement 1–Figure 13—figure supplement 2. We have available OS information for all PDAC patients, and for 116/126 RCC patients.

B) In regard to analysis of RFS or DSS, we do not have any annotation regarding evidence of disease following surgery for 26 PDAC patients, so that only 102/128 PDAC patients are analyzed for this property in this revision. Similar lack of annotation caused us to only analyze DSS in 115/126 RCC patients.

C) We have made a comprehensive summary of available data (regarding age, gender and other clinical and history particulars of the complete PDAC and RCC patient cohorts, as well as more than 50 biomarker readouts from the simultaneous multi-color immunofluorescent (SMI) approach, available online in a linked database (https://www.foxchase.org/sites/fccc/files/assets/cukierman_Franco-Barraza%20SMIA-CUKIE-Dec-2016.xlsx), to ensure complete transparency and to support future in silico searches.

*However, the most obvious problem is that since there was no effect of stromal active α5β1 levels on OS, the authors have compared RFS – but apparently half of the cohort had a RFS of 0 (even though many have a OS of years!).*

We acknowledge the issue with RFS was problematic, and in the modified Results section discuss limitations of the dataset directly. We believe that part of the difficulty in achieving significance lies in size of the sample cohorts, and part in the fact that localization of the active integrin conformation to intracellular vesicles versus at the PM emerged in this study as a critical factor, which it is more difficult to robustly image in primary tumor specimens. In this revision, we have enlarged our analytic cohort of PDAC specimens (making the RFS data more reliable), and analyzed more markers that are less subject to localization control, and hence potentially more robust. We now clearly show that active TGFbeta signaling, represented by levels of pSMAD2/3 at stromal (but not tumor) locations, correspond with significantly lower OS times. Statistical analysis indicates that SMAD2/3 and active α_5_β_1_-integrin at 3D adhesion positive areas were each separately significant in prediction of RFS, even given the limitations of the data set. We believe that future studies using larger numbers of samples and examining additional markers will result in significant protective OS outcomes, since we did observe some trends which did not reach statistical significance.

*Combined with the seemingly arbitrary cut-off for low vs. high (rather than above and below median which might be considered standard), this effectively leaves 3 patients with low stromal active α5β1 at 3D-adh. The authors then claim RFS benefits of about 11 months with high stromal active α5β1 at 3D-adh, but the median of those patients is about 11 months, compared with the survival of the 3 'low' patients at 10, 10 and 19 months.*

Addressing the statement about seemingly arbitrary partitioning of the data used to create Kaplan-Meier curves, in the “Bioinformatics and Statistics” section of Materials and methods, we now clearly state how we use a unified CART framework that embeds recursive binary partitioning into the theory of permutation tests. As defined by our biostatistician, significance-testing procedures were applied to determine whether no significant association between any of the clinical variables and the response could be stated, or whether the recursion would need to stop. This approach is commonly used for this type of analyses by clinical biostatisticians. Median partitioning of the data was also queried in the univariate and multivariate testing we conducted, but rendered no significant results with regards to the 7 markers.

*In addition, despite assessing levels of α5β1 away from 3D-adh, and pFAK and pSmad, the relationship to OS or RFS is not shown or discussed, leading the reader to presume the data were negative. Far from supporting the conclusions of the* in vitro *work, this section raises questions as to the relevance to cancer.*

In the first submission of the manuscript, these data were not significant for PDAC and hence not discussed. However, due to the much-enhanced PDAC patient cohort analyzed for this revision, some of these values are indeed now significant for prediction of OS, including notably the stromal increases in pSMAD2/3. These results are now shown in new Figure 13 and are included in the new Discussion section. Moreover, three independent SMIA-CUKIE quantitative analyses each implicated α5β1 activity and in particular its stromal localization at 3D-adhesions (new Figure 13—figure supplement 2), and nowprovide a stronger support for the relevance of *in vitro* data to pancreatic cancer in patients. Similarly, this revision now includes all significant Kaplan-Meier curves analyzing biomarkers in RCC patients, and better supports the pathophysiological validity of the *in vitro* 3D stroma system.

*It also appears highly likely that the patient samples used in (Figure 13 etc. represent a subset of pancreatic cancer patients (i.e. the ~20% that are candidates for surgical resection). Thus, they may not fully represent the patient population and some of the most aggressive forms of the disease. Some discussion of this seems appropriate.*

We agree that PDAC patients that are surgical candidates typically do not have the most advanced disease, as we now clearly state in the Results and Legends to Figure 13 and to Figure 13—figure supplement 1–Figure 13—figure supplement 4, and comment on in the Discussion. In this context, we view it as compelling that in spite of the bias of our analytic cohorts to earlier stage patients, differences in OS and/or RFS were evident correlating with some of the markers we analyzed.

*3) The TGF-β experiments described starting in Figure 1 are interesting and insightful but also raise a number of important questions that are not adequately addressed or discussed in the manuscript in its current form. For instance, what is really changing when TGF-β signaling is blocked? Are the activated myofibroblasts in general reverting to a more phenotypically quiescent state, is ECM secretion changing resulting in fundamentally different ECM composition, are the CAFs simply less contractile or polarized etc.? While this reviewer fully understands that each of these factors cannot be examined in detail in the current work, a more encompassing discussion of these potential shifts would improve the manuscript. Likewise, if data exists from evaluating these cultures over time it would strengthen the manuscript. Further, the data suggests that the differential influence of the N v D ECM is largely due to the differences in ECM alignment. Can a case be made that this is the dominant factor or one of the critical factors?*

We agree that the mechanism of TGFbeta regulation is important. Given the constraints of fitting additional information into an already very long manuscript, we have now included a new paragraph in the Discussion providing context from recent studies addressing a proposed positive feedback loop between TGFbeta accessibility and regulation of myofibroblastic activation during chronic fibrosis (and desmoplasia). The Discussion includes examples in which this signaling has been directly associated to anisotropic ECM production as well as ECM alignment features being independently indicative of deteriorating patient outcomes (e.g., publications by Klingberg (JCB, 2014), Achterberg (J Invest Dermatol, 2014), Conklin (AJP, 2011) and Bredfeldt (J Pathol Inform., 2014). Our data does not allow us to make definitive statements about changes of protein composition of ECMs arising from TGFbeta inhibition, although we do state that induction of collagen I and the differential spliced form of fibronectin are known traits of desmoplasia.

*4) How effective are the employed antibodies at reducing integrin activity in each instance? The consistently intermediate (trending yet non-significant) levels of SMA following inhibition of alpha5-beta1 are a bit perplexing and generate some concern the that inhibition is simply weak. As such the CRISPR data is appreciated, however it would appear to be much more insightful to also have an approach to knockdown alpha5-beta1 levels.*

A) We thank the reviewers for this comment, as it resulted in a significant change to the study (see below comments about newly generated mutant fibroblasts). Regarding the specificity of the functional antibodies: these antibodies are broadly used and very well accepted in the field. We now state this fact and cite references to the work that defined the functional properties of these valuable reagents.

B) In spite of the specificity of the reagents, we now also clearly acknowledge at multiple places in the text the fact that inhibition using antibodies is transient, while genetic ablation negates receptor expression, which may be impact alternative functions, making the two types of assay non-equivalent. For example, the revised text now includes a statement that gross expression of several of the integrins analyzed is needed for naïve-to-myofibroblastic activation in response to D-ECM, while α_5_β_1_-integrin activity, regulated by α_v_β_5_-integrin, opposes D-ECM stimulation of this transition.

C) We acknowledge the reviewer’s point regarding the intermediate levels of α-SMA, but we do not believe that the reason for α-SMA levels being less affected than its F-actin localization are due to antibodies presenting a weak inhibitory effect. Rather, we believe that signaling between alphavbeta5 and alpha5beta1 is important for maintaining α-SMA at stress fibers, but not for its expression, in naïve cells responding to D-ECM, while levels of α-SMA expression instead depend on TGFbeta signaling, and that the intermediate levels we see reflect the complexity of the signaling stimuli. This fact is generally supported by the findings that alphaSMA mRNA levels are unaffected by overnight D-ECM incubation of naïve fibroblasts while high levels of α-SMA expression are necessary in CAFs producing D-ECMs. The revised Discussion section now also cites work related to the complexity of control of α SMA: for instance, evidence that α_5_β_1_-positive elongated ‘super mature’ adhesion structures, known to control tension-dependent recruitment of αSMA to stress fibers while not affecting protein levels, play important functional roles in myofibroblasts.

D) We agree with the reviewers asking for knockout (CRISPR) inclusion of alpha5 fibroblasts (both in naïve fibroblastic stellate cells and in CAFs). Indeed, we now include not only these mutants, but also mutants for alphav as well as beta3 integrins, in both normal fibroblasts and CAFs. Extensive data comparing these new models are presented in new Figure 3—figure supplement 1–Figure 3—figure supplement 2, Figure 4—figure supplement 2–Figure 4—figure supplement 3, Figure 5—figure supplement 5, Figure 7, Figure 7—figure supplement 1, Figure 10 and Figure 10—figure supplement 1. Some of the added text emphasizes the fact that results using mutants recapitulated results using inhibitors, while others highlight specific differences, providing improved mechanistic detail to the study.

*5) Currently, very little mechanistic insight is provided into the underlying mechanism of altered integrin traffic. While defining the exact mechanism of how a5v5 inhibition drives increased a5b1 localization to 3D adhesion is probably too much to ask a few obvious things could be done. i) Given the high level of expertise in imaging demonstrated in the manuscript the authors should be able to describe the intracellular localization of the active a5b1 with more accuracy (are these early endosomes, recycling endosomes…). ii) There is a wealth of literature in cancer cells showing that inhibition of avb3 integrin with osteopontin, cilengitide or cRGD drives a5b1 recycling from intracellular endosomes to the membrane. Could a similar mechanism be involved in the ALULA-driven increase of a5b1 in 3D adhesions? This should be at least discussed.*

A) We have added a new section to the revised manuscript to analyze the intracellular location(s) of alpha5beta1 integrin. This analysis indicated accumulation of this activated integrin in late endosomes (Rab7- and Rab11-positive) and multivesicular (CD81-positive) endosomes (new Figure 9), but not early endosome positive for AAE1 and Rab5. In addition, in new electron microscopy images using the same experimental conditions, we utilized double immunogold to detect 3D-adhesion locations versus active alpha5beta1 (new Figure 8—figure supplement 1). This analysis indicates the intracellular integrin activity induced by D-ECM by naïve fibroblastic cells does not correspond to clatherin coated generated endosomes, and that multivesicular endosomes positive with this integrin are lost upon beta5integrin ablation.

B) We thank the reviewers for bringing to our attention previous work showing integrin regulation in receptor recycling. We have now cited relevant literature, and included a statement to the fact that integrin binding ligands such as soluble RGD, osteopontin and cilengitide have been shown to induce integrin crosstalk between α_v_β_3_ and α_5_β_1_ resulting in the association of α_5_β_1_ with Rab-coupling protein, which regulates its localization concomitant with EGFR. Besides citing the work that provided this data, we mention the possibility that the use of antibodies to inhibit avb5 integrin activity in this study may have triggered activation of the Rab cellular trafficking machinery, which is otherwise restricted by α_v_β_5_ activity during tumor-associated desmoplasia, which could explain the observed accumulation of α_5_β_1_ to late endosomes.